# Impacts of flocculation on the distribution and diagenesis of iron in boreal estuarine sediments

Tom Jilbert[1,2], Eero Asmala[1,2,3], Christian Schröder[4], Rosa Tiihonen[1,2], Jukka-Pekka Myllykangas[1,2], Joonas J. Virtasalo[5], Aarno Kotilainen[5], Pasi Peltola[6], Päivi Ekholm[7], and Susanna Hietanen[1,2]

[1]Department of Environmental Sciences, Faculty of Biological and Environmental Sciences, University of Helsinki, P.O. Box 65, 00014 University of Helsinki, Finland
[2]Tvärminne Zoological Station, University of Helsinki, J.A. Palménintie 260, 10900 Hanko, Finland
[3]Department of Bioscience -Applied Marine Ecology and Modelling, Aarhus University, Frederiksborgvej 399, 4000 Roskilde, Denmark
[4]Biological and Environmental Sciences, Faculty of Natural Sciences, University of Stirling, Stirling FK9 4LA, Scotland, United Kingdom
[5]Marine Geology, Geological Survey of Finland (GTK), P.O. Box 96, 02151 Espoo, Finland
[6]Boliden Rönnskär, 932 81 Skelleftehamn, Sweden
[7]Department of Food and Environmental Sciences, P.O. Box 66, 00014 University of Helsinki, Finland

*Correspondence to*: Tom Jilbert (tom.jilbert@helsinki.fi)

**Abstract.** Iron (Fe) plays a key role in sedimentary diagenetic processes in coastal systems, participating in various redox reactions and influencing the burial of organic carbon. Large amounts of Fe enter the marine environment from boreal river catchments associated with dissolved organic matter (DOM) and as colloidal Fe oxyhydroxides, principally ferrihydrite. However, the fate of this Fe pool in estuarine sediments has not been extensively studied. Here we show that flocculation processes along a salinity gradient in an estuary of the northern Baltic Sea efficiently transfer Fe and OM from the dissolved phase into particulate material that accumulates in the sediments. Flocculation of Fe and OM is partially decoupled, likely due to the presence of discrete colloidal ferrihydrite in the freshwater Fe pool which responds differently from DOM to estuarine mixing. Further decoupling of Fe from OM occurs during sedimentation. While we observe a clear decline with distance offshore in the proportion of terrestrial material in the sedimentary particulate organic matter (POM) pool, the distribution of flocculated Fe in sediments is modulated by focusing effects. Labile Fe phases are most abundant at a deep site in the inner basin of the estuary, consistent with input from flocculation and subsequent focusing. The majority of the labile Fe pool is present as Fe (II), including both acid-volatile sulfur (AVS)-bound Fe and unsulfidized phases. The ubiquitous presence of unsulfidized Fe (II) throughout the sediment column suggests Fe (II)-OM complexes derived from reduction of flocculated Fe (III)-OM, while other Fe (II) phases are likely derived from the reduction of flocculated ferrihydrite. Depth-integrated rates of Fe (II) accumulation (AVS-Fe + unsulfidized Fe (II) + pyrite) for the period 1970–2015 are greater in the inner basin of the estuary with respect to a site further offshore, confirming higher rates of Fe reduction in near-shore areas. Mössbauer [57]Fe spectroscopy shows that refractory Fe is composed largely of superparamagnetic Fe (III), high-spin Fe (II) in silicates, and, at one station, also oxide minerals derived from past industrial activities. Our results highlight that the cycling of Fe in boreal estuarine environments is complex, and that the partial decoupling of Fe from OM during flocculation and sedimentation is key to understanding the role of Fe in sedimentary diagenetic processes in coastal areas.

**Copyright statement**

The authors certify the following:

- All co-authors have approved publication of the manuscript and authorized the corresponding author to enter into copyright agreements for publication in *Biogeosciences*

- The work described has not been published before

- The work described is not under consideration for publication elsewhere

- The work does not contain any previously copyrighted material

# 1 Introduction

Iron (Fe) is present in marine and freshwater sediments in a wide range of phases. Reactive Fe minerals, such as oxides, sulfides, phosphates and carbonates, are involved in diagenetic reactions in sediments and consequently influence the cycling of carbon and nutrients (e.g., Berner, 1970; Slomp et al., 1996a,b; Lovley et al., 2004; Jilbert and Slomp, 2013; Kraal et al.,
2015; Robertson et al., 2016). Iron has also recently been shown to stabilize organic carbon in sediments, promoting carbon burial (Lalonde et al., 2012; Shields et al., 2016). Hence, the lateral and vertical distribution of Fe in sediments is important for broader biogeochemical cycles. Critical to understanding the distribution of sedimentary Fe is a knowledge of the processes converting Fe between its various forms, and how they vary spatially in aquatic systems.

In boreal terrestrial environments, Fe is released during the chemical weathering of Fe-bearing minerals in soils. These include
silicates in the fine fraction of till (Lahermo et al., 1996), and, especially in areas overlain with late- or post-glacial lacustrine and brackish water sediments, previously deposited reactive Fe minerals (Virtasalo and Kotilainen, 2008). During weathering under oxic conditions in the absence of organic ligands, Fe (II) is quickly oxidized to Fe (III), which in turn precipitates as oxides (Schwertmann and Taylor, 1977). Typically, the first-formed oxide mineral is amorphous, labile ferrihydrite. The structure of ferrihydrite is still debated, but a recent study suggested the relatively FeO-rich (and Fe-OH poor) formula of
$Fe_{10}O_{14}OH_2.nH_2O$ to be most accurate (Hiemstra, 2013). Ferrihydrite may subsequently mature into crystalline, refractory oxides such as goethite and hematite (Raiswell, 2011). Although such maturation is rapid in tropical and temperate systems, under the cold, low pH conditions of boreal aquatic environments its half-life may be several years (Schwertmann et al., 2004). However, goethite may also form directly during the oxidation of Fe (II) in sedimentary environments (e.g. van der Zee et al., 2003).

Weathering of Fe in boreal systems also frequently occurs under anoxic conditions, in the presence of dissolved organic compounds such as humic and fulvic acids (Krachler et al., 2016), for example in peatland environments. These compounds are effective chelators of dissolved Fe, and form complexes with Fe (II) in anoxic soil solution (Sundman et al., 2014). Such complexes are typically nanoparticulate–colloidal in size and hence pass through 0.2–0.45 μm pore-size filters, to operationally classify as dissolved material. The stability of Fe (II)-organic complexes (Fe (II)-OM) in river systems depends on the
concentration of chelating organic compounds and the time available for oxidation (Ingri and Conrad, 2015). Typically, dissolved Fe in upstream areas of boreal catchments consists of both Fe (III)-organic complexes (Fe (III)-OM) and ferrihydrite (Neubauer et al., 2013). Ferrihydrite itself is nanoparticulate (Raiswell, 2011), and its high surface area favors continued association with DOM colloids via sorption (Dzombak et al., 1990; Eusterhues et al., 2008). However, the fraction of riverine Fe present as discrete ferrihydrite particles has been shown to increase with rising pH, as is typically observed with increasing
stream order towards the coastal zone (Neubauer et al., 2013).

In estuarine environments, elevated electrolyte strength along salinity gradients induces the flocculation of DOM (Sholkovitz et al. 1978) and Fe (Boyle et al., 1977) from river waters. This phenomenon is usually explained by the cation-induced aggregation of colloidal humic substances, which carry a net negative surface charge (Eckert and Sholkovitz 1976). Any Fe associated directly with DOM is therefore expected to aggregate passively during this process. Flocculation of DOM is typically selective for humic substances of high molecular weight and larger colloidal particle size (Uher et al., 2001; Asmala et al., 2014). Consequently, the 'truly dissolved' DOM-associated Fe which passes through the flocculation zone of estuaries (e.g., Dai and Martin, 1995) is associated with DOM of lower molecular weight and smaller colloidal particle size (e.g., < 3 nm), most likely in the form of fulvic acids (Stolpe and Hassellöv, 2007). This component may be more substantial than previously thought and hence play a role in providing Fe as a micronutrient to the oceans (Kritzberg et al., 2014; Krachler et al., 2016). However, the majority of riverine dissolved Fe is retained in estuaries (Raiswell, 2011), implying an important role for flocculation in the removal of dissolved Fe. Importantly, the flocculation behavior of Fe in boreal estuaries appears to differ from that of bulk DOM (Asmala et al., 2014), which may be partly related to factors influencing the discrete ferrihydrite-bound Fe component rather than Fe directly associated to DOM. These factors include pH gradients and the concentration of suspended clay material (Forsgren et al., 1996)

Together with the deposition of riverine particulate Fe close to river mouths (Poulton and Raiswell, 2002, Li et al., 2016), flocculation may thus be expected to act as an important mechanism of Fe sedimentation in the coastal zone. The role of flocculation may be particularly enhanced in boreal estuarine systems due to the high DOM and dissolved Fe concentrations in this region (Kritzberg et al., 2014). However, few studies have attempted to investigate the connection between flocculation and the Fe distribution in boreal estuarine sediments. This is a significant gap in existing knowledge, since an increasing number of studies have demonstrated the importance of reactive Fe minerals in sedimentary diagenesis in boreal coastal areas, including their roles in the anaerobic oxidation of methane (AOM) (Slomp et al., 2013; Egger et al., 2015a) and in phosphorus retention in sediments (Reed et al., 2011; Norkko et al., 2012; Egger et al., 2015b). Furthermore, Fe has recently been suggested to play an important role in carbon burial (Lalonde et al., 2012; Shields et al., 2016) and nitrogen cycling (Robertson et al., 2016) in marine sediments.

Understanding the distribution of Fe minerals in boreal sedimentary environments will improve our knowledge of the broader role of Fe in sediment biogeochemistry. Here we present a combined study of water column, sediment and pore water chemistry in a non-tidal estuarine system in the northern Baltic Sea, to investigate the impact of flocculation on the distribution of Fe in boreal coastal sediments and the subsequent diagenetic transformations during burial. In the estuarine water column, we study the distribution of dissolved and particulate Fe and organic matter, to assess the transfer of these components from the dissolved to particulate phase along the salinity gradient. Using sediment core data from selected locations, we show how processes in the water column control the lateral distribution of labile and refractory Fe, and organic matter, in estuarine sediments. Finally,

we demonstrate how the lateral distribution of labile Fe, together with salinity gradients, influences the vertical diagenetic zonation of the sediments along the estuarine transect and net rates of Fe transformations in the sediment column.

## 2 Study location

The Finnish coastline of the western Gulf of Finland and Archipelago Sea (northern Baltic Sea) is characterized by a mosaic of islands and small bays, intersected by a network of channel-like, non-tidal estuaries (Fig. 1a). The undulating mosaic represents the bedrock surface known as the pre-Cambrian peneplain (Winterhalter et al., 1981), while the channels correspond to fault lines in the bedrock (Hausen, 1948; Virtasalo et al., 2005). The entire area was covered by the Fennoscandian continental ice-sheet during the last glacial (Weichselian) maximum. The ice-margin retreat from the area ca. 12 ka ago was followed by the successive deposition of till, outwash, glaciolacustrine rhythmite, patchily-distributed debrites, postglacial lacustrine clay and brackish-water mud drift (Virtasalo et al., 2007). These deposits provide the source material for mobile Fe in the drainage basins of southern Finland, and each deposit has a distinct Fe mineralogy (Virtasalo and Kotilainen, 2008).

The principal study area is the estuary of the Mustionjoki river and its adjacent archipelago (Fig. 1a). This river and its estuary appear under several alternative Finnish-, Swedish- and English- language names in cartographic material and the scientific literature, including Karjaanjoki (e.g., Asmala et al., 2014), Pohjanpitäjänlahti (e.g., Virta, 1977), Pojoviken (e.g., Niemi, 1977) and Pojo Bay. The First Salpausselkä ice-marginal formation intersects the estuary close to the town of Ekenäs (Fig. 1a). The First Salpausselkä forms a shallow sill of < 10m water depth, separating the inner basin of the estuary (maximum water depth 39 m) from the slope of the archipelago towards the open Gulf of Finland (Fig. 1b). A pronounced estuarine circulation is observed, with continuous freshwater outflow at the surface and intermittent brackish water inflows at depth. Inflows typically occur in winter, when discharge from the river is at a minimum (Virta, 1977). Subsequent stagnation of the deep water masses leads to oxygen depletion, with hypoxic conditions (oxygen concentrations < 2mg L$^{-1}$) observed during the late summer and autumn months of some years (Niemi, 1977).

A blast furnace located near the town of Åminnefors at the mouth of the Mustionjoki river (Fig. 1a) was active from the late 19[th] century until 1977. The blast furnace and associated waste materials serve as a potential source of anthropogenic Fe pollution to the estuary.

## 3 Materials and methods

### 3.1 Hydrographic profiling

Over 2 days of sampling onboard R/V Saduria and R/V J.A. Palmén in June 2015, water column temperature, salinity and dissolved oxygen profiles were collected at Stations A–K in the estuary of the Mustionjoki river and adjacent archipelago, using multiparameter water quality sondes (YSI[TM] CTD with optical oxygen sensor and Valeport MiniCTD). The 11 vertical

profiles for each parameter were interpolated into cross-sectional contour plots using the Sigmaplot$^{TM}$ software package (Fig. 1b). Station A is situated at the mouth of the Mustionjoki river, while Station K is situated 33 km due S of the river mouth (~40 km absolute distance along transect) in the open Gulf of Finland. The precise locations of the stations were selected on the basis of suitability for sediment sampling; all are situated in bathymetric depressions of 10–100 m-lateral scale, where soft

sediments are expected to accumulate.

### 3.2 Sampling and analysis of suspended particulate organic matter

During the sampling campaign in June 2015, discrete water samples were collected at 5 m depth intervals at Stations A–K in the estuary of the Mustionjoki river and adjacent archipelago, using a 5 L Limnos$^{TM}$ water sampler. Water samples were transferred onboard to acid-washed polyethylene bottles, stored at 4°C and filtered within 48 hours of sampling at Tvärminne

Zoological Station, Hanko, Finland. One 500 ml aliquot of each sample was filtered through pre-weighed, pre-combusted (450°C for 4 h) Whatman$^{R}$ GF/F filters (nominal pore size 0.7 µm). Total carbon ($C_{tot}$) and total nitrogen ($N_{tot}$) on the filters, and the stable isotopic ratio of carbon relative to the Vienna Pee Dee Belemnite ($\delta^{13}C$), were estimated by thermal combustion elemental analysis-mass spectrometry (TCEA-MS) at Tvärminne Zoological Station. Precision and accuracy of all parameters as checked by in-house and reference standards was < 2.5% relative standard deviation (RSD). Replicate analysis yielded

results that were identical within measurement error (see Figure R1 in the Response to Referee supplement in the associated Discussion paper). Particulate inorganic carbon and nitrogen are assumed insignificant in this setting, hence $C_{tot}$ and $N_{tot}$ are assumed equal to organic carbon and nitrogen, respectively ($C_{org}$ and $N_{org}$).

### 3.3 Sampling and analysis of particulate and dissolved Fe

During the sampling campaign in June 2015, two additional 250 ml aliquots of water from each sample were filtered through

parallel Whatman$^{R}$ Nuclepore track-etched polycarbonate membrane filters (pore size 0.4 µm). Filtrate was collected in 15 ml centrifuge tubes and acidified to 1M $HNO_3$ for analysis of dissolved Fe and other elements by ICP-MS at University of Helsinki Department of Geosciences and Geography. Filters were freeze-dried and acid-digested in Teflon$^{TM}$ vessels (digestion in 2.5 mL HF + 2.5 mL $HClO_4/HNO_3$ at volumetric ratio 3:2, reflux at 90°C for 12 h, followed by evaporation of acids until gel texture and re-dissolution in 20 mL 1M Suprapur$^{R}$ $HNO_3$). Analysis of particulate Fe (among other elements) in the resulting

digests was performed by ICP-OES at University of Helsinki Department of Food and Environmental Sciences (precision and accuracy < 5% RSD as determined by in-house and reference standards). All values are reported as µmol L$^{-1}$.

### 3.4 Sediment sampling and preparation

In September 2014 onboard R/V Saduria, sediments were collected from Stations A–K on the Mustionjoki estuary transect using a GEMAX$^{TM}$ short gravity corer (internal diameter 9 cm, core length 30–60 cm). Four to five sediment slices of 2 cm

thickness, evenly spaced with depth over the full length of the core, were obtained from each station (e.g., Station K: 0–2 cm;

8.5–10.5 cm, 17–19 cm, 25.5–27.5 cm, 34–36 cm). During sampling campaigns in 2015, GEMAX™ cores were taken from Stations A, D (June) and J (April, June) and sliced completely at 1 cm resolution (0–10 cm depth) and 2 cm resolution (10 cm depth–core base). An additional sample from the surface sediments of the Mustionjoki river, taken close to Station "a" (lower case) in Fig. 1a, was obtained with a grab sampler in September 2015. In all campaigns, whole sediment slices were transferred

immediately to plastic bags, dipped in water to seal the bag, and deposited in a gas-tight jar that was flushed with nitrogen within 1 h of sampling and stored dark at 4°C until further processing. Due to the large volume of tightly-packed sediment in each jar, visible oxidation effects during sampling and storage were minimal.

Subsamples of wet sediment slices were taken from the jars under nitrogen atmosphere, frozen, freeze-dried and homogenized,

and stored in $N_2$-filled gas-tight jars until further processing. Parallel wet samples were stored frozen at -20°C. Water content (% water by weight) and porosity (% water by volume) were estimated from weight loss during freeze drying, assuming a solid-phase density of 2.65 g cm$^{-3}$ (Schulz and Zabel, 2006). The content of salt material in the dried sediment matter was estimated from water content and the measured bottom water salinity at each site.

### 3.5 Analysis of sedimentary organic matter

Selected sediment samples were prepared for analysis of sedimentary organic matter. Sub-samples of dried, powdered sediments were weighed into aluminium capsules. Total sedimentary carbon ($C_{tot}$) and nitrogen ($N_{tot}$), and the stable isotopic ratio of carbon reported relative to Vienna Pee Dee Belemnite ($\delta^{13}C$), were estimated by thermal combustion elemental analysis-mass spectrometry (TCEA-MS) at Tvärminne Zoological Station and University of California, Davis, USA. Precision and accuracy of all parameters as checked by in-house and reference standards was < 2.5% RSD. Sedimentary inorganic carbon

and nitrogen are assumed insignificant in this setting, hence $C_{tot}$ and $N_{tot}$ are assumed equal to organic carbon and nitrogen, respectively ($C_{org}$ and $N_{org}$).

### 3.6 Quantification of organic matter sources

A simple two-component mixing model was applied for a first-order quantification of the relative contributions of terrestrial plant-derived organic material (%$OC_{terr}$), vs. riverine–estuarine phytoplankton (%$OC_{phyt}$), to total organic matter in both water

column and sediment samples. The calculation uses only the molar N/C ratio of organic matter, and end-member values, $N/C_{EM}$, based on the study of Goñi et al. (2003):

$$\%OC_{phyt} = \frac{\left(N/C_{sample} - N/C_{EM-terr}\right)}{\left(N/C_{EM-phyt} - N/C_{EM-terr}\right)} \times 100 \tag{1}$$

$$\%OC_{terr} = 100 - \%OC_{phyt} \tag{2}$$

where $N/C_{EM-terr} = 0.04$, and $N/C_{EM-phyt} = 0.13$. The calculation assumes that plant matter and phytoplankton are the only sources of organic material present in the samples, that their N/C values are spatially and temporally fixed at the end-member values, and that these values do not alter significantly during sedimentation and burial of organic matter. Fields in N/C vs. $\delta^{13}C$ space, also taken from Goñi et al. (2003) and corresponding to riverine–estuarine phytoplankton, marine phytoplankton, and terrestrial
C3 plants respectively, were used in the interpretation of the data.

### 3.7 Analysis of sedimentary Fe, S and Pb by ICP-OES

Selected sediment samples were prepared for ICP-OES analysis. Sub-samples of dried, powdered sediments were weighed into Teflon digestion vessels and digested in an acid cocktail (digestion in 2.5mL HF + 2.5 mL $HClO_4/HNO_3$ at volumetric ratio 3:2, reflux at 90°C for 12 h, followed by evaporation of acids until gel texture and re-dissolution in 20 mL 1M Suprapur$^R$
$HNO_3$). ICP-OES analysis for total Fe, sulfur (S) and lead (Pb), among other elements, was performed at University of Helsinki Department of Food and Environmental Sciences (precision and accuracy < 5% as determined by in-house and reference standards). Total Fe and S from ICP-OES analysis were used in combination with other extraction data to estimate pyrite and residual silicate-bound Fe ("Subsample 3" in Table 2).

### 3.8 Estimate of sedimentation rates using sedimentary Pb profiles

Sedimentation rates for Stations A, D and J were estimated on the basis of total Pb ($Pb_{tot}$) profiles measured on the three GEMAX$^{TM}$ cores from 2015. Each core profile showed a distinct peak in $Pb_{tot}$ (Supplementary Fig. 1) which was assigned to the year 1970 (Renberg et al., 2001; Zillen et al., 2012). A first order estimate of sedimentation rate was calculated assuming constant sedimentation over the period 1970–2015.

### 3.9 Sequential extraction and analysis of sedimentary Fe phases

A set of complementary extraction procedures was performed on sediment samples to assist in the identification of labile and refractory Fe phases (Table 2). The distinction between labile and refractory in this study is made on the basis of the extractions employed, and does not directly translate to reactive vs. non-reactive Fe. For example, refractory phases include crystalline oxides and pyrite, which classify as reactive Fe by common definitions. Initially, for the surface sediment samples from 2014 (0–2 cm), and full downcore profiles (minimum 10 samples) from Stations A, D and J from 2015, the sequential extraction
procedure for Fe described by Poulton and Canfield (2005) was carried out (Table 2, "Subsample 1"). Here, Stages 1 and 2 of this protocol are considered to extract labile Fe, while the remaining components are considered refractory. Sub-samples of dried, powdered sediments were weighed into extraction vessels and a series of reagents was applied sequentially. After each addition, samples were placed in an orbital shaker for the duration of the extraction, then centrifuged at 3000 rpm for 5 minutes before decanting the supernatant. To limit the risk of oxidation affecting the Fe speciation during the extraction, Stages 1–4 of
the extraction procedure were performed under nitrogen atmosphere, and reagents were purged with nitrogen for 30 mins prior

to addition to the samples. All supernatants were analyzed for Fe (among other elements) by Microwave Plasma-Atomic Emission Spectroscopy (MP-AES) at University of Helsinki Department of Geosciences and Geography. Replicate extraction of parallel samples yielded RSD values of < 15% for all stages of the procedure.

Subsequently, 5 additional samples from each of the 2015 cores (Stations A, D, and J) were subjected to a 1 hour room-temperature 1M HCl extraction (Burton et al., 2011), to further investigate the labile Fe phases (Table 2, "Subsample 2"), potentially including Fe (II)-OM and Fe (III)-OM complexes as described in Yu et al. (2015). These samples were taken from wet sediments, frozen shortly after sampling and dried under nitrogen prior to the extraction. Two parallel weighed subsamples were treated with, respectively, 1M HCl and 1 M HCl + 1M hydroxylamine-HCl for 1 hr. All reagents were purged with nitrogen for 30 mins prior to addition to the samples, and the addition of reagents was performed under nitrogen before transfer to an orbital shaker. Both extractions are expected to dissolve all labile Fe (II) and Fe (III) phases. After extraction, the 1M HCl extract may contain both Fe (II) and Fe (III), whereas in the combined extract, the reducing agent hydroxylamine-HCl maintains all dissolved Fe in the divalent state. Fe (II) in each extract was determined spectrophotometrically by the 1,10 phenanthroline method (APHA, 1998), allowing 1M-HCl soluble Fe (II) and 1M HCl-soluble Fe (III) to be deconvolved as follows:

$1M\ HCl\ soluble\ Fe\ (III) = Total\ 1M\ HCl\ soluble\ Fe - 1M\ HCl\ soluble\ Fe\ (II)$ (all units µmol g$^{-1}$)  (3)

A trap was added to the 1M HCl extraction vessel to collect evolved hydrogen sulfide ($H_2S$) released during the dissolution of acid-volatile sulfur (AVS). The trap consisted of an open test tube inside the closed extraction vessel as described in Burton et al. (2008). The test tube was filled with a solution of 0.2 M zinc acetate in 1M NaOH and the extraction vessel was placed on the shaker in vertical orientation. Evolved $H_2S$ from the sample was visibly trapped as a ZnS precipitate in the alkaline solution. The concentration of AVS in the samples was determined by iodometric titration of the ZnS precipitate after redissolution by acidification (Burton et al., 2008). Assuming AVS to be dominantly present as iron monosulfide (FeS), the concentrations of sulfidized and unsulfidized 1M HCl-soluble Fe (II) were estimated thus:

$Sulfidized\ 1M\ HCl\ soluble\ Fe\ (II) = AVS\ Fe$ (all units µmol g$^{-1}$)  (4)

$Unsulfidized\ 1M\ HCl\ soluble\ Fe\ (II) = Total\ 1M\ HCl\ soluble\ Fe\ (II) - AVS\ Fe$ (all units µmol g$^{-1}$)  (5)

where AVS-Fe is Fe bound to AVS. Furthermore, assuming zero-valent sulfur to be a negligible component of total S in sediments from this region (see Yu et al., 2015), the fraction of pyrite ($FeS_2$)-bound Fe was estimated from ICP-OES-derived total S, and AVS, as follows:

$Pyrite\ Fe = \frac{Total\ S - AVS}{2}$  (all units µmol g$^{-1}$)  (6)

where AVS = AVS-Fe due to the 1:1 stoichiometry of FeS. For depth intervals of the 2015 cores where no AVS-Fe data was available, the division of total S into AVS and pyrite was estimated from mean ratios of these components for each core. For the surface sediments from 2014, all S-bound Fe was assumed to be pyrite as per equation 3 of the associated Discussion paper. Total S data were corrected prior to equation (6) for the contribution of crystallized sulfate salts during sample drying. The contribution of sulfate to the total salt content of dried sediment matter was determined for each sediment depth by reference to the corresponding pore water sulfate profile. Crystallized salt contributes approximately 25% of total S in the surface sediments at Station J, where salinity, sulfate concentration and water content are all at maximum values. The contribution declines significantly with depth and at lower salinity sites (e.g., 1.3% of total S in the surface sediments at Station A).

The sum of the 5 stages of the Poulton and Canfield (2005) sequential extraction procedure, plus the estimated contribution of pyrite-bound Fe, were subtracted from total Fe as determined by ICP-OES, to estimate residual (non-soluble) Fe, assumed to be present in unreactive silicate minerals:

$$Residual\ Fe = Total\ Fe - \sum Stages\ \mathbf{1}\ to\ 5 - Pyrite\ Fe \text{ (all units µmol g}^{-1}) \qquad (7)$$

## 3.10 Mössbauer spectroscopy of sedimentary Fe phases

Additional information about Fe phases present in the sediments was gathered from room temperature (RT) $^{57}$Fe Mössbauer spectroscopy (e.g., Murad and Cashion, 2004; Gütlich and Schröder, 2012). A total of seven dried, powdered samples were analyzed at the University of Stirling, U.K. For each sample, 50–100 mg of dried material was placed in acrylic glass tubs with a circular cross section of ~1 cm². Mössbauer spectra were collected using a miniaturized Mössbauer spectrometer (MIMOS II, Klingelhöfer et al. 2003) set up in backscattering geometry, or a standard transmission Mössbauer spectrometer (Wissel, Germany). Both instruments used a $^{57}$Co in Rh matrix radiation source in constant acceleration mode. Two samples (Station A, 0–1 cm and Station D, 0–1 cm) were analyzed in backscattering mode. Peaks in backscatter spectra display as emission maxima (Supplementary Fig. 2). The remaining five samples (river bed sediment from Station 'a', Station A (26–28 cm), Station D (26–28 cm), Station J (0–1 cm) and Station J (30–32 cm)) were analyzed in transmission mode. Troughs in transmission spectra correspond to absorption maxima (Supplementary Fig. 2). All spectra were calibrated against alpha-iron at room temperature. Backscatter Mössbauer spectra were evaluated using an in-house routine (Mbfit) with Lorentzian line profiles, based on the least-squares minimization routine MINUIT (James 2004). Transmission Mössbauer spectra were evaluated using Recoil software (University of Ottawa, Canada) and the Voigt-based fitting routine (Rancourt and Ping, 1991).

Mössbauer spectroscopy allows estimation of the relative contribution of Fe (II) and Fe (III) phases to a bulk sample, due to the distinctly different isomer shift of Fe (II) vs. Fe (III) compounds (see Fig. 7). Allocation of the subspectral components to known (groups of) Fe (II) and Fe (III) phases was performed by comparison of the hyperfine parameters – isomer shift ($\delta$) in mms$^{-1}$, quadrupole splitting ($\Delta$EQ) in mms$^{-1}$, and internal magnetic field (B$_{hf}$) in T – of each subspectrum with those of library

reference spectra (e.g., Stevens et al., 2002) for a range of Fe-bearing compounds. The subspectral components were assigned either to distinct Fe minerals (in the case of unambiguous hyperfine parameters), or to generic groups of minerals characterized by similar isomer shift, quadrupole splitting and internal magnetic field characteristics (e.g., "superparamagnetic (SP) Fe (III)", "silicate-Fe (II)"). Assuming equal recoil-free fractions of total absorbed energy – no f-factor correction was applied – the concentration of each component was estimated directly from its contribution to the area of the Mössbauer absorption/emission spectrum, and bulk Fe (II): Fe (III) ratios for each sample were estimated directly from the total contribution of Fe (II) and Fe (III) components.

Diamagnetic and paramagnetic mineral phases generally display as a doublet (two related peaks) in Mössbauer spectra whereas magnetically ordered phases display as a sextet (six related peaks). It should be noted that size of the crystal domain influences magnetic ordering in some minerals. This is particularly important in the case of goethite, a crystalline oxide which at room temperature displays superparamagnetic behavior (hence a doublet spectrum) for particle sizes < approximately 12 nm, but magnetically ordered behavior (sextet spectrum) for particle sizes > approximately 12 nm (van der Zee et al., 2003). Furthermore, spectral interferences between some components limit the sensitivity of RT Mössbauer spectroscopy for minor components of total sedimentary Fe. For example, the doublet for pyrite ($FeS_2$) shows a strong overlap with that of superparamagnetic Fe (III). To aid the reader, we present a selection of relevant Mössbauer reference spectra alongside the sample data of this study (Fig. 7 and Supplementary Fig. 2).

### 3.11 Pore water sampling

Prior to sediment slicing during the June 2015 campaign, pore water was sampled through pre-drilled holes ($\varnothing$ 4 mm) in the GEMAX™ coring tubes, using Rhizons™ mounted on a purpose-built plastic rack. Two parallel series of samples (vertical resolution 2 cm) were obtained for each core; one for analysis by ICP-OES, the other for analysis of dissolved hydrogen sulfide ($H_2S$). Samples were collected in polyethylene syringes connected directly to the Rhizons™, which were held open by a wooden spacer to create a vacuum. The syringes of the $H_2S$ series were pre-filled with 1 ml 10% zinc acetate solution to trap sulfide as ZnS. All samples were transferred from the syringes to 15 ml centrifuge tubes in the laboratory within 2 h of sampling. From the first series, a sub-sample for ICP-OES analysis was taken immediately and acidified to 1 M $HNO_3$. A parallel GEMAX™ core, pre-drilled with holes of $\varnothing$ 15 mm, was used for sampling for dissolved methane ($CH_4$). 10 ml wet sediment was collected through each hole using a cut-off syringe and transferred to a 65 ml glass vial filled with saturated NaCl solution (Egger et al., 2015a). Vials were capped with a rubber stopper, and a headspace of 10 ml $N_2$ (quality 5.0) was injected using a gas-tight glass syringe. Methane in pore waters was assumed to be quantitatively salted out into the headspace during equilibration (O'Sullivan and Smith, 1970). Samples were stored upside-down until analysis such that headspace gas was not in contact with the rubber stopper.

## 3.12 Pore water analysis

Acidified pore water sub-samples from the first series of Rhizons[TM] were analyzed for total Fe and S, among other elements, by ICP-OES at University of Helsinki Department of Food and Environmental Sciences. Iron is assumed to be present in pore waters as $Fe^{2+}$, while S is assumed to represent $SO_4^{2-}$ only, due to the loss of $H_2S$ during sample acidification (Jilbert and Slomp, 2013). $H_2S$ concentrations in pore water samples from the second series of Rhizons[TM] were determined by spectrophotometry (670 nm) after direct addition of an acidic solution of $FeCl_3$ and n,n-dimethyl-p-phenylenediamine (Cline, 1969; Reese et al., 2011) to the sample vials. This procedure dissolves the ZnS precipitate and immediately complexes S as methylene blue for spectrophotometric analysis. $H_2S$ concentrations were calibrated against a series of standard solutions of $Na_2S•3H_2O$, fixed in Zn acetate in the same manner as the samples. The exact concentration of S in the $Na_2S•3H_2O$ stock solution was determined by iodometric titration (Burton et al., 2008).

For analysis of dissolved $CH_4$, 1 ml headspace gas was sampled from the 65 ml vials using a gas-tight glass syringe. An equivalent volume of salt solution was allowed to flow into the vial through a parallel syringe to equalize pressure in the headspace. Gas samples were then injected into 12 mL gas-tight glass Exetainer[TM] vials (LabCo model 839W). An additional 20 mL $N_2$ gas was injected into the Exetainers to generate overpressure prior to analysis. $CH_4$ concentrations were analyzed using an Agilent Technologies 7890B gas chromatograph (GC) at University of Helsinki Department of Environmental Sciences, equipped with flame ionization detector (FID) at 250°C, oven temperature 60°C, 2.4 m Hayesep Q column with 1/8" connection, 80/100 mesh range, 1.0 mL sample loop and helium carrier gas at flow rate 21 mL min$^{-1}$. Raw peak area data were converted to mole fraction (ppm) using a 4-point linear calibration of standard gas mixtures (certified concentrations $\pm$ 2%) and blanks, analyzed prior to each sample series. Single standards were analyzed after every 10 samples to monitor within-series drift, which was observed to be negligible. Concentrations in the pore water of the original 10 ml wet sediment sample were back-calculated using a best-fit line through the porosity profile from the parallel core for solid-phase sampling.

## 3.13 Calculation of diagenetic process rates

To investigate instantaneous relative rates of anaerobic diagenetic processes, the portion of the sediment column corresponding to 1970–2015 for each of the Stations A, D, and J was used to define a 1-dimensional model domain. For the analysis of pore water profiles, a simplified version of the 1-dimensional mass conservation equation of Boudreau (1997) and Berg et al. (1998) was used:

$$\frac{d}{dx}\left(\varphi D_s \frac{dc}{dx}\right) + R = 0 \tag{8}$$

in which x = depth, $\varphi$ = porosity, $D_s$ = molecular diffusivity of a given species, corrected for tortuosity, salinity and temperature, C = concentration of the species in the pore water or bottom water, and R = the net rate of production of the species due to

diagenetic reactions, expressed per unit volume of sediment. This simplified form of the mass conservation equation assumes steady state conditions, and neglects pore water advective processes, biodiffusion and bioirrigation. The equation states that the net rate of production or consumption of a dissolved species in the pore waters at a given depth can be calculated from the change in gradient of the concentration (i.e. the second derivative of the concentration, C) at the same depth, provided the diffusivity and porosity are known.

The above equation was used to estimate net rates of production and consumption of pore water iron ($Fe^{2+}$), sulfate ($SO_4^{2-}$) and methane ($CH_4$) within discrete depth intervals of the sediment column from the measured pore water profiles of these species. Calculations were performed using the PROFILE software (Berg et al., 1998), which selects the optimum number of discrete depth intervals of production and consumption based on a least squares fitting routine.

10  To study the longer-term relative rates of Fe reduction and sulfate reduction at the same stations, we estimated depth-integrated accumulation rates of reduced solid-phase Fe and S in the sediments within the interval 1970–2015. Assuming solid-phase reactive Fe to accumulate at the sediment water interface as Fe (III) only, we estimated the mean rate of production of reactive Fe (II) from 1970–2015 within the sediment column as:

$$\sum Fe\ (II) = Unsulfidized\ \mathbf{1}M\ HCl\ soluble\ Fe\ (II) + AVS\ Fe\ + Pyrite\ Fe \quad \text{(all units µmol cm}^{-2}\text{ yr}^{-1}) \tag{9}$$

15  A similar exercise was done for sulfur. Assuming all sedimentary sulfur to be derived from the reduction of seawater sulfate, we estimated the mean rate of production of reduced sulfur from 1970–2015 within the sediment column as:

$$\sum S\ = AVS\ + Pyrite\ S \quad \text{(all units µmol cm}^{-2}\text{ yr}^{-1}) \tag{10}$$

Note that pore water $Fe^{2+}$ and $H_2S$ are considered negligible contributors to $\Sigma S$ and $\Sigma Fe$ (II).

Because the individual data series have variable depth resolution, prior to the above calculations, weighted mean mass concentrations of each sediment parameter for the interval 1970–2015 were estimated for each core. Volume concentrations were then estimated from weighted mean mass concentrations as follows:

$$C_v = C_m \times \mathbf{2.65} \times (\mathbf{1} - \varphi) \tag{11}$$

where $C_v$ is weighted mean volume concentration in µmol cm$^{-3}$, $C_m$ is weighted mean mass concentration in µmolg$^{-1}$, and 2.65 is the density of solid matter in sediments in g cm$^{-3}$.

### 3.13 Additional supporting data

N/C and $\delta^{13}$C of DOM was measured from surface-water samples from six locations along the Mustionjoki estuary transect (Stations *a–f*, Fig. 1a) during three sampling campaigns (April, August and October) in the year 2011, as reported in Asmala et al. (2014) and Asmala et al. (2016). Sampling locations are given in Table 1.

Surface sediment (0–1 cm) total Fe and Al data was generated for six locations in the estuary of the Paimionjoki river and its adjacent archipelago (Stations L–Q, Fig. 1a). Samples were obtained in August and September 2001 during a 94-Station survey of Archipelago Sea sediments, as reported in the studies of Virtasalo et al. (2005) and Peltola et al. (2011). For comparability with the present study, new subsamples of this material were digested and analyzed as described in Section 3.7. The locations and water depths of Stations L–Q are given in Table 1.

## 4 Results

### 4.1 Hydrography of the transect

At the time of the primary sampling campaign in June 2015, the water column in the estuary of the Mustinojoki river and the adjacent archipelago was strongly stratified. Vertical temperature stratification was evident throughout the transect (Fig. 1b, top), while salinity stratification was also present in the inner basin of the estuary (Fig. 1b, middle). The freshwater input from the Mustionjoki river was sufficient to generate a surface-water lens of salinity 0–2 extending across the entire inner basin north of the First Salpausselkä, which forms the sill at Ekenäs. The halocline shallowed towards the sill, and the salinity isolines between Stations C and G were strongly inclined. Deep waters upstream of the sill showed depleted concentrations of dissolved oxygen relative to surface-water values (7–8 mg L$^{-1}$ vs. 10–11 mg L$^{-1}$).

### 4.2 Dissolved and particulate Fe in the water column

Dissolved Fe concentrations in the surface water at the mouth of the Mustionjoki river in June 2015 (Station A, salinity 1.0) were 1.3 µmol L$^{-1}$ (Fig. 2a). Concentrations decreased offshore to values around 0.02 µmol L$^{-1}$ in the open waters of the Gulf of Finland (Station K). The isolines of [Fe$_{diss}$] in the estuarine water column were inclined similarly to those of salinity (Fig. 1b), with a relatively deep surface layer of Fe-rich waters at the river mouth shallowing towards the sill at Ekenäs. However, surface water salinity and [Fe$_{diss}$] along the transect show a strongly non-linear relationship, indicating non-conservative mixing between river and offshore water with respect to [Fe$_{diss}$], due to removal of Fe$_{diss}$ from solution (Fig. 2b).

Particulate Fe in the water column of the estuary in June 2015 showed a contrasting distribution to that of [Fe$_{diss}$] (Fig. 2a). Although maximum [Fe$_{part}$] was also observed at the river mouth (3.9 µmol L$^{-1}$ in surface waters at Station A), values decreased rapidly within a short distance offshore (Station B surface water = 1.4 µmol L$^{-1}$; Station B, 5 m depth = 0.6 µmol L$^{-1}$). Further

away from the river mouth, [Fe$_{part}$] showed higher values in a zone extending from 15 m depth at Station C to the surface waters at Station G (Fig. 2a), approximately coinciding with the halocline of salinity = 2–4 (Fig. 1b). In the archipelago region of the transect (Stations G–J), [Fe$_{part}$] declined gradually offshore.

## 4.3 Particulate organic matter in the water column

Particulate organic carbon (POC) and nitrogen (PON) concentrations in the water column of the estuary in June 2015 ranged from 5–75 µmol L$^{-1}$ and 0.5–7 µmol L$^{-1}$, respectively, and were consistently highest in surface waters (see Figure R7 in the Response to Referee supplement of the associated Discussion paper). Moreover, surface waters throughout the transect were characterized by a relative enrichment of N (N/C = 0.14–0.17, Fig. 3a). In contrast, deeper waters had lower concentrations of particulate organic matter and a relative depletion of N (N/C = 0.08–0.13). The region close to the river mouth displayed the
most pronounced N/C enrichments anywhere on the transect.

The distribution of $\delta^{13}$C$_{POC}$ showed a general similarity to that of N/C. Relatively depleted values were observed in surface waters (-29‰ – -31‰), with the most depleted values observed close to the river mouth, while deep water values were relatively enriched (-26‰ – -28‰) (Fig. 3a, bottom). One anomalous sample of relatively enriched values (approx. -26‰) was observed in the surface waters at site G, close to the sill at Ekenäs. When N/C and $\delta^{13}$C$_{POC}$ values are plotted in x-y space,
surface water samples for most stations, regardless of salinity, plot close to the riverine end of the riverine–estuarine phytoplankton continuum. At each site, samples from deeper in the water column trend away from this region of the diagram towards the field corresponding to terrestrial C3 plants (Fig. 3b).

## 4.4 Sedimentary organic matter along the transect

Mean total organic carbon (C$_{org}$) contents of the upper 30–50 cm of sediments sampled in September 2014 were highest in the
archipelago (Stations H–K, approximately 4%–5%), followed by the estuary (Stations A–E, approximately 4%) and lowest at the sill (Stations F–G, approximately 2–3%, Fig. 4b). The four samples from site A, at the mouth of the Mustionjoki river, all showed molar N/C ratios of 0.05–0.09 and $\delta^{13}$C$_{org}$ of -26‰ – -29‰, hence plot close to the terrestrial C3 plants field in N/C vs. $\delta^{13}$C$_{org}$ space (Fig. 4a). With increasing distance along the transect, from the estuary through the sill to the archipelago, mean N/C increases and $\delta^{13}$C$_{org}$ becomes more enriched, hence successive stations plot progressively closer to the
riverine–estuarine phytoplankton continuum. Samples from Station K showed molar N/C ratios of 0.12–0.14 and $\delta^{13}$C$_{org}$ of -23‰ – -24‰, close to the estuarine end of the continuum. Correspondingly, the computed contribution of phytoplankton-derived organic matter to sedimentary C$_{org}$ increases systematically along the transect. As a first-order estimate based on the assumed end-member values, >70% of sedimentary C$_{org}$ at Station A is derived from terrestrial plant material, whereas sedimentary C$_{org}$ at Station K is entirely phytoplankton-derived. Station C and D stand out from the offshore trend with slightly
elevated contributions of %OC$_{phyt.}$ relative to %OC$_{terr.}$.

## 4.5 Total sedimentary Fe along the transect

Surface-sediment total Fe concentrations in 2014 were highest in the estuary (Stations A–E, approximately 1000–1700 µmol g$^{-1}$), followed by the archipelago (Stations H–K, approximately 800–1000 µmol g$^{-1}$), and lowest at the sill (Stations F–G, approximately 500–700 µmol g$^{-1}$, Fig. 4b). A maximum value of >1600 µmol g$^{-1}$ was recorded in the surface sediments at Station B. The downcore profiles from Stations A, D and J from 2015 show that total sedimentary Fe has been consistently higher at the stations in the estuary (A and D) relative to the archipelago (J) throughout the studied interval (in each case at least 50 years based on the position of the 1970 depth horizon). Extreme enrichments of Fe are observed at Station A, the site closest to the river mouth, peaking in the interval above the 1970 depth horizon and declining towards the present. Iron enrichments in the uppermost part of the estuary also appear to be highly spatially variable, as evidenced by the difference between the surface sediment value at Station A during the 2014 campaign (1043 µmol g$^{-1}$, Fig. 4b) and the 2015 campaign (1557 µmol g$^{-1}$, Fig. 5), measured in cores taken within 10 m of each other.

## 4.6 Sedimentary Fe speciation 1: labile Fe

Multiple phases of Fe are present in the sediments along the transect. Based on a combination of evidence from the extraction protocols and Mössbauer spectroscopy, we propose a general scheme for the interpretation of the combined data (Table 3). Flocculated Fe is assumed to accumulate as labile Fe in the form of ferrihydrite and Fe (III)-OM, both of which are subject to diagenetic transformations after sedimentation.

Labile Fe phases (Stage 1- and Stage 2- soluble Fe in the Poulton and Canfield (2005) extraction) are present in the surface sediments at all sites (Fig. 4b), and at all depths in the downcore profiles from Stations A, D, and J (Fig. 5). The near 1:1 match between the combined Stage 1+2 fraction and total 1 M HCl-soluble Fe from the Burton et al. (2008; 2011) extraction suggests that the two approaches are consistent in extracting 100% of the labile Fe pool (Fig. 6b). However, Na acetate- (Stage 1)- soluble Fe dominates over hydroxylamine-HCl- (Stage 2)- soluble-Fe in all samples (Fig, 4b, 5). Candidate phases for Na acetate- soluble Fe are Fe (II) carbonates such as siderite and ankerite, as targeted by the extraction protocol, but also labile organic complexes, whose behavior in this extraction scheme is not well defined, and AVS (FeS), which is thought to partially dissolve during Stage 1 (Egger et al., 2015a). Meanwhile, candidate phases for hydroxylamine-HCl-soluble Fe are poorly crystalline Fe oxides such as ferrihydrite and lepidocrocite, any remaining labile organic complexes and FeS from Stage 1, and Fe (II) phosphates such as vivianite (Dijkstra et al., 2014) (Table 3).

By determining the fractions of 1M HCl-soluble sulfidized and unsulfidized Fe (II), and Fe (III), in the parallel extraction, we can further deconvolve the likely composition of the labile Fe pool. The persistent presence of unsulfidized Fe (II), accounting for an average of 53% of labile Fe in the 15 measured samples (Fig. 6a), suggests important contributions of either Fe (II)-OM, Fe (II) carbonates, and/or Fe (II) phosphates. We suggest that Fe (II)-OM contributes significantly to this pool of unsulfidized Fe (II), for two principal reasons. First, Fe (II)-OM has been observed as an important component of total Fe in

sediments from a nearby boreal estuary on the basis of X-Ray Absorption Spectroscopy (Yu et al., 2015). Second, Fe (II) carbonates and phosphates if present are expected to form only in the deeper part of the sediment column, where pore water Fe, dissolved inorganic carbon and phosphate concentrations are sufficient to exceed saturation with respect to these minerals (e.g., Egger et al., 2015b for vivianite). In contrast, we find unsulfidized Fe (II) throughout the sediment column at all three
stations (Fig. 6a), including within the sulfate-methane transition zone (SMTZ), where the presence of pore water $H_2S$ keeps pore water Fe concentrations close to zero and hence should prevent carbonate and phosphate formation.

The 1M HCl extractions also confirm the presence of labile Fe (III) in most samples (Fig. 6a), including at depths corresponding to the SMTZ. Although ferrihydrite is almost certain to be present at the sediment surface, its persistence in the SMTZ, where pore water $H_2S$ concentrations exceed 100 µmol $L^{-1}$ (Station D) and 200 µmol $L^{-1}$ (Station J), seems improbable.
The half-life of the sulfidization of ferrihydrite is measured in hours (Raiswell and Canfield, 2012), whereas the residence time of a sediment layer in the $H_2S$ zone during burial at the calculated sedimentation rates in this study is in the order of years. Hence we suggest that the labile Fe (III) present in the sediments is indicative of $H_2S$-resistant Fe (III)-OM derived from the flocculated material.

The Mössbauer data yield further information concerning the labile Fe pool. Although the bulk spectra are dominated by the
major refractory Fe phases (Section 4.7), the presence or absence of minor Fe components can be inferred, especially when these are characterized by distinct hyperfine parameters and hence do not display significant spectral interference with the major components. On this basis, we find no direct evidence for the presence of siderite or other Fe (II) carbonates, whose Mössbauer spectra are all characterized by a distinct doublet with narrower quadrupole splitting than observed for Fe (II) in silicate minerals (Fig. 7 and Morris et al., 2010). Furthermore, the bulk spectra do not show evidence for either Fe (II) oxalate
or vivianite – two possible analogs for Fe (II)-OM and Fe (II) phosphates in the sediments (Fig. 7). To reconcile this information with the results of the extractions, we interpret the absence of these phases to indicate that Fe (II)-OM and Fe (II) phosphates in our samples must be present in forms other than Fe (II) oxalate and vivianite, and which are characterized by Mössbauer doublet spectra that interfere with silicate Fe (II) and hence cannot be independently resolved. Indeed, Mattievich and Danon (1977) showed that Fe (II) phosphates of varying degrees of hydration display a wide range of hyperfine parameters
including various overlaps with silicate Fe (II) as detected in our samples ($\delta$ = 1.08–1.19 mms$^{-1}$, $\Delta EQ$ = 2.48–2.65 mms$^{-1}$, Table 4, Section 4.7). The range of hyperfine parameters for different Fe (II)-OM phases is not well established in the existing literature, but our results suggest that Fe (II) oxalate alone may not be a suitable analog for bulk sedimentary Fe (II)-OM.

### 4.7 Sedimentary Fe speciation 2: refractory Fe

Refractory Fe compounds (by our definition, all Fe remaining after Stages 1+2 of the Poulton and Canfield (2005) extraction)
constitute the majority of Fe in all samples (Fig. 4b, 5). These components are expected to derive principally from suspended minerogenic matter accumulated in the estuarine sediments, although some refractory phases – such as pyrite – may derive

from the diagenesis of flocculated labile Fe (Table 3). Mössbauer spectra from five of the 7 analyzed samples (all samples except those from Station A) can be deconvolved into two doublets accounting for >95% of total Fe (Fig. 7 and Supplementary Fig. 2). The first represents superparamagnetic Fe (III) ($\delta$ = 0.25–0.40 mms$^{-1}$, $\Delta$EQ = 0.70–0.86 mms$^{-1}$, Table 4), and accounts for 49–62% of the modeled spectra in these samples (Table 4). Superparamagnetic Fe (III) in RT $^{57}$Fe Mössbauer analysis of

sediments has been reported to represent various phases, including ferrihydrite, goethite of particle size < approximately 12 nm (van der Zee et al., 2003) and amorphous ferric aluminosilicates (Manning and Ash, 1978; Manning et al., 1980). As outlined in Section 4.6, a major contribution of ferrihydrite deeper in the sediments seems unlikely, which leaves nanocrystalline goethite and amorphous ferric silicates as prime candidates for the dominant superparamagnetic Fe (III) phases. According to Canfield (1989), goethite is soluble in citrate-dithionite solution (Stage 3 of the Poulton and Canfield (2005)

extraction), while amorphous ferric silicates, if present, should be extracted by either citrate-dithionite or ammonium oxalate (Stage 4). However, the contribution of 49–62% superparamagnetic Fe (III) exceeds the sum of Stage 3+4 -soluble Fe in the corresponding extraction (typically 20–30%, Fig. 5). Several explanations are possible for this offset. First, the spectral overlap with diamagnetic pyrite may elevate the estimated contribution of superparamagnetic Fe (III) (Fig 7). Second, assuming Fe (III) oxalate as a reasonable analog for Fe (III)-OM (Barber et al., 2017), this component of labile Fe may also contribute a

fraction of total superparamagnetic Fe (III) (Fig. 7). Finally, it is possible that a fraction of goethite and/or amorphous ferric aluminosilicates in our samples may resist dissolution in Stages 3+4 and instead dissolve in Stage 5 (boiling 12 M HCl), thereby giving an underestimate for Stage 3+4 -soluble Fe.

The second major doublet in these five samples is high-spin Fe (II) in silicate minerals, identified by its comparatively high isomer shift and quadrupole splitting values ($\delta$ = 1.08–1.19 mms$^{-1}$, $\Delta$EQ = 2.48–2.65 mms$^{-1}$, Table 4). Iron (II)

(alumino)silicates in Baltic Sea sediments include micas such as biotite, as well as secondary clay minerals such as chlorite and illite (Frančišković-Bilinski et al., 2003). Due to strong overlaps in the Mössbauer parameters of these phases, it is difficult to deconvolve high-spin Fe (II) further, and for this reason we report these phases together simply as "silicate Fe (II)". This component typically accounts for 38–49% of the bulk spectrum in these five samples (Table 4), and may include contributions of overlapping labile Fe (II) phases as discussed in Section 4.6.

At Station A, total Fe contents are strongly elevated and the majority of the additional Fe is soluble in Stages 3+4 of the Poulton and Canfield (2005) extraction (Fig. 5). According to the interpretations already given, these fractions should include nanocrystalline goethite and amorphous ferric aluminosilicates, both of which would yield superparamagnetic Fe (III) in the Mössbauer spectra. However, both Mössbauer spectra from Station A (0–1 cm and 26–28 cm) also show evidence for two additional Fe phases, which together account for >50% of total Fe in these samples (Fig. 7) and therefore likely dominate Stage

3+4-soluble Fe at this site. Of these, the major phase is represented by a doublet located between those of superparamagnetic Fe (III) and silicate Fe (II), with isomer shift and quadrupole splitting parameters indicative of an Fe (II) compound ($\delta$ = 0.78–0.94 mms$^{-1}$, $\Delta$EQ = 0.76–0.83 mms$^{-1}$, Table 4). In the associated Discussion paper we attributed this phase to Fe (II)-OM.

However we now favor the hypothesis that this is in fact wüstite (FeO) (Fig. 7). Wüstite is an Fe (II) oxide with a distinct set of hyperfine parameters corresponding to a doublet between those of superparamagnetic Fe (III) and silicate Fe (II), which closely matches the dominant additional phase present in our samples (Fig. 7). The second additional phase in Station A samples is unequivocally magnetite, as evidenced by the broad sextet spectrum indicative of magnetic splitting of the absorption peaks (Supplementary Fig. 2) and thus non-zero internal magnetic field parameters at room temperature (Table 4). The magnetite spectrum can be further deconvolved into $Fe^{2.5+}$ and $Fe^{3+}$ components, each characterized by a distinct set of hyperfine parameters. Hematite is also observed in the sample from 26–28 cm.

### 4.8 Diagenetic zonation of the sediments

Active remineralization of organic matter (OM) at Stations A, D and J is evidenced by declining $C_{org}$ contents from the surface downwards at each site (Fig. 5). Since oxygen penetration in muddy coastal sediments of the northern Baltic Sea typically does not exceed 4 mm (e.g. Hietanen and Kuparinen, 2008; Bonaglia et al., 2013), anaerobic processes initiate within the uppermost centimeter and dominate the pore water chemistry of the upper 50 cm of the sediment column (Sawicka and Brüchert, 2017). These processes are expected to influence the vertical distribution of Fe in the sediments, with a strong impact on flocculated labile Fe phases.

The vertical zonation of the pore water chemical profiles at Stations A, D, and J is broadly similar, indicating a common set of anaerobic diagenetic processes at each site (Fig. 8). The following formulations are simplified from the reaction network of Reed et al. (2011), in which OM is given the form $\mathbf{(CH_2O)}_a\,\mathbf{(NH_4^+)}_b\,\mathbf{(H_3PO_4)}_c$. $Fe(OH)_3$ represents all Fe oxides, including ferrihydrite derived from flocculation. In the upper 5 cm of the sediment column, we observe evidence for anaerobic OM remineralization coupled to reduction of both Fe oxides (indicated by pore water accumulation of $Fe^{2+}$) and sulfate ($SO_4^{2-}$) (indicated by a concave decline of pore water $SO_4^{2-}$ with depth) (Fig. 8):

$$OM_{(s)} + \mathbf{4}aFe(OH)_{3(s)} + \mathbf{12}aH_{(aq)}^+ \rightarrow aCO_{2(aq)} + bNH_{4(aq)}^+ + cH_3PO_{4(aq)} + \mathbf{4}aFe_{(aq)}^{2+} + \mathbf{13}aH_2O_{(l)} \tag{12}$$

$$OM_{(s)} + \mathbf{0.5}aSO_{4(aq)}^{2-} + aH_{(aq)}^+ \rightarrow aCO_{2(aq)} + bNH_{4(aq)}^+ + cH_3PO_{4(aq)} + \mathbf{0.5}aH_2S_{(aq)} + aH_2O_{(l)} \tag{13}$$

Below the depth at which sulfate is exhausted, methane ($CH_4$) accumulates in pore waters, indicative of anaerobic OM remineralization via methanogenesis:

$$OM_{(s)} \rightarrow \mathbf{0.5}aCO_{2(aq)} + bNH_{4(aq)}^+ + cH_3PO_{4(aq)} + \mathbf{0.5}aCH_{4\,(aq)} \tag{14}$$

These observations are broadly consistent with the classic zonation of primary diagenetic reactions (Claypool and Kaplan, 1974; Froelich et al., 1979), except that the diagenetic zones strongly overlap, which is typical for coastal areas of the Baltic Sea (Sawicka and Brüchert, 2017). The presence of sulfate reduction (reaction 13) in sediments containing Fe oxides therefore

leads to the following reaction, which likely also contributes to the accumulation of $Fe^{2+}$ in the pore waters of the upper sediments:

$$2Fe(OH)_{3(s)} + H_2S_{(aq)} + 4CO_{2(aq)} \rightarrow 2Fe^{2+}_{(aq)} + S_{0\ (s)} + 4HCO^-_{3\ (aq)} + 2H_2O_{(l)} \tag{15}$$

Hydrogen sulfide ($H_2S$) and $Fe^{2+}$ then react to form iron monosulfide (FeS):

$$Fe^{2+}_{(aq)} + H_2S_{(aq)} \rightarrow FeS_{(s)} + 2H^+_{(aq)} \tag{16}$$

Iron monosulfide (FeS) is subsequently converted to pyrite ($FeS_2$) by reaction with native sulfur produced in reaction 15:

$$FeS_{(s)} + S_{0(s)} \rightarrow FeS_{2(s)} \tag{17}$$

A shallow sulfate-methane transition zone (SMTZ) is observed at each of the three stations, within which anaerobic oxidation of methane (AOM) coupled to sulfate reduction occurs (e.g., Egger et al., 2015a; Sawicka and Brüchert, 2017):

$$CH_{4(aq)} + SO^{2-}_{4(aq)} + CO_{2(aq)} \rightarrow H_2S_{(aq)} + 2HCO^-_{3\ (aq)} + H_2O_{(l)} \tag{18}$$

This reaction also produces $H_2S$, which accumulates in the pore waters at the SMTZ, hence keeping pore water $Fe^{2+}$ concentrations low and favoring the precipitation of sulfide minerals (reactions 16 and 17).

At all stations, pore water $Fe^{2+}$ increases below the SMTZ, indicative of excess production of $Fe^{2+}$ over $H_2S$ (Fig. 8). Such deep production of $Fe^{2+}$ may be associated with persistent organoclastic Fe reduction (reaction 12) or with AOM coupled to
15  Fe oxides (Beal et al., 2009; Sivan et al., 2011; Egger et al., 2015a):

$$CH_{4(aq)} + 8Fe(OH)_{3(s)} + 15H^+_{(aq)} \rightarrow HCO^-_{3\ (aq)} + 21H_2O_{(l)} + 8Fe^{2+}_{(aq)} \tag{19}$$

We note that indirect coupling of Fe oxide reduction to methane oxidation may also occur via the "cryptic sulfur cycle" (Holmkvist et al., 2011). Here, native sulfur and other intermediates of sulfide oxidation by Fe oxides (reaction 15), undergo disproportionation to $H_2S$ and $SO_4^{2-}$. The following example is given for thiosulfate ($S_2O_3^{2-}$):

$$S_2O^{2-}_{3(aq)} + H_2O_{(l)} \rightarrow SO^{2-}_{4(aq)} + H_2S_{(aq)} \tag{20}$$

The $H_2S$ produced in reaction 20 then feeds back into reaction 15, while the $SO_4^{2-}$ feeds back into reaction 18. However, as shown in Egger et al. (2016) in a study from the Black Sea, such cryptic sulfur cycling is restricted to the depth interval close to the downward-diffusing $H_2S$ front from the SMTZ. This can be explained by the fact that only 3 moles of $H_2S$ are produced by reaction 20, for every 4 moles of $H_2S$ that initially participate in reaction 15 (Egger et al., 2016). Hence, cryptic sulfur

cycling is a net consumer of H$_2$S and cannot persist significantly deeper than the downward-diffusing H$_2$S front from the SMTZ. For this reason we favor the interpretation that pore water Fe$^{2+}$ accumulating below the SMTZ is produced via reactions 12 and 19.

All stations show a broad maximum of solid-phase S in the post-1970 interval of the sediment column (Fig. 5). This feature is distinctive for sediments from the northern Baltic Sea (e.g., Egger et al., 2015a; Rooze et al., 2016) and represents a recent shoaling of the SMTZ to its current position, in response to eutrophication caused by external nutrient inputs to the Baltic (Gustafsson et al., 2012). The extra input of organic material to the sediments has led to enhanced rates of sulfate reduction (reaction 13), decreasing the sulfate penetration depth and intensifying reactions 15–17 at a shallower horizon in the sediments.

The above reaction network principally describes the diagenetic processes which may impact on flocculated ferrihydrite and its subsequent transformation to sulfide-bound Fe (II) phases. We note that under conditions of elevated pore water Fe (II) concentrations such as observed in the deeper sediments at Stations A and D, precipitation of Fe (II) phosphates may also occur (Egger et al., 2015b), although the Mössbauer data suggest that these phases, if present, are distinct from pure vivianite (Section 4.6). Furthermore, the effect of diagenetic reactions on Fe-OM complexes is not well understood and thus is typically not included in diagenetic models of Baltic Sea sediments. For the purposes of this study we assume that Fe (III)-OM derived from flocculation may be subject to reduction to Fe (II)-OM in the sediment column, as described in Yu et al. (2015), but that these components do not otherwise partake in the diagenetic reactions.

### 4.9 Instantaneous rates of diagenetic processes and their relative depth in the sediment column

The PROFILE output broadly confirms the above description of the diagenetic zonation. All sites show evidence for sulfate and methane consumption within the SMTZ. Instantaneous rates of sulfate reduction are in the range 0.0001–0.0008 µmol L$^-$$^1$ s$^{-1}$, which is similar to the range presented recently by Sawicka and Brüchert (2017) for a Swedish estuary. At Station A, low salinity restricts the sulfate reduction zone to the uppermost 5 cm of the sediments. Consequently, the SMTZ is shallower at Station A than at the other sites. However, instantaneous rates of sulfate reduction in the upper sediments of Station A are in fact the highest of the three sites. Net production of pore water Fe is observed in the upper 5 cm at all sites, as well as consumption in the SMTZ. Although Station A and D show strongly elevated concentrations of pore water Fe below the SMTZ, the PROFILE output indicates that production rates in this zone are low, suggesting an upwards diffusion of Fe from reactions occurring deeper in the sediments.

### 4.10 Long-term rates of Fe and sulfate reduction

Depth-integrated burial rates of S for the period 1970–2015 were highest at Station D, followed by Station A and lowest at Station J (Fig. 6c). By definition, the same pattern is observed in the AVS-Fe and pyrite-Fe fractions of the depth-integrated burial rate of reactive Fe (II). However, the additional component of unsulfidized Fe (II) at Stations A and D enhances the

contrast with the Station J, such that reactive Fe (II) burial at Station D is more than a factor 2 greater than at Station J (Fig. 6c).

## 5 Discussion

### 5.1 Evidence for flocculation of DOM

Up to 94% of total organic carbon (TOC) in Finnish river catchments is present as dissolved organic carbon (DOC) (Mattsson et al., 2005). Hence, the vast majority of the organic matter input to Finnish estuaries occurs in the dissolved form. Asmala et al. (2013) showed that DOC concentrations in the Mustionjoki estuary decrease from approx. 600 µmol L$^{-1}$ at the river mouth to 350 µmol L$^{-1}$ in the offshore region. The decline in DOC along the transect is controlled by mixing of river water with sea water of lower DOC content, and by transformations in the estuarine water column, which may cause non-conservative mixing

behavior of DOC vs. salinity. Yet, our compositional analysis shows that the DOM pool retains a strongly terrestrial character throughout the system (Fig. 3b). DOM data cluster around the terrestrial C3 plants field in N/C vs. $\delta^{13}$C space, with only a minor deviation towards higher N/C observed in the offshore samples.

The principal transformation leading to non-conservative loss of DOM in the Mustionjoki estuary is salinity-mediated flocculation – the aggregation of small particles of organic matter into larger ones under conditions of increasing electrolyte

strength (Asmala et al., 2014). Alternative transformations, such as microbial degradation and photolytic mineralization of DOM in the estuarine environment (Dalzell et al., 2009; Moran et al., 2000) are considered of lesser importance in this system. Flocculation decreases the fraction of organic matter in the water column which passes through regular filters, and therefore transfers terrestrial organic material from the DOM to POM pool.

At the time of sampling in June 2015, POM in surface waters throughout the Mustionjoki estuary and adjacent archipelago

was dominated by phytoplankton material, as evidenced by the relatively high N content of POM (circles with letters in Fig. 3a). This strong signal of autochthonous material apparently obscures any evidence for POM derived from flocculation of DOM. However, in deeper waters throughout the transect, POM is characterized by lower N contents and more isotopically enriched C. Consequently, deep water POM samples in N/C vs. $\delta^{13}$C space trend away from the riverine–estuarine phytoplankton continuum and towards the field corresponding to terrestrial C3 plants (Fig. 3b). This suggests that a second

fraction of POM contributes to the net N/C and $\delta^{13}$C values of the deeper samples, and that this fraction may be derived from flocculation of DOM to POM. The dominance of flocculation-derived material in the deeper samples may be due to both the loss of fresh phytoplankton material due to remineralization during settling, and the typically higher salinity of deep waters.

## 5.2 Flocculation of Fe and partial decoupling from DOM

Due to the close association of Fe with DOM in boreal riverine systems, transformations of DOM in the Mustionjoki estuary are expected to strongly influence Fe cycling. However, the degree of negative deviation from conservative mixing of $Fe_{diss}$ vs. salinity along the transect (Fig. 2b) is far greater than observed previously in this system for DOC (Asmala et al., 2014), implying some decoupling of Fe from DOM during flocculation. This is consistent with earlier results for the Öre estuary, Sweden (Forsgren, et al., 1996) which showed that the removal of Fe from solution during estuarine mixing experiments is far more efficient than that of DOC.

In the Forsgren et al. (1996) study, enhanced removal of Fe from solution was suggested to be caused by the presence of clay particle surfaces to which Fe oxyhydroxides can sorb during estuarine mixing. While this mechanism is indeed possible in our system, we note that partial decoupling of Fe from DOM likely occurs already in river water upstream of the estuary. The typical pH of the Mustionjoki drainage system is in the order 6.0–6.5 (Lahermo et al., 1996). As shown by Neubauer et al. (2013), a significant fraction of Fe in boreal river waters of pH 6.0–6.5 may exist in the form of discrete ferrihydrite particles, rather than OM-Fe complexes. Moreover, the modal size class of these particles increases with pH, such that a further pH rise in the estuarine environment should stimulate flocculation of Fe, independent of salinity effects or the presence of clay particles. The pH gradient of the estuary spans from 6.0–6.5 in the Mustionjoki river to 8.0–8.4 in the open Gulf of Finland (Omstedt et al., 2010). Hence, the strongly non-conservative behavior of $Fe_{diss}$ along the transect may represent the combined influence of salinity (influencing flocculation of DOM-bound Fe) and pH gradients (influencing the flocculation of discrete ferrihydrite particles) as well as the role of clay particles in the subsequent aggregation of particulate material.

Particulate Fe ($Fe_{part.}$) also shows an overall negative deviation from conservative mixing along the salinity gradient. The initial strong decline in $Fe_{part.}$ away from the river mouth likely reflects settling of suspended minerogenic matter due to energy dissipation in the estuarine environment (Syvitski and Murray, 1981). This is supported by a similar trend in suspended particulate aluminum (Al) (not shown). However, samples in the salinity range 2–4 show a clear positive deviation, suggesting an input of $Fe_{part}$ through flocculation of $Fe_{diss}$. Accordingly, the 2D cross-section of $Fe_{part}$ along the transect confirms a pronounced zone of higher values close to the halocline of the inner estuary (Fig. 2a). This material has no corresponding Al enrichment, confirming that it is not derived from suspended minerogenic matter and supporting the flocculation hypothesis. Further offshore, $Fe_{part}$ concentrations decline to a background value of ~0.5 μmol $L^{-1}$, implying partial settling of the flocculated material to the sediments.

## 5.3 The impact of flocculation on sedimentary OM and Fe

In the following, we assume that the input of flocculated Fe to sediments occurs in the form of ferrihydrite and Fe (III)-OM, in accordance with the observations of Yu et al. (2015) for a nearby estuary. These fractions in sediments are expected to be observed in the labile Fe pool (i.e. soluble in Stages 1+2 of the Poulton and Canfield (2005) extraction, or in 1M HCl). The

same is true for the diagenetic products of flocculated Fe, with the exception of pyrite, which classes as a refractory phase in our scheme (Table 3).

The transfer of terrestrially-derived OM and Fe from the dissolved to the particulate phase promotes the accumulation of both components in the sediments. Yet, our data suggests a further decoupling of terrestrial OM and Fe during sedimentation. A clear trend of decreasing terrestrial OM content in sediments is observed along the transect, consistent with enhanced input of flocculated material in the inner estuary (Fig. 4b). In contrast, the distribution of labile Fe in sediments is less clearly controlled by distance offshore, at least within the range of the Mustionjoki transect (Stages 1+2-soluble Fe, Fig. 4b). Rather, elevated labile Fe contents are seen at Station D, the deepest site in the inner estuary, while anomalously low values are observed at Station F, close to the sill at Ekenäs. This suggests that the bathymetry of the estuary has a modulating effect on the distribution of flocculated Fe.

The modulating effect of bathymetry on labile Fe may be related to two factors. The first is redox shuttling, during which Fe is focused downslope in low-oxygen marine environments due to repeated cycles of reduction and oxidation under the influence of gravity (e.g., Raiswell and Anderson, 2005; Lenz et al., 2015). The second is physical reworking by currents, which favors the transport of fine-grained material such as labile Fe away from shallow areas (Virtasalo et al., 2005). In particular, the upstream flow of saline water across the sill (Fig. 1) may favor the transport of labile Fe away from Station F and towards Station D (Fig. 4). We note that although this mechanism also influences sedimentary organic material, it is the phytoplankton-derived component which appears to be most susceptible. Both the absolute concentration of phytoplankton-derived OM, and its contribution to total OM, are elevated at Station D with respect to its position on the transect (Fig. 4b). This implies that the flocculated Fe fraction most affected by reworking to deeper areas should be ferrihydrite, rather than OM-Fe (III), since the latter is expected to behave similarly to bulk terrestrial OM.

To test whether the impact of flocculation on sedimentary Fe contents could be distinguished above focusing effects over a longer offshore transect, we compared Fe/Al ratios of surface sediments from the Mustionjoki transect with those from the nearby Paimionjoki transect (Fig. 1) and values reported in literature for the open shelves of the Baltic Sea (note that stations A and B have been removed from this analysis for reasons described in Section 5.4). Fe/Al correlates positively with labile Fe at the Mustonjoki sites ($R^2$ = 0.45, n = 8), hence serves as a first-order proxy for the input of flocculated material. The bathymetry of the Paimionjoki transect is less variable, thus the influence of focusing on the distribution of labile Fe is expected to be reduced. Accordingly, Fe/Al along the Paimionjoki transect shows a steady decline from Fe/Al = 0.93 to Fe/Al = 0.73 over >80 km S from the river mouth (Fig. 9). Furthermore, typical Fe/Al values for oxic shelf areas of the northern Baltic Proper – >100 km offshore from likely riverine Fe sources – are only ~0.5–0.6 (e.g., Lenz et al., 2015). Data from the Mustionjoki transect, though variable due to focusing effects, show a similar mean value to stations at the corresponding position on the Paimionjoki transect. Taken together, these results suggest that there is indeed an offshore decline in the input of flocculated Fe to sediments in the Baltic Sea, whose impact can be observed over large distances (>100 km) from the

coastline. This conclusion is supported by our suspended $Fe_{part}$ data, which shows a background of ~0.5 µmol $L^{-1}$ $Fe_{part}$ in the water column of Station K in the open Gulf of Finland (Fig. 2). Hence, although flocculation itself likely occurs further upstream at the contact between fresh and brackish water masses, a fraction of the suspended $Fe_{part}$ apparently evades settling in the estuarine environment and is transported offshore before sedimentation.

Considering the boreal coastal zone more broadly, the spatial impact of flocculation on sedimentary Fe chemistry is likely determined by the steepness of the offshore salinity gradient and the magnitude of the riverine Fe input. The Baltic Sea is characterized by a nearly 2000 km-long N–S surface-water salinity gradient of ~2–15, from the Bothnian Bay to the Danish Straits (Leppäranta and Myrberg, 2009). In the lowest-salinity regions in the Bothnian Bay and Bothnian Sea, the potential zone of estuarine flocculation extends further offshore and the trapping of Fe in estuarine sediments may be expected to be

less efficient. At the same time, riverine Fe inputs in this region are higher than elsewhere in the Baltic Sea (Asmala et al., 2014), due to the significant release of Fe from peatlands in the catchment areas (Kortelainen et al., 2006) and the potential release of Fe from acid-sulfate soils (Boman et al., 2010). Due to this combination of factors, flocculated $Fe_{part.}$ is likely dispersed over a much larger area in the Bothnian Bay and Bothnian Sea with respect to the Baltic Proper, which could explain the consistently high sedimentary Fe contents in the offshore areas of the Bothnian Sea (Slomp et al., 2013). At the other end

of the spectrum, in boreal estuaries draining into fully marine systems such as in the Arctic (e.g., Dai and Martin, 1995), steeper salinity gradients likely limit the dispersion of flocculated material to a relatively narrow zone close to the river mouths.

**5.4 Impact of industrial Fe pollution**

Extreme Fe enrichments of up to >25% Fe by weight are observed in the deeper sediments at Station A, principally due to elevated concentrations of Stages 3+4-soluble Fe (Fig. 5). The [57]Fe Mössbauer spectra from both the surface- and deep-

sediment samples from this station deviate significantly from all other measured samples, showing the presence of wüstite and magnetite alongside the typical doublets of superparamagnetic Fe (III) and silicate Fe (II) (Fig. 7). Such high concentrations of these minerals are rarely observed in natural aquatic sediments and suggest a significant impact of industrial pollution at this site.

The lack of wüstite and magnetite in the river bed sample at Station 'a' imply that the most likely source of these minerals in

Station A sediments is the blast furnace at Åminnefors (Fig. 1a), which is located <1 km from the river mouth and was active from the late 19[th] century until 1977. Both minerals may form as oxidation films on the surface of metallic Fe (Kim et al., 2000), which likely occurred continuously at the blast furnace immediately following the smelting process. Indeed, inputs of wüstite to sediments in Hamilton Harbor, Ontario, were interpreted to derive from precisely such a source (Manning et al., 1980). The magnitude of the enrichments of these minerals at Station A suggest that waste material may have been dumped or

eroded directly into the estuary.

The timing of maximum input of industrial Fe pollution shortly after 1970, and decline thereafter, is consistent with the timeline of activity at the blast furnace. However, the signal of wüstite and magnetite in the surface sediments at Station A, and considering the extraction results, likely also Station B (Fig. 4b), suggests a legacy effect in which this material is both cycled upwards in the sediment column by bioturbation, and gradually spread downstream by bottom transport. At present, there is no evidence for wüstite or magnetite at Station D, implying that the signal is currently restricted to the uppermost kilometers of the estuary. The presence of industrial Fe pollution a Stations A and B complicates the interpretation of offshore trends in Fe/Al, and therefore these stations were removed from Figure 9.

## 5.5 Impact of flocculation and sediment focusing on sedimentary diagenesis

The diagenetic zonation at Stations A, D and J is broadly similar (Fig. 8), suggesting that the pollution-derived material at Station A does not significantly affect the major diagenetic reactions in the sediments. The principal difference that can be discerned from the pore water profiles is that the SMTZ at Station A is shallower in the sediment column and is not defined by a pore water $H_2S$ enrichment such as observed at the other sites. The shallowness of the SMTZ at Station A is related to the low bottom-water sulfate concentration in the upper estuary, which limits the depth to which sulfate can penetrate before it is consumed. This itself may be the cause of the lack of $H_2S$ accumulation, due to the restricted zone in which sulfate reduction occurs and hence the total consumption of produced $H_2S$ by $Fe^{2+}$ (reaction 16). However, it is notable that both instantaneous sulfate reduction rates (Fig. 8), and net burial of S during the period 1970–2015 (Fig. 6c) are actually higher at Station A than at Station J, the furthest offshore of the three sites. Hence, bottom-water sulfate is apparently not limiting the net rate of sulfate reduction and subsequent formation of AVS and pyrite in this system. This result would appear to be consistent with the modeling simulations of Rooze et al. (2016), which showed that variations in bottom water sulfate in northern Baltic Sea sediments should not strongly impact on the solid-phase S profiles (Figure 6d in their paper), despite affecting other aspects of sedimentary diagenesis.

Net burial of S during the period 1970–2015 is highest at Station D, suggesting that high sedimentation rates and focusing of phytoplankton-derived OM in the deep basin of the inner estuary promotes higher long-term rates of sulfate reduction than observed at either Station A or Station J (Fig. 6c). Furthermore, the presence of unsulfidized Fe (II) in the sediments at both Stations A and D enhances net rates of reactive Fe (II) burial at these sites, over and above the contribution from AVS and pyrite (Fig. 6c). This implies that net rates of Fe reduction (including reduction of both ferrihydrite and Fe (III)-OM) are greater within the estuary than in the offshore region. Again, the highest rate is observed at Station D, consistent with the focusing of labile flocculated Fe to the deep inner basin (Fig. 4b).

The PROFILE output indicates that the high concentrations of pore water $Fe^{2+}$ in the deep sediments at Stations A and D are not produced *in situ* and hence likely represent Fe produced below the model domain (e.g. via reactions 12 and 19) which is currently diffusing vertically towards the SMTZ (Fig. 8). Therefore, it is possible that a fraction of the Fe (II) precipitated in

the sediment interval corresponding to 1970–2015 is in fact derived from earlier-deposited labile Fe (III) which is remobilized during deep burial, and therefore that the numbers for recent Fe (II) burial may be overestimated. Still, this observation reinforces our assertion that Fe reduction rates throughout the whole sediment column are higher at the sites in the estuary than further offshore, due to the more intense accumulation of flocculated labile Fe.

While the behavior of flocculated ferrihydrite during sedimentary diagenesis can be well described by the reaction network in Section 4.8, the details of the reductive transformation of Fe (III)-OM to Fe (II)-OM during burial remain unresolved by this study. We have assumed the model of Yu et al. (2015) to apply in our system, in which Fe (III) complexed to carboxylate and phenolate functional groups within humic material in the water column is reduced *in situ* in sediments, retaining its association with these functional groups during burial. However it is also possible that Fe (II) may associate with OM after release from

the reduction of ferrihydrite in the absence of $H_2S$. Moreover, the retention of a portion of Fe (III)-OM during burial, as implied by the 1M HCl extraction data (Fig. 6a) suggests either that not all of the flocculated Fe (III)-OM is reduced in sediments, or that new complexes of Fe (III) and relatively labile OM are formed during remineralization of OM in the sediment column and preserved during burial (Lalonde et al., 2012: Barber et al., 2017).

Interestingly, the higher rates of Fe and sulfate reduction observed at Station A with respect to Station J run contrary to the

assumption that the dominance of terrestrial plant-derived OM in the upper estuarine sediments should lower the net degradability of OM in the sediments (Arndt et al., 2013). Flocculated DOM is characterized by high contents of humic substances, derived from complex polymers in vascular plants, such as cellulose, lignin, cutin and cutan (de Leeuw and Largeau, 1993), which are relatively resistant to degradation by the sediment microbial community (e.g. Hedges et al., 2000). In contrast, phytoplankton-derived OM contains high concentrations of degradable compounds such as proteins, nucleic acids

and simple carbohydrates, and in the shallow coastal system may be expected to experience relatively little pre-ageing before deposition in the sediments. We suspect that the relatively high rates of diagenetic processes at Station A are related to high mass accumulation rates at this site. Although sedimentation rate is approximately similar to Station J (Fig. 5, 6), porosity of the sediments at Station A is significantly lower, yielding mass accumulation rates a factor 1.7 higher than Station J and therefore similarly increasing the accumulation rate of degradable organic material.

**5.6 Temporal stability of flocculation impacts on sedimentation**

The coastal zone of the Baltic Sea has been impacted severely by anthropogenic activities during the last century. First of all, enhanced nutrient inputs have altered coastal ecosystems and triggered hypoxia in many areas (Conley et al., 2011). Also land use changes such as ditching and forest clearance have influenced the inputs of particulate material to the coastal zone (Yu et al., 2015). The transport of riverine Fe and DOM into the Baltic has increased in recent decades as a consequence of

brownification (Kritzberg et al., 2012), related primarily to the recovery of boreal freshwater systems from industrial acidification and elevated ionic strength in the mid-20[th] century (Monteith et al., 2007). Finally, the coastal zone is impacted

by variable direct inputs of industrial Fe pollution such as those observed in this study. Each of these changes may be expected to influence both the sediment composition and diagenetic processes in the sediments at our study sites through time.

The most pronounced effect of recent coastal eutrophication on sediment chemistry has been to increase the flux of phytoplankton-derived organic matter to the sediments. As outlined in Section 4.8, this increased carbon loading has led to a vertical migration of the redox zones of the sediments. Recent shoaling of the SMTZ at Stations A, D and J is indicated by peaks in total sulfur (S) in the post-1970 sediments (Fig. 5), consistent with the time-dependent modeling simulations of Rooze at al. (2016), in which rates of AVS and pyrite precipitation in the SMTZ were shown to increase in response to carbon loading in the late 20th century.

Despite the various potential anthropogenic impacts, the role of flocculation in determining sedimentary Fe chemistry along the Mustionjoki transect appears to have remained largely unchanged throughout the studied interval. Labile Fe contents at Station D are quite constant throughout the core profile, implying that this site has served as a trap for flocculated material since at least the mid-20th century (Fig. 5). Similarly, labile Fe concentrations at Station D have been consistently lower during the same interval. Only at Station A is there evidence for slightly enhanced inputs in the past, coincident with the strong industrial pollution input in the 1970s.

## 6 Conclusions

In boreal estuaries, flocculation of DOM and Fe strongly influences the chemistry of the sediments and the diagenetic reactions involving Fe. We can draw the following main conclusions from the study:

- Flocculation of DOM and Fe are partially decoupled, likely due to the presence of discrete ferrihydrite colloids in the freshwater input which show a secular response to estuarine mixing from Fe associated with DOM
- Flocculation of Fe is reflected in strongly non-conservative mixing of $Fe_{diss.}$ along the estuarine salinity gradient. $Fe_{diss.}$ is preferentially removed from solution with increasing salinity, and potentially pH.
- The POM generated by flocculation of DOM can be detected in suspended particulate matter using $\delta^{13}C$ and N/C of DOM as end-member reference values. Due to the presence of phytoplankton material in surface waters at the time of sampling, flocculated OM in this study was primarily detected in deeper waters.
- The zone of Fe flocculation in the estuarine water column can be identified using parallel measurements of $Fe_{diss}$ and $Fe_{part}$. This occurs at low salinities, close to the primary contact between fresh and saline water masses. In the Mustionjoki estuary, this zone corresponds to the halocline of the stratified inner basin.
- Flocculation causes accumulation of labile Fe in the form of ferrihydrite and Fe (III)-OM in near-shore areas. The spatial scale of the flocculation signal in offshore sediment chemistry is likely dependent on the steepness of the salinity gradient, with greater dispersal in low-salinity systems. Redox shuttling and physical reworking modulate

the influence of flocculation on sedimentary Fe chemistry, by focusing flocculated labile Fe into bathymetric depressions.

- In contrast to Fe, the contribution of flocculated terrestrial OM to total sedimentary OM shows a clear offshore decline less impacted by focusing effects, as indicated by $\delta^{13}$C and N/C of sedimentary OM.
- During diagenesis, flocculated ferrihydrite and Fe (III)-OM are reduced in the sediment column, leading to the accumulation of AVS, pyrite and unsulfidized Fe (II), including Fe (II)-OM.
- Net rates of Fe reduction in sediments are higher in the inner estuary with respect to offshore areas, consistent with enhanced inputs of flocculated labile Fe.
- Industrial activities in coastal areas can lead to significant signals of Fe pollution in near-shore sediments.

## Data availability

All data from Figures 1–9 of the manuscript and Supplementary Figures 1 and 2 will be made available in Pangaea upon publication of the article.

## Supplement link

Supplementary Figures 1 and 2 are uploaded in parallel with the manuscript and are available from the Biogeosciences website.

## Author contributions

TJ devised the study, carried out field and laboratory work, interpreted the data, produced the figures and wrote the paper. EA interpreted the data and assisted with writing the paper. CS carried out the Mössbauer analysis, interpreted the data and assisted with writing the paper. RT carried out fieldwork and sequential Fe extractions, interpreted the data, produced figures and wrote the MSc thesis which formed the basis for the paper. J-PM carried out fieldwork, analyzed methane samples and assisted with writing the paper. JV, AK and PP provided sample material, interpreted the data and assisted with writing the paper. PE assisted with ICP-OES analysis. SH managed the project which principally funded the study, carried out fieldwork, interpreted the data and assisted with writing the paper.

## Competing interest statement

The authors declare no competing interest.

## Acknowledgements

We acknowledge support in field campaigns from Tvärminne Zoological Station, Göran Lundberg, Veijo Kinnunen, Ari Ruuskanen, Mikael Kraft, Petra Tallberg, Anni Jylhä-Vuorio and Kaarina Lukkari. Analytical and laboratory assistance was provided by Juhani Virkanen, Antti Nevalainen, Jaana Koistinen and Asko Simojoki. The comments of Peter Kraal and one anonymous reviewer substantially improved the study. This research was funded by Academy of Finland projects 139267, 272964 and 267112, Maa ja Vesitekniikan Tuki projects 31749 and 32861, and by the BONUS COCOA project (grant agreement 2112932-1), funded jointly by the EU and Danish Research Council. C.S. acknowledges a Travel and Instrumentation Award from the STFC-funded Geological Repositories Network and funding from the Carnegie Trust for the Universities of Scotland (Trust Reference No: 50357).

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

**Tables**

| Campaign and Station code | Co-ordinates (degree-decimal) °N | °E | Water depth (m, sediment Stations only) | Water sampling | Sediment sampling |
|---|---|---|---|---|---|
| *Mustionjoki September 2014; June 2015* | | | | | |
| A | 60.091617 | 23.554630 | 7 | Full profile (2015) | 4 samples (2014) Full profile (2015) |
| B | 60.079833 | 23.531167 | 11 | Full profile (2015) | 5 samples (2014) |
| C | 60.054300 | 23.509517 | 22 | Full profile (2015) | 5 samples (2014) |
| D | 60.022650 | 23.474600 | 39 | Full profile (2015) | 5 samples (2014) Full profile (2015) |
| E | 60.014033 | 23.467950 | 21 | Full profile (2015) | 5 samples (2014) |
| F | 59.994750 | 23.452300 | 8 | Full profile (2015) | 5 samples (2014) |
| G | 59.961383 | 23.396730 | 6 | Full profile (2015) | 5 samples (2014) |
| H | 59.920117 | 23.332650 | 17 | Full profile (2015) | 5 samples (2014) |
| I | 59.907367 | 23.326200 | 21 | Full profile (2015) | 5 samples (2014) |
| J | 59.855286 | 23.261780 | 33 | Full profile (2015) | 5 samples (2014) Full profile (2015) |
| K | 59.789867 | 23.335430 | 47 | Full profile (2015) | 5 samples (2014) |
| *Paimionjoki August-September 2001* | | | | | |
| L | 60.354667 | 22.563000 | 18 | | Surface (0–2 cm) |
| M | 60.313333 | 22.509833 | 46 | | Surface (0–2 cm) |
| N | 60.140970 | 22.410722 | 27 | | Surface (0–2 cm) |
| O | 60.057288 | 22.355018 | 46 | | Surface (0–2 cm) |
| P | 59.913438 | 21.753072 | 35 | | Surface (0–2 cm) |
| Q | 59.764387 | 21.706720 | 107 | | Surface (0–2 cm) |
| *Mustionjoki April-August-October 2011 (water); September 2015 (sediments)* | | | | | |
| a | 60.095467 | 23.590867 | | Surface only | Surface (0–2 cm) |
| b | 60.036650 | 23.484183 | | Surface only | |
| c | 59.977800 | 23.421300 | | Surface only | |
| d | 59.917300 | 23.324433 | | Surface only | |
| e | 59.855567 | 23.261967 | | Surface only | |
| f | 59.816317 | 23.271450 | | Surface only | |

**Table 1. Sampling campaigns and stations (see also Fig. 1).**

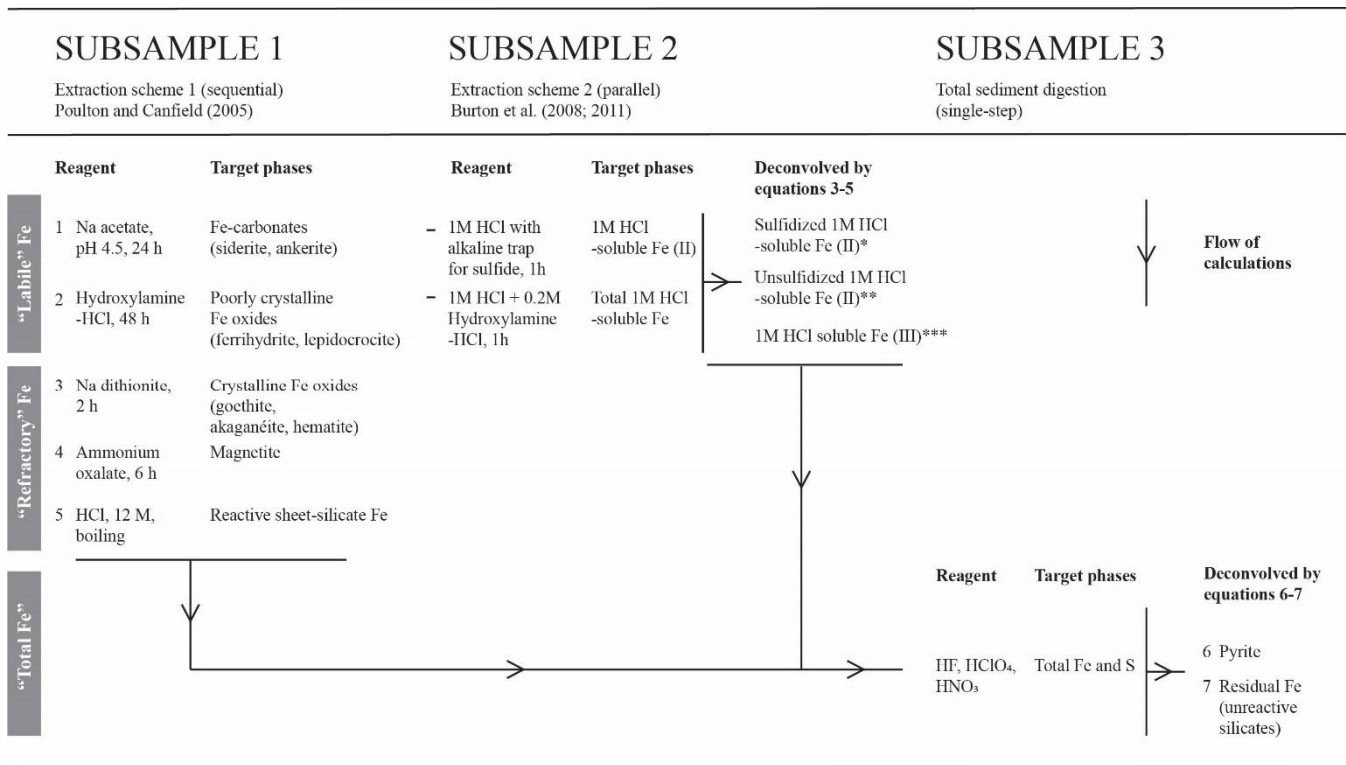

**Table 2. Protocol for sediment Fe extractions employed in this study. The results of the extractions are combined according to equations 3–7 (see text). *Sulfidized 1M HCl-soluble Fe (II) is assumed to be present as FeS (AVS). ** Unsulfidized 1M HCl-soluble Fe (II) may include carbonates (e.g., siderite, ankerite), phosphates (e.g., vivianite) and labile Fe (II)-OM. ***1M HCl-soluble Fe (III) includes poorly crystalline Fe oxides (ferrihydrite, lepidocrocite) and labile Fe (III)-OM.**

| | Solubility in Poulton and Canfield (2005) extraction | Solubility in 1M HCl extraction | Allocation in Mössbauer subspectra | Presence in samples of this study |
|---|---|---|---|---|
| **Labile Fe (II) phases** | | | | |
| Fe carbonates (siderite, ankerite) | Stage 1 | Yes | Not detected | Unlikely |
| Acid-Volatile Sulfur (FeS) | Stage 1-2 (inferred) | Yes | Not resolvable | Likely |
| Labile Fe (II)-OM | Stage 1-2 (inferred) | Yes | Not resolvable | Likely |
| Fe (II) phosphates | Stage 2 (inferred) | Yes | Not resolvable | Likely |
| **Labile Fe (III) phases** | | | | |
| Poorly crystalline Fe oxides (ferrihydrite, lepidocrocite) | Stage 2 | Yes | SP Fe (III) | Likely |
| Labile Fe (III)-OM | Stage 1-2 (inferred) | Yes | SP Fe (III) | Likely |
| **Refractory Fe (II) and Fe (III) phases** | | | | |
| Goethite (< 14 nm) | Stage 3 (-4 inferred) | No | SP Fe (III) | Likely |
| Goethite (> 14 nm) | Stage 3 (-4 inferred) | No | Not detected | Unlikely |
| Hematite | Stage 3 | No | Hematite | Likely |
| Akaganéite | Stage 3 | No | Not detected | Unlikely |
| Amorphous ferric aluminosilicates | Stage 3-4 (inferred) | No | SP Fe (III) | Likely |
| Wüstite | Stage 3 (inferred) | No | Wüstite | Likely |
| Magnetite | Stage 4 | No | Magnetite (Fe $^{2.5+}$) Magnetite (Fe $^{3+}$) | Likely |
| Reactive sheet silicate Fe (II) | Stage 5 | No | Silicate Fe (II) | Likely |
| Pyrite | No | No | Not resolvable | Likely |
| Residual unreactive silicate Fe (II) | No | No | Silicate Fe (II) | Likely |

**Table 3. Scheme for the interpretation of sedimentary Fe phases based on the extraction protocols and Mössbauer data. Solubilities in the Poulton and Canfield (2005) extraction listed as *inferred* indicate that these were not defined in the original protocol, but have either been investigated by later studies or are assumed for reasons described in the text. SP Fe (III) = superparamagnetic Fe (III).**

| Mineral Phase | $\delta^a$ (mm/s) | $\Delta_{EQ}^b$ (mm/s) | $B_{hf}^c$ (T) | Area$^d$ (%) | Fe(III)/Fe (II) |
|---|---|---|---|---|---|
| *River bed sediment (Station 'a')* | | | | | |
| SP $Fe^{3+}$ | 0.34 | 0.69 | - | 56 | 1.27 |
| Silicate $Fe^{2+}$ | 1.15 | 2.57 | - | 44 | |
| | | | | | |
| *Station A 0–1 cm* | | | | | |
| SP $Fe^{3+}$ | 0.32 | 0.67 | - | 26 | f |
| Silicate $Fe^{2+}$ | 1.08 | 2.61 | - | 18 | |
| Wüstite | 0.78 | 0.83 | - | 42 | |
| Magnetite $Fe^{2.5+}$ | [0.67]$^e$ | [0.00] | [46.5] | 9 | |
| Magnetite $Fe^{3+}$ | [0.30] | [0.03] | [49.8] | 5 | |
| | | | | | |
| *Station A 26–28 cm* | | | | | |
| SP $Fe^{3+}$ | 0.40 | 0.73 | - | 8 | f |
| Silicate $Fe^{2+}$ | 1.12 | 2.65 | - | 15 | |
| Wüstite | 0.94 | 0.76 | - | 42 | |
| Magnetite $Fe^{2.5+}$ | 0.66 | -0.01 | 45.5 | 16 | |
| Magnetite $Fe^{3+}$ | 0.26 | -0.01 | 48.6 | 9 | |
| Hematite | 0.35 | -0.08 | - | 10 | |
| | | | | | |
| *Station D 0–1 cm* | | | | | |
| SP $Fe^{3+}$ | 0.25 | 0.86 | - | 51 | 1.04 |
| Silicate $Fe^{2+}$ | 1.19 | 2.48 | - | 49 | |
| | | | | | |
| *Station D 26–28 cm* | | | | | |
| SP $Fe^{3+}$ | 0.36 | 0.73 | - | 62 | 1.63 |
| Silicate $Fe^{2+}$ | 1.13 | 2.58 | - | 38 | |
| | | | | | |
| *Station J 0–1 cm* | | | | | |
| SP $Fe^{3+}$ | 0.35 | 0.75 | - | 49 | 1.17 |
| Silicate $Fe^{2+}$ | 1.15 | 2.56 | - | 46 | |
| Unspecified Fe (III) oxide | 0.30 | -0.04 | 50.3 | 5 | |
| | | | | | |
| *Station J 30–32 cm* | | | | | |
| SP $Fe^{3+}$ | 0.34 | 0.69 | - | 53 | 1.13 |
| Silicate $Fe^{2+}$ | 1.14 | 2.57 | - | 47 | |

$^a$Isomer or center shift.

$^b$Quadrupole splitting. For 6-line spectra (magnetite, hematite and unassigned Fe oxide), values represent quadrupole shift $<\varepsilon>$ (same units)

$^c$Internal magnetic field.

<sup>d</sup>Subspectral area ratio, to first order proportional to relative amount of total Fe contained in mineral phase. A general uncertainty of ±2% absolute is applied.

<sup>e</sup>Values in square brackets were fixed to values from library spectra during the fitting process.

<sup>f</sup>Values for Fe (III):Fe(II) are only given for those samples unaffected by industrial pollution (see text).

5 **Table 4. Mössbauer parameters corresponding to spectra in Fig. 7 and Supplementary Fig. 2.**

**Figures**

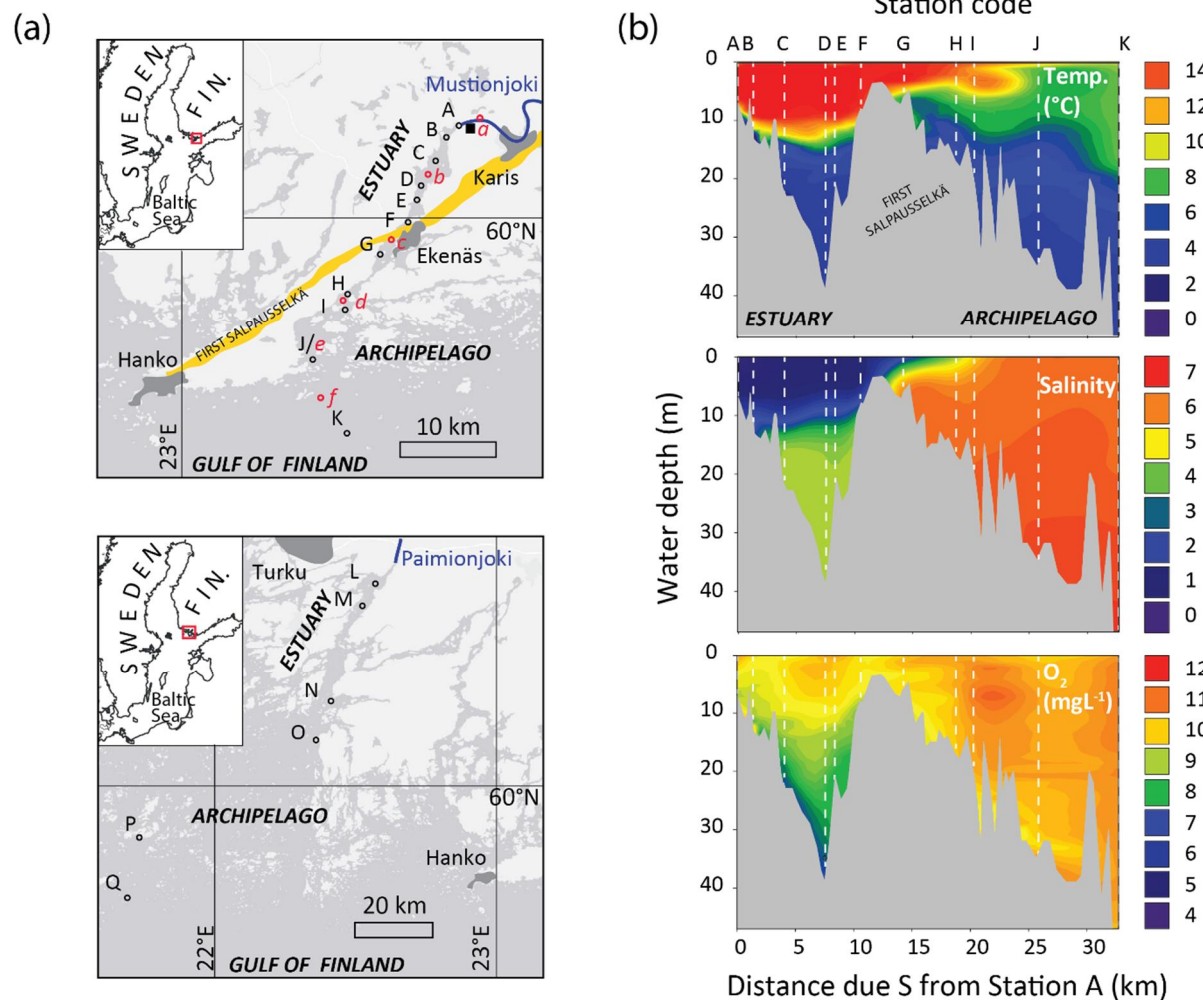

Figure 1. (a) Location of the Mustionjoki estuary transect (top) and the Paimionjoki estuary transect (bottom). In both systems, point-source river inputs discharge into a channel-like estuary, which in turn connects into the archipelago coastline of the Gulf of Finland, northern Baltic Sea. Sediment and water column sampling locations are indicated A–Q (of which L–Q were first reported in Virtasalo et al., 2005). Dissolved organic matter (DOM) sampling locations (Asmala et al., 2014, 2016) are indicated *a-f*. The location of the Åminnefors blast furnace is indicated by the black square. The First Salpausselkä ice-marginal formation is indicated in yellow. (b) Water column characteristics of the Mustionjoki transect during sampling in June 2015. 2D contour plots were generated by extrapolation between the

measured profiles at Stations A–K using Sigmaplot™ software. Distances along transect are reported as distance directly due S from Station A at the Mustionjoki river mouth.

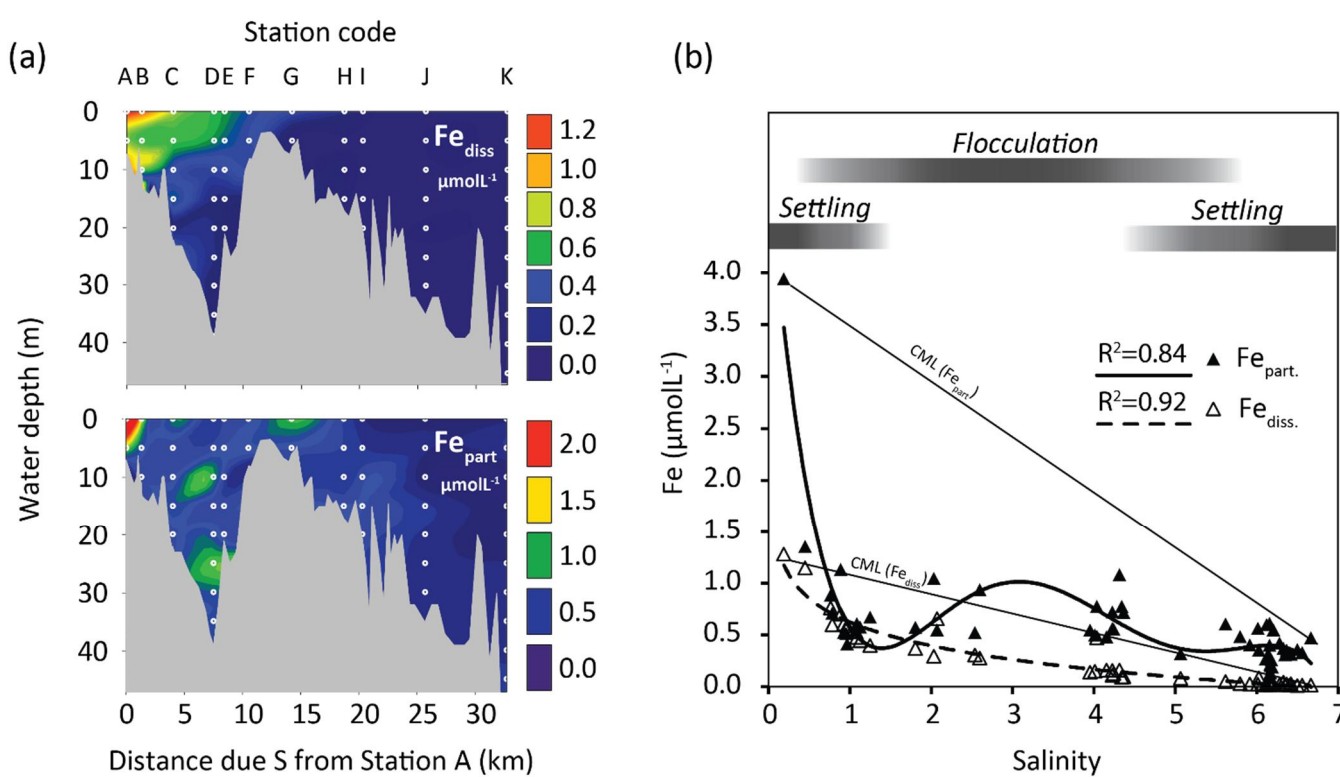

Figure 2. (a) 2D contour plots of dissolved (top) and particulate (bottom) Fe in the water column along the Mustionjoki transect (Stations A–K), operationally defined by filtration at 0.45 µm, June 2015. White circles represent sampling positions (vertical depth resolution = 5m). (b) Data from *a* plotted against salinity, including trendlines for $Fe_{part.}$ (polynomial) and $Fe_{diss.}$ (logarithmic). Linear Conservative Mixing Lines (CML) are drawn between the high- and low-salinity end-member samples for $Fe_{part}$ and $Fe_{diss}$. The inferred dominant processes controlling $Fe_{part}$ along the salinity transect are indicated by the grey bars.

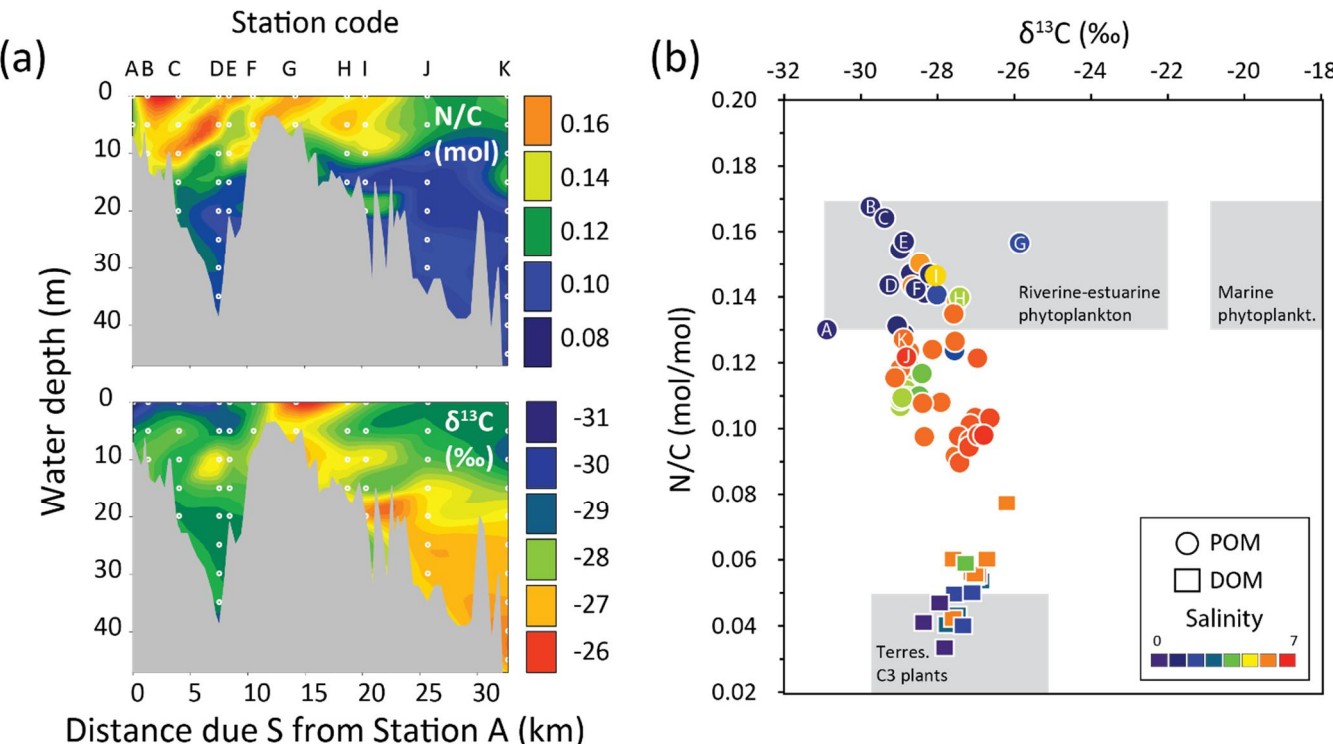

**Figure 3. (a) 2D contour plots of molar N/C (top) and δ¹³C (bottom) of particulate organic matter (POM) along the Mustionjoki transect, operationally defined by filtration at 0.45 μm, June 2015. White circles represent sampling positions (vertical depth resolution = 5m). (b) Cross plot of molar N/C vs. δ¹³C of POM in June 2015 (circles, each representing a single sample from the 2D plot in *a*) and of published data for dissolved organic matter (DOM) from the same transect (squares, surface water only, 6 samples each from campaigns in April, August and October 2011). In-situ salinity at the time and location of sampling is indicated by the color scale. Samples marked with letters indicate surface water samples. Organic matter source fields are taken from Goñi et al. (2003).**

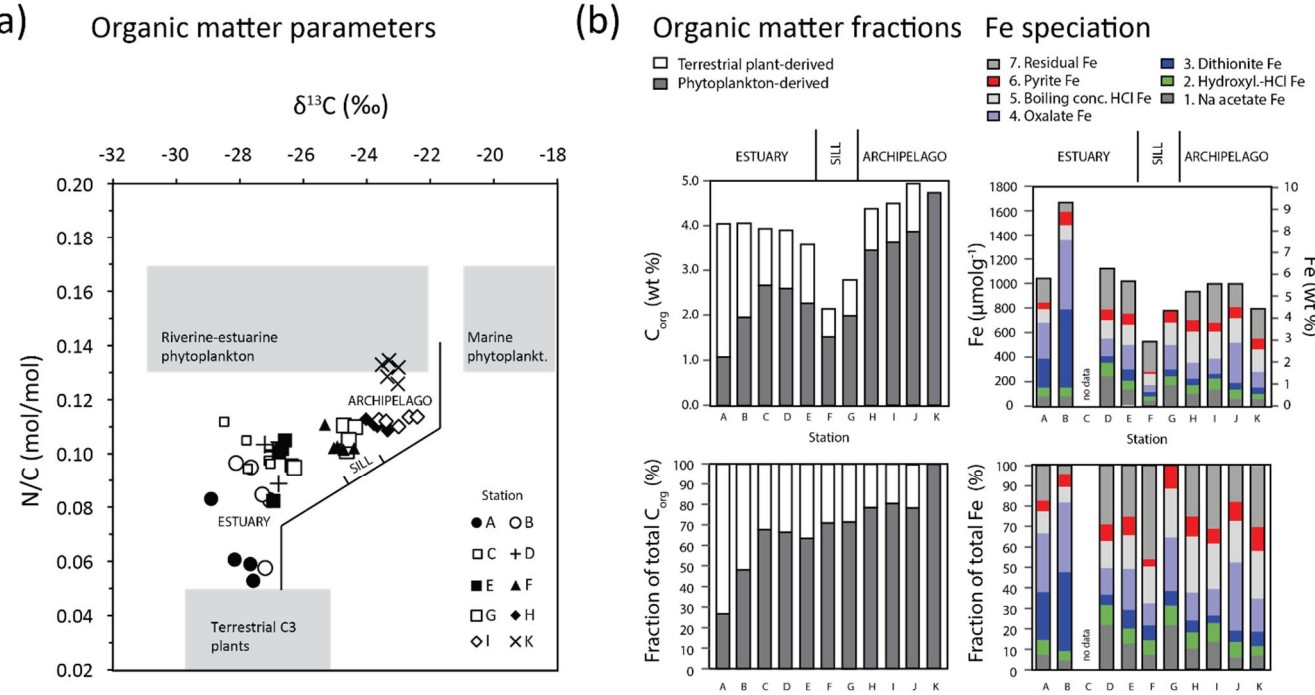

**Figure 4. (a)** Cross plot of molar N/C vs. $\delta^{13}C$ of sedimentary organic matter along the Mustionjoki transect. Stations are indicated by the symbols. No isotope data was available for Station J. Four or five samples are plotted for each Station, representing evenly spaced 2 cm thick slices throughout a GEMAX$^{TM}$ core of 30–60 cm length (e.g. Station K: 0–2 cm; 8.5–10.5 cm, 17–19 cm, 25.5–27.5 cm, 34–36 cm). **(b)** Organic matter fractions of the same sediment samples, derived from molar N/C ratios, assuming end-member values of N/C = 0.04 (terrestrial-derived) and 0.13 (phytoplankton-derived). Mean values are reported of the 4 or 5 samples from each core; Operational Fe speciation of surface-sediment samples (0–2 cm). No extraction data was available for Station C.

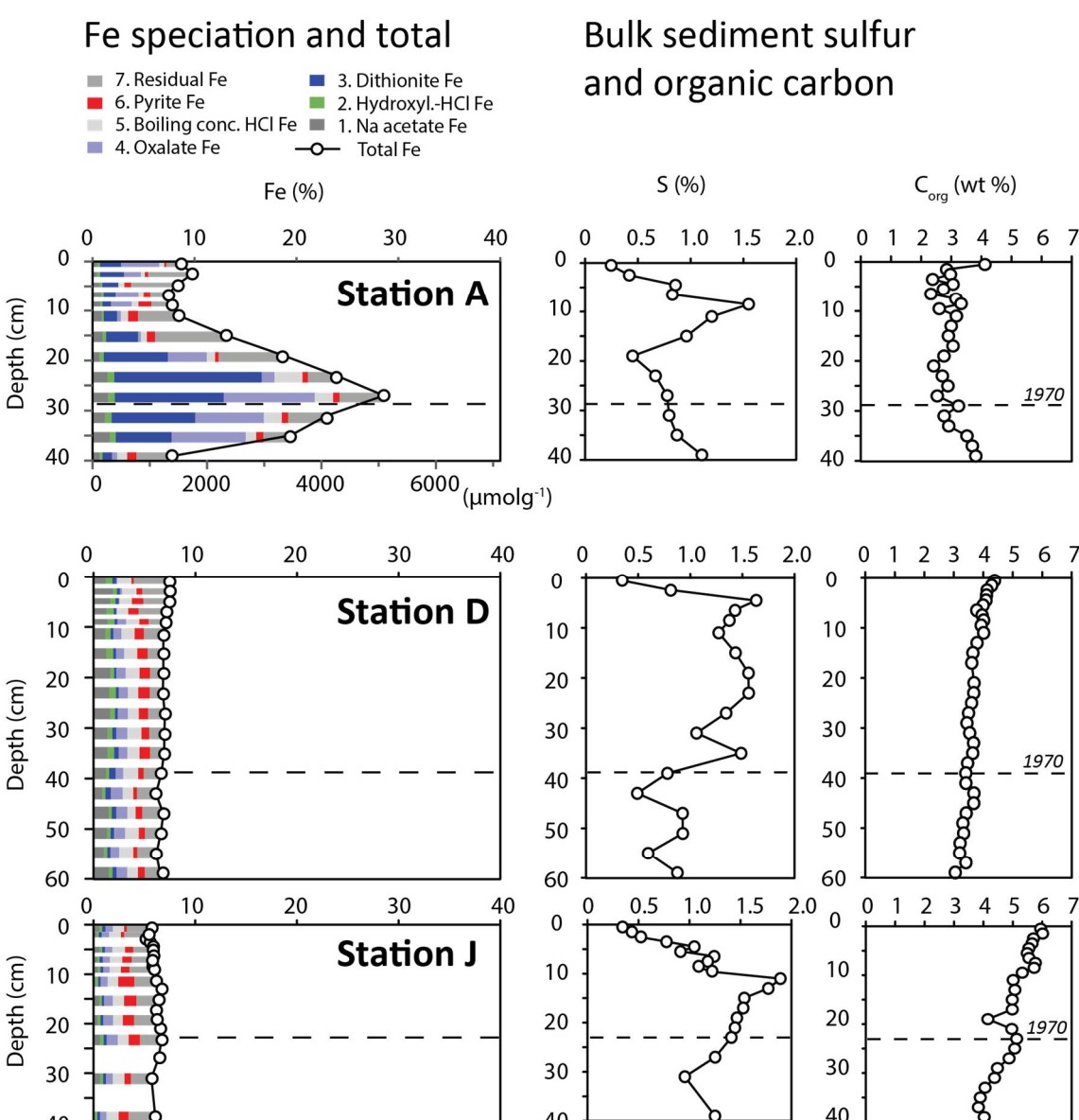

**Figure 5. (left)** Down-core operational Fe speciation and total sedimentary Fe content for Stations A, D, and J. Thickness of bars corresponds to thickness of sampled interval (i.e. 1 or 2 cm). Note that not all depth intervals were sampled. **(right)** Bulk chemical profiles from the same cores. The depth interval corresponding to 1970 is estimated from the peak in concentrations of total lead ($Pb_{tot}$) (see Supplementary Fig. 1).

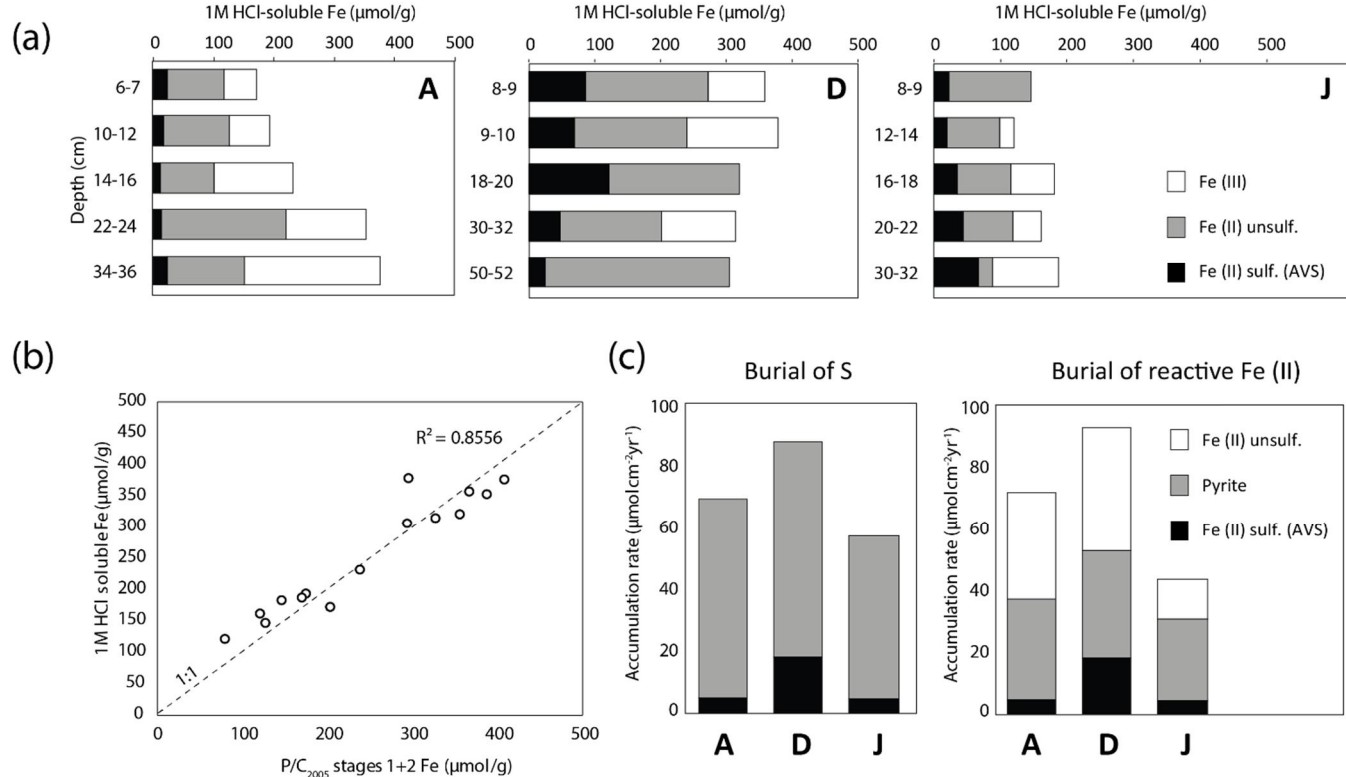

**Figure 6. (a) Results of the 1M HCl extraction, carried out on 15 samples (5 each from Stations A, D, J). (b) Comparison of total 1M HCl-soluble Fe and the sum of Stages 1+2-soluble Fe in the Poulton and Canfield (2005) protocol. Where no equivalent sample was available, adjacent samples have been compared (n =4). Dashed line represents 1:1 and the least-squares regression is performed against this line. (c) Depth-integrated rates of burial of S and reactive Fe (II) over the period 1970–2015 at Stations A, D and J.**

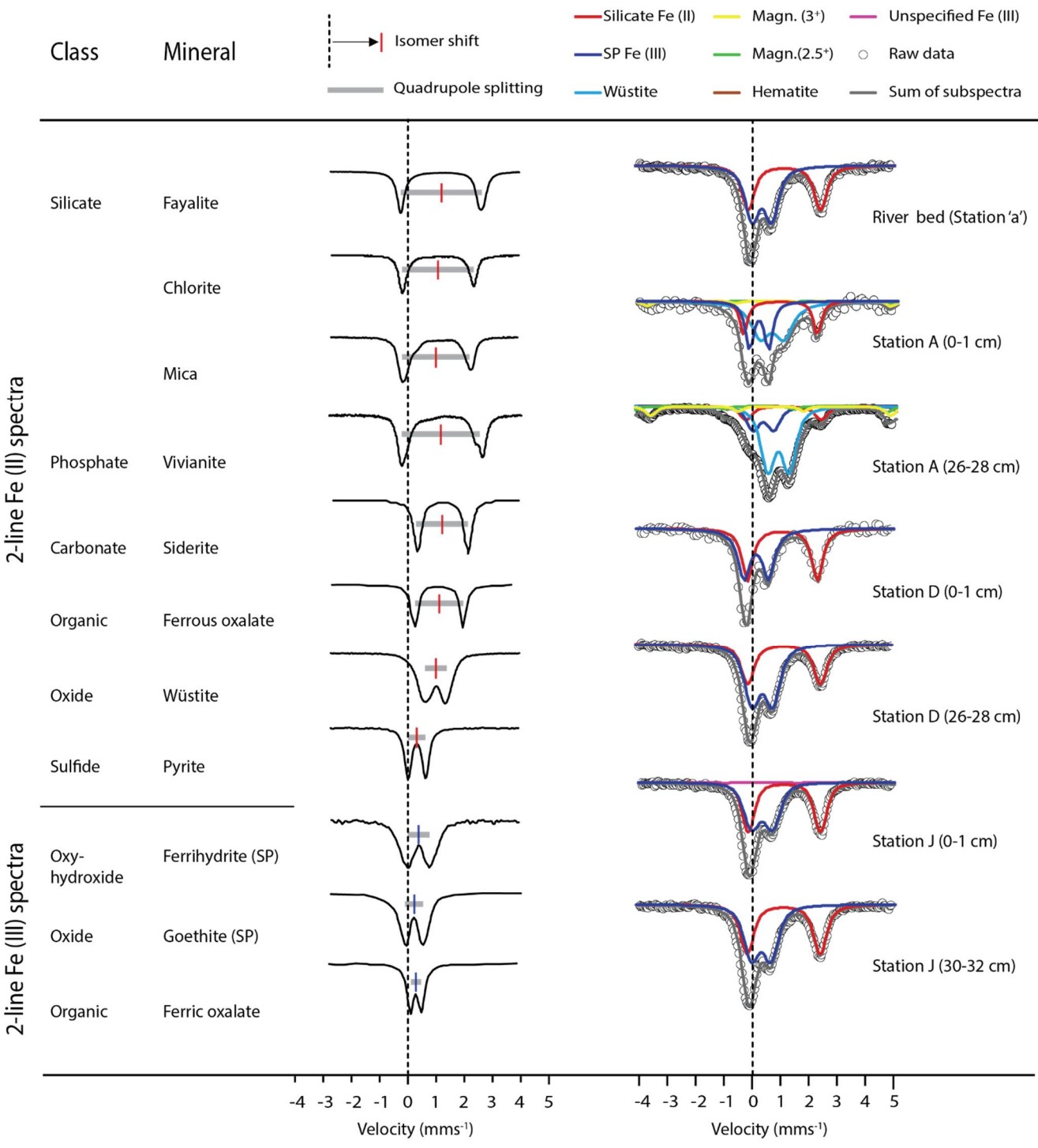

**Figure 7. (a) Room temperature (RT) 2-line Mössbauer $^{57}$Fe reference spectra from various sources. Spectra have been trimmed and normalized for visual intercomparison. The arbitrary y-axis indicates the intensity of gamma-ray transmission, hence troughs correspond to absorption maxima. Spectra for fayalite, chlorite, mica, vivianite, siderite, wüstite, pyrite and ferrihydrite are reproduced by permission of M. Darby Dyer from the Mineral Spectroscopy Database of Mount Holyoke College. These spectra are previously unpublished with the exception of fayalite (Belley et al., 2009) and vivianite (Dyar et al., 2014). A plot digitization software was used to extract Fe (II) and Fe (III) oxalate spectra from D'Antonio et al. (2009), and a superparamagnetic goethite (goethite SP) spectrum from van der Zee et al. (2003). The isomer shift and quadrupole splitting of each spectrum are indicated. Note that with the exception of pyrite, Fe (II) phases typically show a more positive isomer shift (position of center of doublet relative to zero velocity) and larger quadrupole splitting (distance between two peaks of a doublet on the velocity axis) than Fe (III) phases. (b) RT Mössbauer $^{57}$Fe spectra from this study. Spectra have been trimmed and normalized, and all spectra are presented in the same orientation for visual intercomparison. Spectra from Station A 0–1 cm and Station D 0–1 cm were collected in backscatter mode, hence the troughs correspond to backscatter emission maxima. All other spectra were collected in transmission mode. See Supplementary Fig. 2 for complete original spectra in true orientation, including 6-line standards. The subspectra used in the model fit for each bulk spectrum are shown in the legend. Relative spectral areas of each component are given in Table 4.**

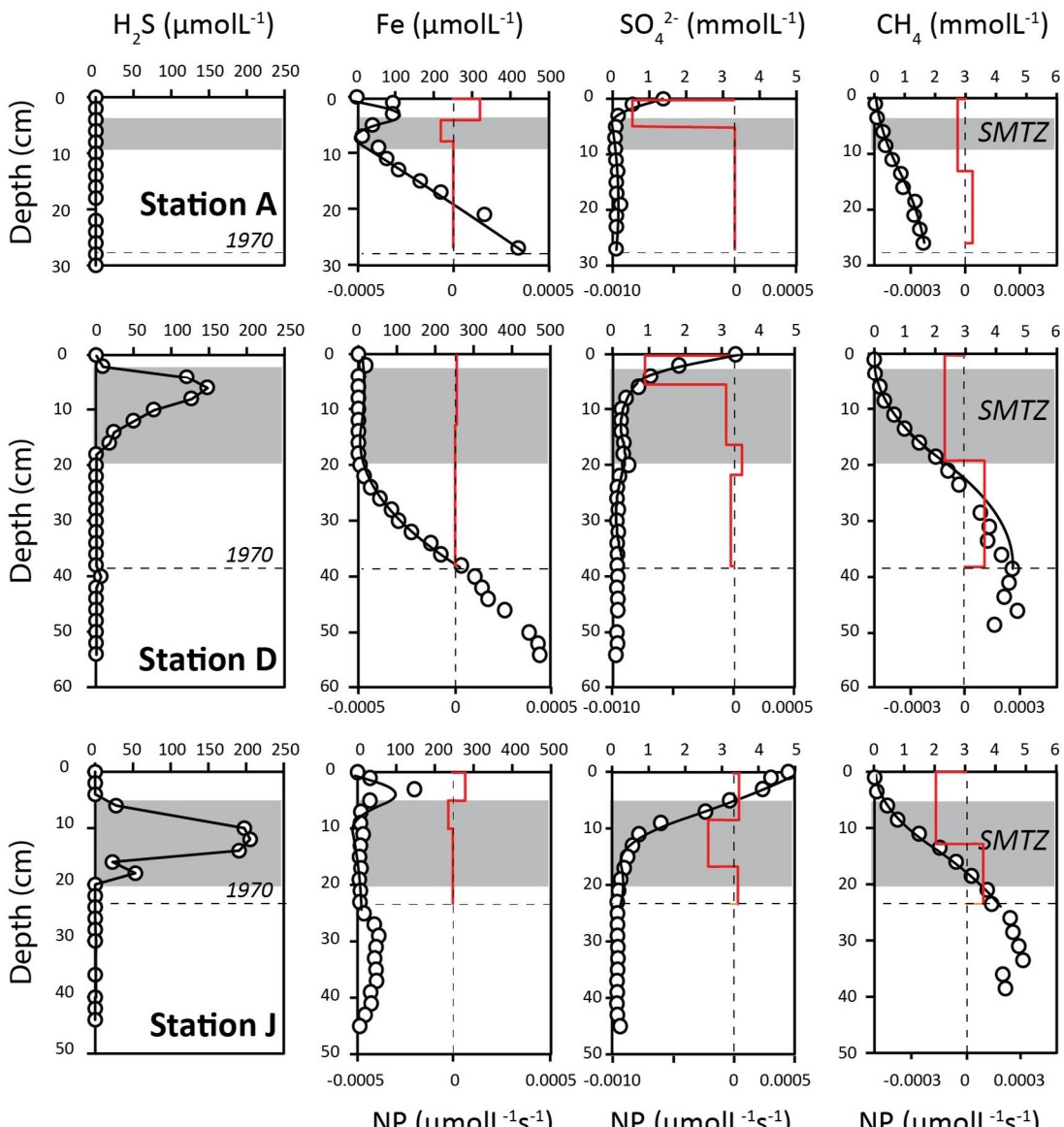

**Figure 8. Pore water data from Stations A, D and J. Hydrogen sulfide (H₂S) data are connected point-by point. The**
**depth of the H₂S peak is used to define the sulfate-methane transition (SMTZ). Where no H₂S is present (i.e., at Station**
**A), the SMTZ is defined by the corresponding minimum in pore water Fe. For all other parameters, parabolic best fit**
**lines were generated using PROFILE. These fits were used to determine the zones, and instantaneous rates, of net**
**production (NP) of each species within the depth interval 1970–2015 (red lines). The rate of net production is**

**determined from the change in gradient of the concentration as described by equation 8. Negative rates indicate net consumption.**

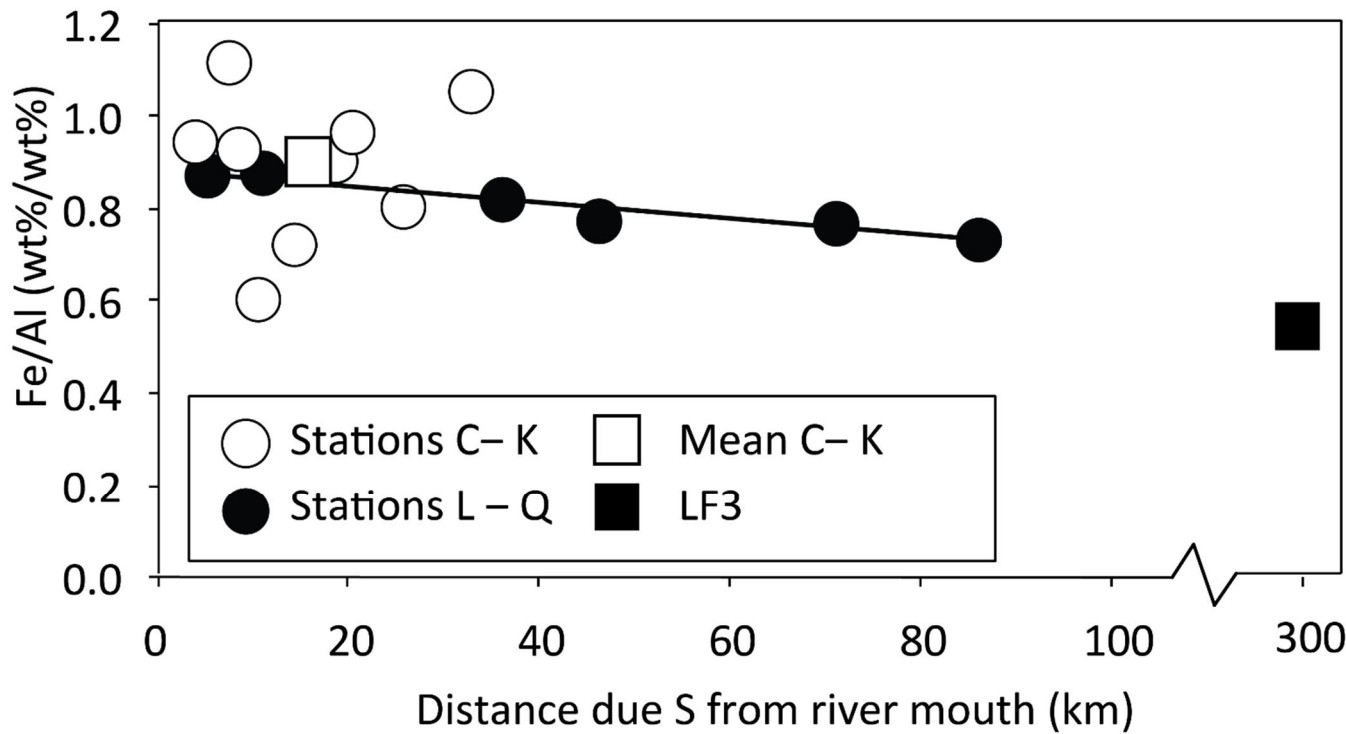

**Figure 9. Fe/Al weight ratios for surface sediment samples along the Mustionjoki transect (0–2 cm; Stations C–K) and Paimionjoki transect (0–1 cm; Stations L–Q), plotted against distance from the respective river mouth. Stations A and B have been removed from the analysis due to the influence of industrial Fe pollution (see text). Linear regression line is shown for the Paimionjoki transect. Typical values from a site on the Baltic Sea shelves (LF3, Eastern Gotland basin Fe/Al =0.5–0.6, Lenz et al., 2015) are shown for comparison. LF3 is given an arbitrary distance of 300 km from the river systems of the southern coast of Finland, although Fe at this site may be sourced from more proximal landmasses.**

**Supplementary information**

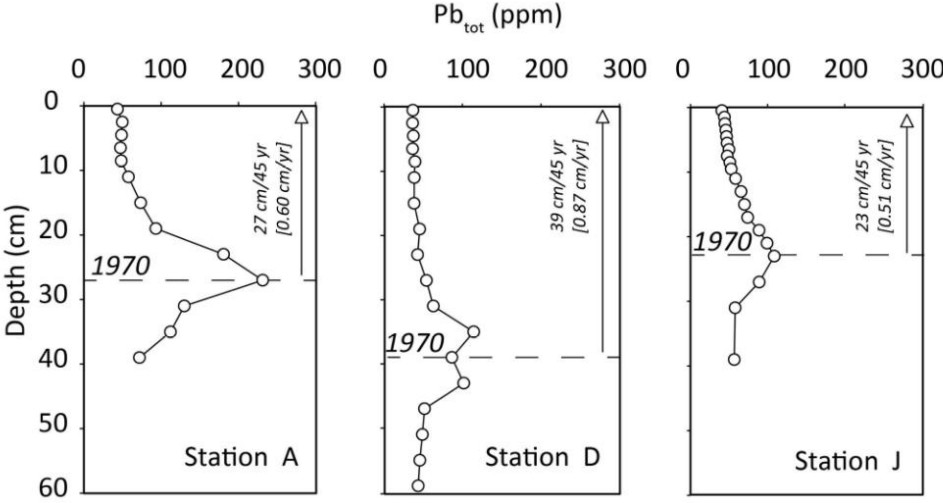

**Supplementary Figure 1.** Total lead ($P_{tot}$) concentrations in sediment cores from stations A, D and J in 2015, determined by ICP-OES. The peak in $P_{tot}$ is assumed equivalent to the year 1970 (Zillen et al., 2012). Mean sedimentation rates for each core were estimated for the interval 1970-2015 as indicated in the panels.

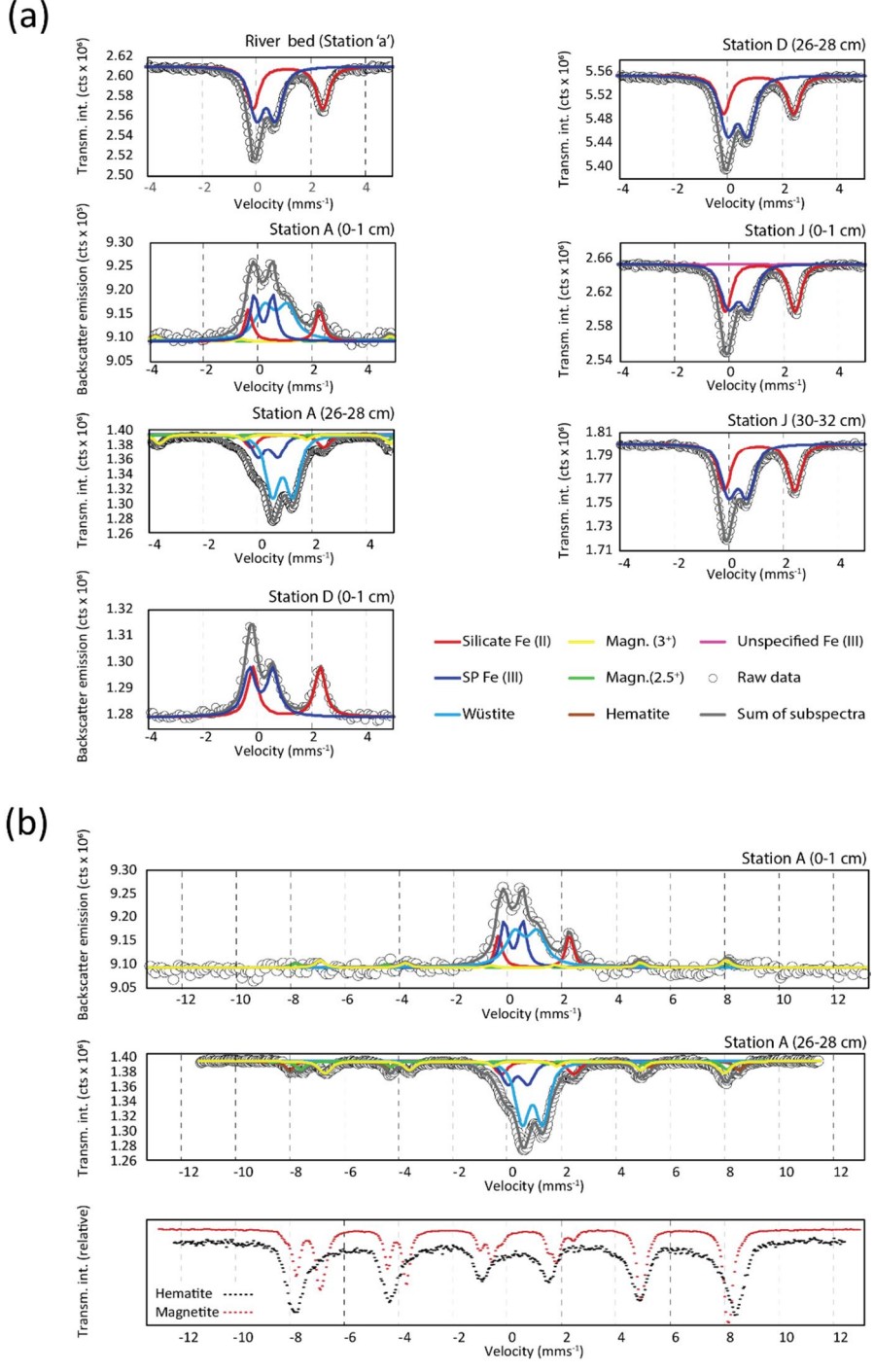

**Supplementary Figure 2. (a) Mössbauer spectra from all samples presented in true orientation. Samples from Station A (0-1 cm) and Station D (0-1 cm) were analyzed in backscatter mode. All other samples were analyzed in transmission mode. (b) Expanded spectra from Station A, showing sextet components. Reference spectra for magnetite and hematite are reproduced by permission of M. Darby Dyer from the Mineral Spectroscopy Database of Mount Holyoke College.**

