# Peer review of "Impacts of flocculation on the distribution and diagenesis of iron in boreal estuarine sediments"

_Biogeosciences, 2017_

## Referee Comment (RC1) · P. Kraal (Referee) · 27 Jun 2017

Review of BG-2017-181

With interest I have read this manuscript, in which the authors explore the role of salinity-driven flocculation of DOM and dFe in controlling the settling and diagenetic fate of riverine Fe along a salinity transect in a Baltic Sea estuary. The authors suggest a key role for flocculation in transferring Fe(III) in the form of Fe(III)-OM complexes and Fe(III) (oxyhydr)oxides from the water column to the sediment. Moreover, the Fe-OM pool is relatively stable and this Fe does not seem to participate in the "normal" reductive diagenetic pathways in organic-rich sediments. The manuscript is well-written, well-structured and an interesting addition to the flourishing research field of Fe-OM

interactions in marine and terrestrial systems.

Together with this review, I have uploaded an annotated pdf document with all my questions and comments. Below, I highlight the main questions that arose while reading the manuscript.

1. On a technical note: If I understand correctly, relative errors for all solid-phase analyses were calculated from replicate analysis of "regular" (powdered) reference materials. I wonder whether this gives relative errors that are also representative for analysis of suspended material on filters. As far as I am aware, such samples are somewhat harder to process and I am curious to know if the authors can comment on how/whether they specifically assessed analytical precision and accuracy associated with filter samples (or why not).

2. There is no information provided on which standards were analyzed with Mössbauer spectroscopy (or whether reference spectra collected previously were included) and how the selection of (number of) standards for LCF was performed. The LCF fitting routine was only explicitly mentioned in the caption to Fig. 5; it should also feature in the main methods section. The key statement "Quantification of iron-bearing phases and iron oxidation states is based on relative subspectral areas" may be expanded a bit (as it is to some extent in the notes of Table 3). Overall, the procedure of obtaining relative proportions, including that of the "undocumented" Fe phase (in my opinion an awkward term, perhaps "unknown" is more appropriate?), should be more clear, as this phase plays a rather crucial role in the manuscript. Goodness-of-fit is also an important parameter in this respect, as it is basically the (areal) mismatch between the fit and the actual spectrum that is used as a measure of "undocumented" Fe. I think it would be good if all (relevant) reference spectra (perhaps including likely candidates that were not present such as siderite) are clearly presented (they are somewhat hard to discern in the current Figure 5: perhaps a stack plot with offset would work).

3. The authors assign the "undocumented" Fe phase fully to complexes of nonsulfidized Fe(II) with organic matter (p 12, L 17-19). They base this on the study of Yu et al. (2015), who found Fe(II)-OM phase "to be a major component of sedimentary Fe in a nearby boreal estuary." In the Yu et al. study, OM-complexed Fe(II) was identified by Fe X-ray absorption spectroscopy using standards of Fe(II) and Fe(III) complexed with organic matter. As far as I could see, no empirical data in support of the assumption of OM-Fe(II) is provided in this manuscript. Because the Fe(II)-OM phase plays such an important role in the discussion, I wonder whether the authors can further substantiate their assumption that all the "undocumented" Fe, that could not be assigned to their (to the reader unknown) library of standards, was present as Fe(II)-OM? It should be clear exactly which Fe phases could be ruled out based on LCF with reference spectra. Some more focus on which spectral features could not be explained by the available reference standards, and how these may point to OM-associated Fe, would also be welcome. In the absence of a "smoking" gun, perhaps some more consideration should be given to the fact that OM-Fe(II) may not necessarily account for all "undocumented" Fe (to what extent do fitting uncertainties play a role?). In particular, the authors may want to address the validity of extrapolating Mössbauer data (key for assigning Stage 3 and 4 Fe to OM-Fe(II)) for the uppermost sample (0-1 cm) to the whole sediment record (up to 60 cm) at all sites (see also Comment 7).

4. p 14, L 13-19. The authors attribute the excess removal of Fediss relative to DOC to "preferred association of Fe with higher molecular weight compounds, which are more sensitive to flocculation (Asmala et al., 2014) or a mechanistic enhancement of flocculation by the presence of Fe (Forsgren et al., 1996)." Firstly, the second mechanism deserves some explanation (it now implies enhancement of flocculation of Fe by Fe?). Secondly, as the authors also observe a ferrihydrite signal in their Fe flux to the sediment, could flocculation of Fe independent of DOM -> POM dynamics play a role in the removal of Fe from the water column (driven by salinity, pH, perhaps DO)? The authors mention that "Flocculated material in the oxic estuarine water column is likely present as Fe (III) partitioned between organic-Fe (III) complexes and ferrihydrite (Neubauer et al., 2013)." (p 16, L 20-21). Changes in surface charge of Fe(III) particles due to

adsorption of ions at higher salinity may also affect the solubility/flocculation of Fe(III) particles?

5. The reasoning behind the conclusion "that flocculation of DOM to POM in the estuarine environment may provide the second fraction of POM detected in the N/CPOM and $\delta$13CPOM data." is somewhat unclear to this non-expert in that field (Fig. 3 and 4 and section 5.3). The authors state that the POM signal in surface waters is dominated by phytoplankton and therefore DOM-POM transitions cannot be discerned in N/CPOM vs $\delta$13CPOM plots. In fact, the N/CPOM vs $\delta$13CPOM signature of surface water-DOM from a previous study plot in the field of C3 plant material, suggesting that phytoplankton is not an important component of surface water DOM? I found this in itself surprising, as I would expect the DOM to be impacted by decomposition (POM -> DOM) of fresh algal material. The POM data from deeper waters from this study plot towards the C3 plant signature. The authors use this trend as an indication that the second OM source besides phytoplankton, i.e. C3 plant material, is transferred from DOM to POM through flocculation. I wonder whether the POM N/CPOM vs $\delta$13CPOM trend with water depth cannot also be determined/impacted by magnification of the C3 signal in the POM reservoir below the phytoplankton-dominated surface waters? I guess this strongly depends on the "rapid remineralization of fresh phytoplankton material during settling" (p 15, L8-9) which would remove the phytoplankton signal from POM and DOM and the persistence of the phytoplankton POM signal.

6. p16, L18-27. The authors interpret dithionite- and oxalate-extractable Fe as OM-Fe(II), and state that this is formed in the water column rather than the sediment. As far as I can see, this is again based on the findings of Yu et al. (2015) for a nearby boreal estuary. It may be good if the authors highlight data from this study or further literature besides Yu et al. (2015) in support of the conclusion that OM-Fe(II) is exclusively formed in the water column and is not of diagenetic origin.

7. p 17, L 4-7. The maximum accumulation of ferrihydrite in a (seasonally oxygen-depleted) "pit" of Station D is interesting, in that redox shuttling apparently causes

maximum accumulation Fh (the most labile and easily reduced Fe(III) phase) in the surface sediment (Fig. 6), where H2S already seems to accumulate (Fig. 7). Striking is also the persistence of Stage 1 and 2 Fe ("ferrihydrite") with depth through sulfidic depth intervals at Station D and to a lesser extent Station J. Could ferrihydrite perhaps be only part of the answer? The authors assign all Fe extracted by Na acetate at pH 4.5 to ferrihydrite, while this mineral is very slow to dissolve at that pH. Iron monosulfide was not specifically quantified in this study, could the presence of FeS help explain the Na acetate-extractable Fe pool and the persistence of Stage 1 and 2 Fe with depth? Egger et al. 2014, ES&T, mention that FeS in Baltic surface sediments is extracted in Stage 1 and especially Stage 2 of the same Fe extraction scheme as used in this study. The Mössbauer data indicate abundant superparamagnetic Fe(III) at Stations A and to a lesser extent Station D, but these are data from the uppermost sample (0-1 cm) that was in contact with bottom water O2 that was still $\sim$ 75 % (Station A) to 30 % (Station D) of saturation, eyeballing Fig. 1. FeS persistence far below the zone of H2S accumulation may be unlikely (unless the system is very dynamic and far from any steady state). This zone, characterized by abundant Fe2+ but no sulfide, is often associated with formation of reduced Fe minerals such as vivianite and siderite. Could these also be present here and extracted in the low pH Stage 1 and 2 of the extraction scheme? The fact that these phases were not observed in the Mössbauer data from the surface sample does not rule out their presence at depth. Overall, some more words (beyond "seasonally oxygen depleted") can be spent regarding the steadiness of a geochemical state with apparently co-occurring "Fh" and H2S accumulation, and the nature of Fe extracted from anoxic non-sulfidic and sulfidic sediments during Stages 1 and 2 of the Fe extraction scheme.

I look forward to enjoying the revised version of this manuscript.

Kind regards, Peter Kraal Utrecht University, Department of Earth Sciences-Geochemistry

[Figure]

Please also note the supplement to this comment:
https://www.biogeosciences-discuss.net/bg-2017-181/bg-2017-181-RC1-
supplement.pdf

**Supplement:**

[revised manuscript text omitted]

---

## Referee Comment (RC2) · Anonymous Referee #2 · 10 Jul 2017

This is a paper on the removal mechanisms of Fe to sediments in the riverine-estuarine transition of a long riverine estuary on the Eastern Baltic Sea and the effect of this removal for the corresponding iron diagenesis in the underlying sediment. The key message of this paper, if I understand this right, is that riverine Fe is removed by flocculation in the riverine-estuarine transition of the Eastern Baltic Sea. These results are similar to the Fe story presented in Yu et al (2015) Chemical Geology in another Finnish estuary. The flocculation occurs as an organic Fe complex.

These results are based on the interpretation of Mössbauer spectra, which revealed an unknown Fe phase that was neither well-crystallized silicate, magnetite, nor ferrihydrite. The inference is therefore not direct, but indirect, and that is a major shortcoming.

In the underlying sediments of the outer estuary and the Baltic Sea, Fe accumulation

occurs as different phases than in the riverine part, where more crystalline Fe hydroxides occurred than in the distal part where organic Fe dominates.

In principal I believe some of the story, e.g., the organic-Fe transport and rapid removal in the estuary. But I failed to see how the results translate into different diagenetic Fe processes in the sediment.

In particular, I felt that the story on the anaerobic oxidation of methane by iron was underdeveloped. While this is an impressively large data set from many stations, much of the potential novelty hinges upon the interpretation of the Mössbauer data. The combination of Mössbauer/extraction data call for a major reinterpretation of the operational Fe extractions, in particular of a re-assessment of the dithionite extraction as an organic Fe phase. This has large ramifications for many published papers and the authors need to be very careful in their assessment and interpretations.

I would like the authors to address and comment on a number of questions I have:

Why do you not present Mössbauer spectra of riverine material, dissolved and particulate? This would be most interesting to see.

Why do you claim that hydroxylamine-extracted Fe occurs as Fe2+, when hydroxylamine is a strong reducing agent? No information on prior oxidation state is possible using this extraction method.

Why do you not even consider or discuss the extraction of FexSy phases with dithionite? This is well known. Not all Fe may be organically associated to the same extent throughout the transect and not necessarily as Fe2+, because dithionite is also a reducing agent.

In your table or on the Mössbauer spectra you should show the patterns for FexSy or FeS2 phases to convince the reader that the composite spectrum is not influenced by these phases.

The Mössbauer spectroscopy standardization and reference spectra are not explained.

[Figure]

It is not clear why your conclusion is that the phase must be an organic Fe2+ phase. If it is ferrihydrite associated with organic matter, what kind of association is this?

Is there a possibility that the unknown spectrum is an amorphous Fe-silica phase?

Why is it that there are hardly any changes in Fe speciation at Station D, although the S content changes so significantly and therefore likely the concentrations of FeS and FeS2?

What is the major Fe carrier down the river, i.e., what is the speciation of Fe in river and estuarine water?

How much is associated with the organic fraction – how much is present as ferrihydrite – what is the exact molecular association?

Figure 5 figure caption and left figure panel don't agree. The caption reads Fe2+/FeT, the other Fe3+/FeT ratio.

Figure 6: The down-core operational Fe profiles don't exactly make sense in light of the pronounced changes in S content with depth at Station A. At least, the 100% scaling makes it difficult to associate the species changes with the S concentration changes.

I recommend to show the Fe species as concentrations, e.g., as summed bars totaling to the actual Fe concentration. That would help at least for comparing the data of Station A. The Fe species do not correspond at all to the sulfur concentrations. How is this possible, if FeS/FeS2 forms? My conclusion would be that the dithionite-extracted species seem to be associated at least partly with some FeS/FeS2. This needs to be accounted for.

p.14 l.25: I think the authors mean 'isotopically depleted'

p.16, l.16-17 Fe-Si amorphous phases; FeS mackinawite-like material?: Why Fe2+, could also have been Fe3+?

p.17, l.1-5 hydroxylamine hydrochloride is a strong reducing agent suggesting that the

Fe could have been reduced by the extraction. I would also like to see a transect plot of DOM and POM.

p.18. Are the salinity differences significant enough to impact the Fe-S system? I don't think so.

p.18 l.24 what do you mean by background rates: How are these background rates? Aren't these the major reactions? What about sulfate reduction rates and coupled oxidation of sulfide by ferrihydrite?

The anaerobic oxidation of methane by iron is often invoked these days, but to argue for this process there has to be good direct evidence. I am sure the authors are aware that AOM also can be coupled indirectly to reduction of iron oxyhydroxides through sulfide oxidation, but cannot be distinguished easily without performing specific experiments. Concentration profiles alone are not enough. The authors should refrain from inferring that AOM by iron is a major process controlling deep iron diagenesis when they have not addressed sulfide oxidation processes. They do not even present DIC data to support their assertion. In addition, should this process occur, it is easy to assess the quantitative significance by assessing the methane flux and the required removal of Fe to account for methane oxidation.

Along these lines, generally there is also too little discussion on sulfide oxidation coupled to iron reduction in the surface sediments. These organic-rich sediments likely have very low oxygen penetration depths of a few mm. Based on many other studies in estuarine systems, it is likely that anaerobic degradation processes such as sulfate reduction commence in the first centimeter. This makes it possible that FeS phases already occur in the topmost cm, and that not only iron reduction, but co-existing iron and sulfate reduction take place in the topmost cm.

Finally, although the authors do very well in describing the bathymetry of this estuarine system, they fail to associate the bathymetric features with the current transport/hydrography and the resulting particle transport and accumulation. For example,

[Figure]

Station D likely must be influenced by saline water transport upstream, which is the only explanation to explain the higher bottom salinities. Therefore, inshore/upstream transport of organic material and of Fe has to be considered for the deep depressions. In addition, Station D, being the deepest station of the inner estuary, is clearly a particle trap of fine-grained material, also of organic material. As such, the focusing and accumulation of material here may override the estuarine mixing signal the authors have as their overriding study target. Stations C or E may be more informative in this context. Do the authors note the same signals at stations C, D, and E?

---

## Author Comment (AC2) · 30 Sep 2017

Dear referees,

We wish to thank you for the careful consideration of our manuscript. Please find in the attached document detailed responses to each of your comments. Because our responses to Referee #1 and Referee #2 cross-reference each other substantially, we have uploaded a single document. The responses to Referee #1 are found on pages 1-13; responses to Referee #2 on pages 14-20.

Kind regards on behalf of all co-authors,

Tom Jilbert University of Helsinki, Finland

[Figure]

Please also note the supplement to this comment:
https://www.biogeosciences-discuss.net/bg-2017-181/bg-2017-181-AC2-
supplement.pdf

---

## Author Response (AR1)

**Authors' response document bg-2017-181. Tom Jilbert 19.11.2017**

**Contents**

1. **Responses to online discussion Referee #1 (including list of revisions): pages 1–16**
2. **Responses to online discussion Referee #2 (including list of revisions): pages 17–25**
3. **Marked up revised manuscript: separate page numbering (pages 1–74)**

**1.Responses to online discussion Referee #1 (Peter Kraal)**

The following document contains both the responses given during the Open Discussion of "Flocculation of dissolved organic matter controls the distribution of iron in boreal estuarine sediments" and a description of the subsequent revisions leading to the submission of "Impacts of flocculation on the distribution and diagenesis of iron in boreal estuarine sediments".

**[Referee comments in bold]**

*[Responses in italics]*

[Indication of subsequent revisions in regular text]

**With interest I have read this manuscript, in which the authors explore the role of salinity-driven flocculation of DOM and dFe in controlling the settling and diagenetic fate of riverine Fe along a salinity transect in a Baltic Sea estuary. The authors suggest a key role for flocculation in transferring Fe(III) in the form of Fe(III)-OM complexes and Fe(III) (oxyhydr)oxides from the water column to the sediment. Moreover, the Fe-OM pool is relatively stable and this Fe does not seem to participate in the "normal" reductive diagenetic pathways in organic-rich sediments. The manuscript is well-written, well-structured and an interesting addition to the flourishing research field of Fe-OM interactions in marine and terrestrial systems.**

**Together with this review, I have uploaded an annotated pdf document with all my questions and comments. Below, I highlight the main questions that arose while reading the manuscript.**

*We thank the referee for the careful consideration of our manuscript. Below, we respond to each of the major comments followed by the line-by-line comments extracted from the referee's annotated pdf.*

**1. On a technical note: If I understand correctly, relative errors for all solid-phase analyses were calculated from replicate analysis of "regular" (powdered) reference materials. I wonder whether this gives relative errors that are also representative for analysis of suspended material on filters. As far as I am aware, such samples are somewhat harder to process and I am curious to know if the authors can comment on how/whether they specifically assessed analytical precision and accuracy associated with filter samples (or why not).**

*For the analysis of suspended organic matter, complete filters were combusted after packing and compression in tin cups. This avoids issues of heterogeneity within individual filters and this detail will be added to the methods section. We did not take systematic replicates for all GF/F filters due to limitations of sample volume and processing time. However for each site we did take an additional*

*water column sample from <1m above the seafloor. At the stations where this sample is close to (within 2 vertical meters of) the deepest 5m depth interval sample, we suggest it may be used as a replicate to assess precision between filters (with the added value of coming from a separate sampling cast rather than simply being a duplicate from the same Limnos bottle). When this is done we see that the $\delta^{13}C$ and N/C values for the extra sample are within the analytical error of the sample from the deepest 5m depth interval (see Fig. R1). Hence we conclude that precision for this data is in fact limited by the analysis and not by the sampling procedure. In terms of accuracy, this can only be determined by reference materials analyzed in parallel to the samples. This was done routinely (two standards for every ten samples) and showed accuracy to be <2.5% as stated in the manuscript.*

*When the corresponding exercise is carried out for $Fe_{part.}$ from the polycarbonate filters we observe relative standard deviation (RSD) of up to 15% for "replicate" deep water samples. This is substantially greater than the RSD values for $\delta^{13}C$ and N/C (both less than 1 % in Fig. R1). We interpret this as a consequence of the fact that $\delta^{13}C$ and N/C describe the characteristics of the suspended organic material, but are largely insensitive to variations in its total concentration at a given location. $Fe_{part.}$, on the other hand, describes the absolute concentration of particulate Fe in µmol/L, which (similarly to the absolute concentration of organic matter) is likely to vary on small spatial and temporal scales and may well change in the 10–15 minute interval between Limnos casts at a given location. It should however be noted that the error associated with an RSD of 15%, particularly for low-Fe samples, is several orders of magnitude less than the large-scale*

[Figure]

*Figure R1: N/C (mol) and $\delta^{13}C$ of suspended particulate matter at Stations B and C. All data presented here, except for the deepest sample at each station (taken from <1 m above the seafloor), are included in Fig. 3 of the manuscript. Here, the sample from <1 m above the seafloor is treated as a replicate for the deepest 5m sampling interval for each station (depth offset = approx. 1 m for Station B, approx. 2 m for Station C). Horizontal error bars indicate analytical precision, reported as one standard deviation, as determined by 10 repeated measurements of standard materials (N/C =0.005, $\delta^{13}C$ = 0.15 ‰).*

*changes in $Fe_{part.}$ observed along the salinity gradient which are discussed in the manuscript (Mean value at Station A = 2.54 µmol/L, σ = 1.98 (n = 2); mean value at Station K = 0.30 µmol/L, σ = 0.10 (n = 10)). The concentration of $Fe_{part.}$ at Station A is thus 25 times greater than at Station K.*

The revised manuscript makes reference to the above clarifications (Section 3.2).

**2. There is no information provided on which standards were analyzed with Mössbauer spectroscopy (or whether reference spectra collected previously were included) and how the selection of (number of) standards for LCF was performed. The LCF fitting routine was only explicitly mentioned in the caption to Fig. 5; it should also feature in the main methods section.**

**The key statement "Quantification of iron-bearing phases and iron oxidation states is based on relative subspectral areas" may be expanded a bit (as it is to some extent in the notes of Table 3). Overall, the procedure of obtaining relative proportions, including that of the "undocumented" Fe phase (in my opinion an awkward term, perhaps "unknown" is more appropriate?), should be more clear, as this phase plays a rather crucial role in the manuscript. Goodness-of-fit is also an important parameter in this respect, as it is basically the (areal) mismatch between the fit and the actual spectrum that is used as a measure of "undocumented" Fe. I think it would be good if all (relevant) reference spectra (perhaps including likely candidates that were not present such as siderite) are clearly presented (they are somewhat hard to discern in the current Figure 5: perhaps a stack plot with offset would work).**

*We thank the referee for highlighting the need for a more detailed presentation of the Mössbauer approach. To clarify, the LCF fitting was performed using reference spectra rather than freshly prepared and analyzed standards. In the revised version, we will list the reference materials considered, present their spectra, describe how the selection for the LCF was performed, and expand the discussion of the LCF fitting procedure itself including goodness of fit. Furthermore we are in the process of generating Mössbauer spectra of additional sediment samples from the estuarine transect. When presenting these results in the revised manuscript we will consider the referee's terminological suggestions regarding the use of the terms "undocumented" vs. "unknown" Fe phases.*

The revised manuscript includes substantially expanded sections on Mössbauer spectroscopy, including further details of the methodology (Section 3.10), and new results and revised interpretations of the spectra (Section 4.7).

**3. The authors assign the "undocumented" Fe phase fully to complexes of non-sulfidized Fe(II) with organic matter (p 12, L 17-19). They base this on the study of Yu et al. (2015), who found Fe(II)-OM phase "to be a major component of sedimentary Fe in a nearby boreal estuary." In the Yu et al. study, OM-complexed Fe(II) was identified by Fe X-ray absorption spectroscopy using standards of Fe(II) and Fe(III) complexed with organic matter. As far as I could see, no empirical data in support of the assumption of OM-Fe(II) is provided in this manuscript. Because the Fe(II)-OM phase plays such an important role in the discussion, I wonder whether the authors can further substantiate their assumption that all the "undocumented" Fe, that could not be assigned to their (to the reader unknown) library of standards, was present as Fe(II)-OM? It should be clear exactly which Fe phases could be ruled out based on LCF with reference spectra. Some more focus on which spectral features could not be explained by the available reference standards, and how these may point to OM-associated Fe, would also be welcome. In the absence of a "smoking" gun, perhaps some more consideration should be given to the fact that OM-Fe(II) may not necessarily account for all "undocumented" Fe (to what extent do fitting uncertainties play a role?). In particular, the authors may want to address the validity of extrapolating Mössbauer data (key for assigning Stage 3 and 4 Fe to OM-Fe(II)) for the uppermost sample (0-1 cm) to the whole sediment record (up to 60 cm) at all sites (see also Comment 7).**

*Our detailed response to this comment is partly dependent on the ongoing Mössbauer analysis of an additional five sediment samples from the estuarine transect. In total, the new dataset will comprise seven samples (riverbed sediment, shallow + deep sediment from Station A, shallow + deep sediment from Station D, shallow + deep sediment from Station J). At the time of writing, four of the additional 5 samples have been measured and the fifth is in progress. Analysis of these spectra will be performed by LCF using references, and as stated in the response to the previous comment, the references used*

in the LCF (and the rationale for their selection) will be explicitly stated in the manuscript, complete with presentation of spectra.

See previous. The revised manuscript includes substantially expanded sections on Mössbauer spectroscopy, including further details of the methodology (Section 3.10), and new results and revised interpretations of the spectra (Section 4.7).

We have also performed additional sediment extractions on 15 samples from Stations A, D and J to investigate the robustness of our initial conclusions concerning the dominance of OM-Fe(II) in Stages 3 and 4 of the Poulton and Canfield (2005) method. These results will be included in the revised manuscript and considered alongside the new Mössbauer data. In brief, we carried out parallel extractions of subsamples dried under $N_2$ after frozen storage, and treated under anoxic conditions as follows:

Subsample 1: 1M HCl, 1 hr (including trap for $H_2S$ evolved from Acid Volatile Sulfur)
Subsample 2: 1 M HCl + 1 M hydroxylamine-HCl, 1 hr
Subsample 3: 0.2 M Sodium citrate at pH 4.8, 2 hr

Fe (II) in each extract was then determined by spectrophotometry after complexation with 1,10 phenanthroline, while AVS was determined by iodometric titration. This approach is similar to that used by Yu et al. (2015) to determine sulfidized vs. unsulfidized Fe (II), and Fe(II)/Fe(III), in the cold 1M HCl-soluble fraction of boreal estuarine sediments. However we also included Subsample 3 to mimic Stage 3 of the Poulton and Canfield extraction – but excluding dithionite – to test the hypothesis that the citrate ligand alone may be able to extract Fe (II) direct from OM-complexes. This specifically addresses a comment of Referee #2 concerning our interpretations of citrate-dithionite-soluble (i.e. nominally reducible) Fe as Fe (II).

The results of the additional extractions suggest that our initial assumptions concerning the solubility of OM-Fe (II) in Stages 3 and 4 may indeed require modification. We do detect unsulfidized Fe(II) in the HCl extracts, and its concentration in most samples is higher than both sulfidized Fe (II) (AVS) and HCl-soluble Fe (III) (derived from the difference between Subsamples 1 and 2, see Figs. R2 and R3). Furthermore the parallel citrate-only extraction does appear to dissolve approximately 60% of the unsulfidized Fe (II) pool (see Fig. R4). While these observations support our claim that OM-Fe(II) complexes are present in the sediments and suggest that at least a fraction of this material is citrate-soluble, the total amount of Fe dissolved in the additional extracts is equivalent to (only) the combined total from Stages 1 and 2 of the Poulton and Canfield (2005) procedure (Fig. R5). Hence, the Fe phases dissolved by 1M HCl are likely identical to those dissolved by sodium acetate and hydroxylamine-HCl, while the Stage 3- (citrate-dithionite) soluble Fe fractions remain largely intact during the 1M HCl extraction.

[Figure]

191
192
193

*Fig. R2:* *Comparison of Fe extraction data for the Poulton and Canfield (2005) method reported in the original manuscript (upper panels), and additional extractions with 1M HCl performed in response to the interactive discussion comments (lower panels). All panels are plotted on the same scales for comparability. Where possible, samples for the additional extractions were selected from the set originally extracted by the Poulton and Canfield method. When no sample material was remaining, adjacent samples were taken (n=4).*

194
195

[Figure]

196

*Fig. R3:* *Zoom of lower panels of Fig. R2 for clarity.*

197
198
199

[Figure]

Fig. R4: Comparison of citrate-soluble Fe (II) (Subsample 3) and total unsulfidized Fe (II) (derived from Subsample 1) for the additional extraction samples. Note that approximately 60% of total 1M HCl-soluble unsulfidized Fe (II) is extracted by the citrate solution, as given by the relationship in the final panel (solid line in this panel indicates 1:1).

202
203

[Figure]

Fig. R5. Relationship between Fe extracted in Stages 1+2 of the Poulton and Canfield method, and total 1M HCl-extractable Fe in the additional extractions. Where no equivalent sample was available, adjacent samples have been compared (n =4). Dashed line represents 1:1 and the least-squares regression is performed against this line.

204
205
206 *Therefore, assuming the 1M HCl extraction to be a reliable determinant for OM-Fe(II) as suggested*
207 *by Yu et al., 2015, we should modify our interpretation of the composition of the phases dissolved in*
208 *Stages 3 and 4 of the Poulton and Canfield method. We hope that the forthcoming Mössbauer data*
209 *will clarify this issue, especially the deep sample from Station A, whose combined Stage 3 and Stage*
210 *4-soluble Fe pool is in excess of 3000 µmol/g sediment (i.e. 16.8% of the sediment by weight).*
211
212 We have revised our interpretations of the Fe speciation substantially on the basis of both the
213 additional extraction results and the new Mössbauer data. Figures R2 , R3 and R5 are incorporated
214 into the revised manuscript (in Figures 5, 6a and 6b respectively) and an extensive new interpretation
215 of Fe phases in the sediments is given (Table 3, Sections 4.6, 4.7).
216
217 **4. p 14, L 13-19. The authors attribute the excess removal of Fediss relative to DOC to**
218 **"preferred association of Fe with higher molecular weight compounds, which are more sensitive**
219 **to flocculation (Asmala et al., 2014) or a mechanistic enhancement of flocculation by the**
220 **presence of Fe (Forsgren et al., 1996)." Firstly, the second mechanism deserves some**
221 **explanation (it now implies enhancement of flocculation of Fe by Fe?). Secondly, as the authors**

**also observe a ferrihydrite signal in their Fe flux to the sediment, could flocculation of Fe independent of DOM -> POM dynamics play a role in the removal of Fe from the water column (driven by salinity, pH, perhaps DO)? The authors mention that "Flocculated material in the oxic estuarine water column is likely present as Fe (III) partitioned between organic-Fe (III) complexes and ferrihydrite (Neubauer et al., 2013)." (p 16, L 20-21). Changes in surface charge of Fe(III) particles due to adsorption of ions at higher salinity may also affect the solubility/flocculation of Fe(III) particles?**

*We will clarify these statements in the revised manuscript. Several authors have noted that the respective behavior of Fe and bulk OM along salinity gradients in Baltic Sea estuaries differ significantly, with non-conservative behavior being much more apparent for Fe. This may indeed imply additional mechanisms for flocculation that are specific to Fe, rather than simply to the fraction of OM to which Fe is associated (the context within which this statement was written). These include, as the referee suggests, direct flocculation of Fe oxides in response to pH changes. Typically, pH of the the Mustionjoki river drainage is in the order 6.0–6.5 (Lahermo et al., Geochemical Atlas of Finland, 1996). Along the salinity gradient in the surface waters of the estuary, pH indeed rises, towards the offshore value of 8.0–8.4 in the open Gulf of Finland (exact value depends on season, e.g., Omstedt et al., Tellus B, 62B, 2010), due to mixing between fresh and brackish water masses. As outlined in Neubauer et al., ES&T 47, 2013, pH – independent of salinity – may determine the partitioning of Fe between NOM-Fe complexes and ferrihydrite, and between different size classes of ferrihydrite in natural waters (their Figure 1e). In their model, freshwater at pH 6 should already contain a substantial fraction of Fe oxides that would not pass through a 0.2μm filter. We used 0.4μm filters, but nevertheless measured a majority of Fe in the particulate phase in the freshwater endmember sample (surface water of Station A, Fig. 2) which supports the Neubauer et al. model for a pH 6.0–6.5 river. However we see an immediate loss of particulate Fe between Stations A and B, despite the onset of the salinity gradient and hence the rise in pH. This suggests that also current strength influences the concentration of this material in the water column, due to its susceptibility to sedimentation. Further offshore, we observe a second maximum in particulate Fe (Fig. 2) which we attribute to flocculated formerly dissolved material. We cannot discount that pH, increasing in tandem with salinity, influences this process. The density of minerogenic matter such as clays may also influence the likelihood of flocculation in boreal estuaries (as discussed in Forsgren et al., 1996).*

In the revised manuscript we emphasize the partial decoupling of Fe from DOM during flocculation (see revised Title, Abstract, Introduction paragraph 4, Section 5.2).

**5. The reasoning behind the conclusion "that flocculation of DOM to POM in the estuarine environment may provide the second fraction of POM detected in the N/CPOM and 13CPOM data." is somewhat unclear to this non-expert in that field (Fig. 3 and 4 and section 5.3). The authors state that the POM signal in surface waters is dominated by phytoplankton and therefore DOM-POM transitions cannot be discerned in N/CPOM vs 13CPOM signature of surface water-DOM from a previous study plot in the field of C3 plant material, suggesting that phytoplankton is not an important component of surface water DOM? I found this in itself surprising, as I would expect the DOM to be impacted by decomposition (POM -> DOM) of fresh algal material.**

*To answer this point first: It is true that degradation of phytoplankton material may theoretically impact on DOM content (and therefore net composition). However in this system the mass balance argues against such a hypothesis. Concentrations of river-derived DOM in the estuary are an order of magnitude greater than the potential supply from degrading phytoplankton. In the study of Asmala et al., Biogeosciences 10, 2013, DOM concentrations along the salinity transect of the Mustionjoki*

272  *estuary declined from 585 μmol/L in river water to 363 μmol/L in the offshore endmember (average*
273  *of three seasons). As shown in the same study (and in our Fig. 3), despite transformations in the*
274  *estuarine environment including flocculation and utilization of DOM in foodwebs, bulk DOM retains*
275  *a strongly "terrestrial" C/N (or (N/C) ratio even in the offshore region. In this study, we measured*
276  *POC in the range of 20–60 μmol/L for most stations (see Fig. R7 and convert to molar units). Hence,*
277  *complete degradation of this material to DOM would yield only minor changes in the C/N (or (N/C)*
278  *ratio. We will rephrase the text to emphasize this.*

280  In the revised manuscript we emphasize the magnitude of the DOM pool in this system and its
281  persistent terrestrial character as determined by N/C and $\delta^{13}$C (Section 5.1).

283  **The POM data from deeper waters from this study plot towards the C3 plant signature. The**
284  **authors use this trend as an indication that the second OM source besides phytoplankton, i.e.**
285  **C3 plant material, is transferred from DOM to POM through flocculation. I wonder whether**
286  **the POM N/CPOM vs 13CPOM plots cannot also be determined/impacted by magnification of**
287  **the C3 signal in the POM reservoir below the phytoplankton-dominated surface waters? I guess**
288  **this strongly depends on the "rapid remineralization of fresh phytoplankton material during**
289  **settling" (p 15, L8-9) which would remove the phytoplankton signal from POM and DOM and**
290  **the persistence of the phytoplankton POM signal.**

292  *Our conceptual model for interpreting the plots in Fig. 3 is that there are essentially two end-members*
293  *of POM in the estuarine water column: phytoplankton and (C3) terrestrial plant material. At any*
294  *given location (water depth, distance from river mouth), we then observe a mixture of phytoplankton*
295  *and terrestrial material whose proportions determine the bulk N/C$_{POM}$ and $\delta^{13}$C$_{POM}$. In deeper waters,*
296  *these bulk signals are closer to those of the plant material, partly because remineralization of sinking*
297  *phytoplankton detritus depletes the contribution of this end-member. If we understand the referee's*
298  *question correctly, it concerns the possibility that plant material in the bulk POM may derive from a*
299  *direct POM input to the estuary (rather than from flocculated DOM). This is indeed a possibility*
300  *which cannot be ruled out with the data presented. However as demonstrated by Mattsson et al.,*
301  *Biogeochemistry 76, 2005 in a study of 86 Finnish river catchments, 94% of TOC in river water is*
302  *present as DOC. This implies that transformations of DOC in the estuarine water column are a more*
303  *likely source of plant matter-derived POC than a direct POC input from the catchment. Furthermore*
304  *DOM in the Mustionjoki estuary is known to undergo non-conservative mixing at low salinities*
305  *(considered mainly to be due to flocculation of DOC to POC as outlined in Asmala et al., JGR*
306  *Biogeosciences 10.1002, 2014). Hence we stand by the initial interpretation that the terrestrial plant*
307  *material end-member of POM is likely derived from flocculated DOM.*

309  See previous. In the revised manuscript we emphasize the magnitude of the DOM pool in this system
310  and its persistent terrestrial character as determined by N/C and $\delta^{13}$C (Section 5.1).

312  **6. p16, L18-27. The authors interpret dithionite- and oxalate-extractable Fe as OMFe(II), and**
313  **state that this is formed in the water column rather than the sediment. As far as I can see, this**
314  **is again based on the findings of Yu et al. (2015) for a nearby boreal estuary. It may be good if**
315  **the authors highlight data from this study or further literature besides Yu et al. (2015) in**
316  **support of the conclusion that OM-Fe(II) is exclusively formed in the water column and is not**
317  **of diagenetic origin.**

319  *As outlined in the response to comment 3, our interpretations concerning the role of OM-Fe(II) will*
320  *require modification. The additional 1M HCl extraction data confirm that unsulfidized Fe (II) is*
321  *present in all measured samples (n=15) but the comparison in Fig. R5 shows that it likely contributes*

*to Stages 1+2-soluble Fe (rather than Stages 3+4-soluble Fe) in the Poulton and Canfield method. In fact we did not claim that the OM-Fe(II) complexes were formed in the water column, rather that the association between Fe and OM may initially occur in the water column (where Fe is present as Fe(III)) and persist in the sediments (where Fe is then reduced to Fe(II)). In any case we will rewrite this section based on the new information from the additional extractions and Mössbauer analyses.*

In the revised manuscript, we have updated our interpretations of the Fe speciation substantially on the basis of both the additional extraction results and the new Mössbauer data (Figures 6, 7, Table 3) With respect to Fe (II)-OM, we infer the presence of this component from persistent detection of unsulfidized Fe (II), including in depth horizons where the presence of $H_2S$ should prevent accumulation of unsulfidized Fe (II) mineral phases such as siderite and vivianite (Section 4.6).

**7. p 17, L 4-7. The maximum accumulation of ferrihydrite in a (seasonally oxygen depleted) "pit" of Station D is interesting, in that redox shuttling apparently causes maximum accumulation Fh (the most labile and easily reduced Fe(III) phase) in the surface sediment (Fig. 6), where H2S already seems to accumulate (Fig. 7). Striking is also the persistence of Stage 1 and 2 Fe ("ferrihydrite") with depth through sulfidic depth intervals at Station D and to a lesser extent Station J. Could ferrihydrite perhaps be only part of the answer? The authors assign all Fe extracted by Na acetate at pH 4.5 to ferrihydrite, while this mineral is very slow to dissolve at that pH. Iron monosulfide was not specifically quantified in this study, could the presence of FeS help explain the Na acetate-extractable Fe pool and the persistence of Stage 1 and 2 Fe with depth? Egger et al. 2014, ES&T, mention that FeS in Baltic surface sediments is extracted in Stage 1 and especially Stage 2 of the same Fe extraction scheme as used in this study.**

*To answer this point first: As outlined in the response to comment 3 we now have a more detailed picture of the likely composition of the Na-acetate- and hydroxylamine-HCl-soluble fractions, which helps to answer these questions. First we recall that total 1M HCl-soluble Fe is equivalent to total Stage 1+2-soluble Fe from the Poulton and Canfield method (Fig. R5), implying that the same phases are dissolved in both approaches. The results of the 1M HCl extractions therefore confirm the referee's suggestion that FeS accounts for a portion of total Na-acetate- and/or hydroxylamine-HCl-soluble Fe (Fig. R3). The remainder is made up of unsulfidized Fe (II) and HCl-soluble Fe (III). It is notable, as the referee also states, that Station D is characterized by the highest total proportions of these "labile" phases (Figs. R2 and R3) which is very likely a consequence of lateral Fe shuttling into the bathymetric depression, followed by diagenetic cycling of Fe in the sediment. This diagenetic cycling may include reduction of ferrihydrite and precipitation as FeS.*

*Although the additional extractions were performed on only 5 samples per station – and therefore trends with depth in the relative proportions of FeS, unsulfidized Fe(II) and HCl-soluble Fe(III) are difficult to discern – the data suggests that HCl-soluble Fe (III) is indeed present in most samples. This implies a persistent occurrence of ferrihydrite at depth in the sediments throughout the transect, as originally suggested in the manuscript. An alternative theory would be that the HCl-soluble Fe (III) is contributed by labile OM-Fe(III) complexes, which may be less susceptible than ferrihydrite to reaction with $H_2S$. These possibilities will be discussed in more detail in the revised version. (We note that three of the 15 samples appear not to yield any HCl-soluble Fe(III) (Fig. R3), but that this may be an artefact caused by heterogeneity between Subsamples 1 and 2, which were used to estimate HCl-soluble Fe (III) from the difference between HCl-soluble Fe (II) and HCl-soluble Fe$_{tot.}$).*

The roles of ferrihydrite and iron monosulfide, as well as other labile Fe phases, are discussed at length in the revised manuscript in the context of the new extraction data (Section 4.6).

**The Mössbauer data indicate abundant superparamagnetic Fe(III) at Stations A and to a lesser extent Station D, but these are data from the uppermost sample (0-1 cm) that was in contact with bottom water O2 that was still 75 % (Station A) to 30 % (Station D) of saturation, eyeballing Fig. 1. FeS persistence far below the zone of H2S accumulation may be unlikely (unless the system is very dynamic and far from any steady state). This zone, characterized by abundant Fe2+ but no sulfide, is often associated with formation of reduced Fe minerals such as vivianite and siderite. Could these also be present here and extracted in the low pH Stage 1 and 2 of the extraction scheme? The fact that these phases were not observed in the Mössbauer data from the surface sample does not rule out their presence at depth. Overall, some more words (beyond "seasonally oxygen depleted") can be spent regarding the steadiness of a geochemical state with apparently co-occurring "Fh" and H2S accumulation, and the nature of Fe extracted from anoxic non-sulfidic and sulfidic sediments during Stages 1 and 2 of the Fe extraction scheme.**

*We are currently analyzing five additional samples by Mössbauer, including one sample from the deeper part of the core from Station D (26–28 cm), below the H2S zone. As indicated above, these data will be included in the revised version of the manuscript along with the relevant reference spectra and expanded details of the LCF procedure. The referee is correct that precipitation of Fe(II) phosphates and carbonates is theoretically possible in this depth interval, as shown for example for Bothnian Sea sediments in Egger et al., GCA 169, 2015. We will acknowledge this in the text as well considering these phases in the interpretation of the Mössbauer data. It is also indeed noteworthy that Station D appears to display co-occurrence of "ferrihydrite" and H2S in the depth interval 5–20 cm, whereas ferrihydrite is expected to react with H2S with a half-life measured in hours (Table 4.1 in Raiswell and Canfield, Geochemical Perspectives 1, 2012). Although we have only sampled Station D pore waters on one occasion, we expect from nearby – repeatedly sampled – stations that the H2S peak will be persistent throughout the year, which supports the need for an explanation why "ferrihydrite" should survive in the sediments. As outlined in the previous response, one possibility is that the HCl-soluble Fe(III) is in fact labile OM-Fe(III), which may be less susceptible than ferrihydrite to reaction with H2S. Another is that ferrihydrite is indeed present, but somehow protected from reaction with H2S by surface sorption processes. We note the referee's own recent conference abstract (Kraal et al., Goldschmidt 2017) which suggests a greater chemical stability for ferrihydrite particles to which P and Si is sorbed.*

The respective behavior of various labile Fe phases is discussed at length in the revised manuscript in the context of the new extraction data (Section 4.6). We currently favor the hypothesis that ferrihydrite is indeed dissolved in the SMTZ and that any retained labile Fe (III) is therefore present at Fe (III)-OM.

**I look forward to enjoying the revised version of this manuscript.**

**Kind regards, Peter Kraal Utrecht University, Department of Earth Sciences-Geochemistry**
**Please also note the supplement to this comment:**
**https://www.biogeosciences-discuss.net/bg-2017-181/bg-2017-181-RC1-supplement.pdf**

Responses to Line by Line supplementary comments from P. Kraal (Referee #1)

**P3 Line 6: And, conversely, OM has been shown to stabilize ferrous Fe (Toner et al, Nat Geosci 2, 2009)**

*We will add the reference to the revised manuscript.*

In fact we have not added this reference. We do not focus in detail on the mechanisms of the Fe-OM associations later in the manuscript so the reference is superfluous to the Introduction.

**P3 Line 13: Michel et al, Science 316, 2007; Hiemstra et al, GCA 105, 2013: $Fe_{10}O_{14}OH_2 \cdot nH_2O$**

*We will clarify that there are several formulas for ferrihydrite (and indeed several models for its structure) in the literature. The simplified formula we use is taken from Raiswell, Elements 2011.*

The Hiemstra reference is added (Introduction paragraph 2).

**P3 Line 25: Fh binding sites are overwhelmingly Fe-O groups, not Fe-OH**

*We will clarify this statement with references to developments in understanding of the structure and sorption characteristics of ferrihydrite. Relevant to the referee's comment, the original OH-rich model for ferrihydrite presented by Drits et al. (Clay Minerals, 28, 1993) has indeed been superseded by more recent works implying a dominance of Fe-O bonds in the structure and thus in sorption sites on the mineral surface, as outlined in Hiemstra, GCA 105, 2013.*

The Hiemstra reference is added (Introduction paragraph 2).

**P3 Line 29: net**

*The text will be modified accordingly.*

The text has been modified (Section 1).

**P5 Line 28: I can imagine that the precision and accuracy of standards may differ from data obtained from solids on filters. Were complete filters combusted, or parts? Were there any replicates for filter analyses?**

*This issue is addressed in detail in our response to the referee's main comments.*

The revised text takes this comment into consideration.

**P6 Line 9: Here I have the same question as above, about the relationship between results from standards and results from (pieces of) filters.**

*This issue is addressed in detail in our response to the referee's main comments.*

The revised text takes this comment into consideration.

**P6 line 10: I do not follow: if total Fe on filter is measured, and the filtered volume is known, why is TSS needed to convert from umol Fe on filter to umol Fe/L water?**

*The referee is correct that TSS is not needed to perform this calculation. In our spreadsheets we had used TSS first to estimate the concentration of $Fe_{part.}$ as µmol/g suspended solid material, and subsequently converted this value back to $Fe_{part.}$ in µmol/L. We now checked the direct conversion as*

*described by the referee and the results are consistent with those initially calculated. We will reword*
*this section.*

The revised text is reworded as stated above.

**P8 Line 15: And, in light of the possible role of FeS, maybe spell out that it's a low estimate for sulfide-associated Fe.**

*This issue is addressed in detail in our response to the referee's main comments.*

The calculations concerning FeS and pyrite are now modified (equations 3-7).

**P8 Line 21: No info on standards.**

*This issue is addressed in detail in our response to the referee's main comments.*

A selection of relevant reference spectra is presented in Figure 7.

**P10 Line 4: Based on measurements of gravimetric water loss?**

*The assumed constant value of 90% is of course a simplification of the true porosity profile, but sufficient for the qualitative discussion of diagenetic processes presented in the original paper. For completeness, we will adjust the $CH_4$ concentration data using a best-fit line through the porosity data derived from gravimetric water loss, which are indeed available.*

The revised manuscript includes more detailed calculations and discussion of diagenetic processes and their rates, hence the generation of gravimetric water loss data – and the subsequent conversion to porosity – are properly described, and the results are used in the calculations (Sections 3.4, 3.12, 3.13).

**P12 Line 8: Are the stacked 100% plots necessary?**

*It is difficult to convey clearly all the information in this large dataset (multiple operational fractions + multiple stations, Fe + organic matter dynamics) in a single figure. We decided to include the stacked 100% plots to improve our chances of succeeding in this. For example in the case of the plot of the organic matter fractions, the absolute concentrations show clearly that there are lower values in the stations close to the sill, while the stacked 100% plot shows that the relative concentrations of terrestrial and phytoplankton material are unaffected by the presence of the sill (i.e. the samples plot approximately where expected in the offshore trend).*

The revised manuscript retains the stacked 100% plots for the reasons given.

**P12 Line 8: Well, station B stands out but for the rest I do not see much of a (general) trend.**

*We will adjust the text accordingly.*

The text has been adjusted (Section 4.5).

**P12 Line 14: The "marked" difference does not become apparent, because the phases normally associated with the stages are not mentioned here. For clarity, I would refer to the operational**

521 **fractions above (L12, 13) as it is the first mention of the "stages" in the results section (instead**
522 **of later on in L25 and 27), mention the corresponding stages and then keep referring to the**
523 **stages.**
524
525 *This issue is addressed in detail in our response to the referee's main comments.*
526
527 The interpretation of the extraction data has been substantially revised (Table 3, Sections 4.6, 4.7).
528
529 **P12 Line 19: Any way to substantiate this? Yu et al used XAS, here it's Mossbauer without**
530 **any Fe-OM reference phase?**
531
532 *This issue is addressed in detail in our response to the referee's main comments.*
533
534 The interpretation of the Mössbauer data has been substantially revised (Sections, 4.7, Figure 7).
535
536 **P12 Line 25: Consider the order of section 4.5 and the place of this paragraph; would it not fit**
537 **better before the Mossbauer results?**
538
539 *We will consider this advice when restructuring the manuscript.*
540
541 The order has been restructured according to the distinction between labile and refractory phases
542 (Sections 4.6, 4.7).
543
544 **P13 Line 15: Why H2S in the text, when HS- (the major species at circumneutral pH) is used**
545 **in the formula?**
546
547 *We will correct this inconsistency.*
548
549 We now use exclusively $H_2S$ as given in the formulations of Reed et al. L&O 56, 2011.
550
551 **P13 Line 18: And why H2S here, while HS- in Eq 5?**
552
553 *We will correct this inconsistency.*
554
555 We now use exclusively $H_2S$ as given in the formulations of Reed et al. L&O 56, 2011.
556
557 **P13 Line 20: What does "efficiently" mean in this context? Rapid reaction?**
558
559 *Yes, we will clarify this.*
560
561 The text related to the diagenetic reactions has been modified substantially (Section 4.8).
562
563 **P13 Line 21: vertical extent**
564
565 *We will adjust the text.*
566
567 The text related to the diagenetic reactions has been modified substantially (Section 4.8).
568
569 **P14 Line 6: Why is there no direct evidence for OM flocculation for this study? Fig 3b presents**
570 **data, but does not show DOM/POM trends along the gradient.**

571

*This is because the original experimental design was focused on Fe and POM did not include measurements of DOM in 2 dimensions June 2015. The DOM data were added later to help explain the distribution of $N/C_{POM}$ and $\delta^{13}C_{POM}$.*

The above response answers the Referee's question.

**P14 Line 10: I find it hard to understand this remark and its context, some more detail would be beneficial. What kind of variations in end-member values?**

*We will expand this section briefly. The statement refers to the fact that the freshwater DOM endmember characteristics (and magnitude) are temporally variable, for example in response to the seasonal cycle or discharge events. In estuaries with a long freshwater residence time this variability may be transmitted downstream slowly, meaning that an instantaneous sampling of an entire transect for DOM characteristics may reveal not only the steady-state mixing dynamics, but also the signal of the changing freshwater endmember. Hence interpretations of conservative vs. non-conservative mixing need to be made with care. These concepts are discussed at length in Asmala et al., Frontiers in Marine Science 12, 2016.*

In fact we have not expanded on this issue due to the fact that the manuscript is now more focused towards Fe behavior independent of DOM and does not require extensive discussion of this issue. We refer the Referee to the above response and citation for further relevant information.

**P14 Line 15: I wonder about effects of changes in pH and DO, that may affect the kinetics of Fe oxidation and precipitation. Are there jumps in these values when going from the river into the estuary?**

*This issue is addressed in detail in our response to the referee's main comments.*

In the revised manuscript we emphasize the partial decoupling of Fe from DOM during flocculation (see revised Title, Abstract, Introduction paragraph 4, Section 5.2), emphasizing some of the factors listed by the Referee.

**P15 Line 5: Crucial. Unsure about the reasoning; Can it simply be the "natural" distribution between DOM and POM for plant material?**

*This issue is addressed in detail in our response to the referee's main comments.*

We refer the Referee to our original response above.

**P16 Line 13: This is the smoking gun; but is it syngenetic or diagenetic? And, there is no direct evidence for Fe-OM from the data, there is just a pool of "undocumented Fe" that is assumed to be organic-bound Fe based on XAS data from Yu et al. I assume that siderite was in the library? Dithionite-citrate may have potential for siderite dissolution? More info on the standards explored is necessary to validate the claim that the Fe was undocumented and thus likely associated with OM.**

*This issue is addressed in detail in our response to the referee's main comments.*

The interpretation of both extraction and Mössbauer data are substantially revised (Sections 4.5, 4.6, 4.7, Figures 5, 6, 7). The undocumented Fe (II) phase at Station A is now thought to be wüstite and in fact a pollution signal from the nearby blast furnace (Fig. 1).

**P16 Line 18: This is pretty speculative (based on Yu); any evidence for the fact that it occurs in the water column? Following sentences infer the role of diagenesis, the Fe(III)-OM is purely hypothetical?**

*This issue is addressed in detail in our response to the referee's main comments.*

The interpretation of both extraction and Mössbauer data are substantially revised (Sections 4.5, 4.6, 4.7, Figures 5, 6, 7). The undocumented Fe (II) phase at Station A is now thought to be wüstite and in fact a pollution signal from the blast furnace.

**P16 Line 33: OK, so siderite was a standard (which is a Fe(II) mineral)**

*Yes. Siderite was one of the reference spectra. This issue is addressed in detail in our response to the referee's main comments.*

A selection of relevant reference spectra is given in Figure 7.

**P16 Line 33: Even fresh ferrihydrite dissolved very slowly at the pH of Na acetate (4.5), are there lit examples of Fh dissolving under those conditions?**

*This issue is addressed in detail in our response to the referee's main comments.*

We have adjusted our interpretation of the extraction behavior of ferrihydrite (Section 4.6, Table 3).

**P17 Line 5: This is interesting, in that redox suttling causes maximum Fh (the most labile and easily reduced Fe(III) phase) accumulation in the surface sediment, where H2S already seems to accumulate (Fig. 7). Perhaps some more words (beyond "seasonally oxygen depleted) can be spent on the equlibrium of a state with apparently co-occurring abundant Fh and H2S accumulation?**

*This issue is addressed in detail in our response to the referee's main comments.*

The revised text includes a discussion of this issue (Section 4.6).

**P18 Line 13: Is there not potential for a role of sediment DOC/POC as Fe(II) sink, i.e. diagenetic OM-Fe(II) formation?**

*Diagenetic formation of OM-Fe complexes is indeed suggested in the papers of Lalonde et al. (2012) and Shields et al. (2016). i.e. according to their model, the association between Fe and OM occurs after sedimentation. Due to the close association of Fe and OM in the water column of boreal estuaries, we have focused on the idea that Fe and OM are transferred together to the sediments. However we cannot discount the possibility that diagenesis affects the nature of the association and we will acknowledge this in the revised text.*

We do not focus on the mechanism of the Fe-OM association in sediments in the revised version. Indeed with the data available it is not possible to state whether the associations implied by the 1M

HCl extraction data are pre- or post-depositional and this is stated in Section 5.5. We worked with the hypothesis that at least a fraction of flocculated Fe is deposited as Fe (III)-OM and that subsequent transformations of this material occur in the sediment.

**P19 Line 3: The mechanism could be explained more explicitly: how does the recovery drive increased Fe and DOM transport?**

*We will expand this section briefly. The mechanism is related to the ionic strength of freshwaters and consequent residence time of DOM in drainage systems.*

We have added the phrase "and elevated ionic strength" in Section 5.6).

**P20 line 25: (**

*We will correct the typo.*

Done.

 **2.Responses to online discussion Referee #2**

692

693 The following document contains both the responses given during the Open Discussion of
694 "Flocculation of dissolved organic matter controls the distribution of iron in boreal estuarine
695 sediments" and a description of the subsequent revisions leading to the submission of "Impacts of
696 flocculation on the distribution and diagenesis of iron in boreal estuarine sediments".

697

698 **[Referee comments in bold]**

699

700 *[Responses in italics]*

701

702 [Indication of subsequent revisions in regular text]

703

704 **This is a paper on the removal mechanisms of Fe to sediments in the riverine-estuarine**
705 **transition of a long riverine estuary on the Eastern Baltic Sea and the effect of this removal for**
706 **the corresponding iron diagenesis in the underlying sediment. The key message of this paper, if**
707 **I understand this right, is that riverine Fe is removed by flocculation in the riverine-estuarine**
708 **transition of the Eastern Baltic Sea. These results are similar to the Fe story presented in Yu et**
709 **al (2015) Chemical Geology in another Finnish estuary. The flocculation occurs as an organic**
710 **Fe complex.**

711

712 **These results are based on the interpretation of Mössbauer spectra, which revealed an unknown**
713 **Fe phase that was neither well-crystallized silicate, magnetite, nor ferrihydrite. The inference**
714 **is therefore not direct, but indirect, and that is a major shortcoming.**

715

716 **In the underlying sediments of the outer estuary and the Baltic Sea, Fe accumulation occurs as**
717 **different phases than in the riverine part, where more crystalline Fe hydroxides occurred than**
718 **in the distal part where organic Fe dominates.**

719

720 **In principal I believe some of the story, e.g., the organic-Fe transport and rapid removal in the**
721 **estuary. But I failed to see how the results translate into different diagenetic Fe processes in the**
722 **sediment.**

723

724 **In particular, I felt that the story on the anaerobic oxidation of methane by iron was**
725 **underdeveloped. While this is an impressively large data set from many stations, much of the**
726 **potential novelty hinges upon the interpretation of the Mössbauer data. The combination of**
727 **Mössbauer/extraction data call for a major reinterpretation of the operational Fe extractions,**
728 **in particular of a re-assessment of the dithionite extraction as an organic Fe phase. This has**
729 **large ramifications for many published papers and the authors need to be very careful in their**
730 **assessment and interpretations.**

731

732 *We thank the referee for the careful consideration of our manuscript. Below, we respond to each of*
733 *the question posed by the referee.*

734

735 **I would like the authors to address and comment on a number of questions I have:**

736

737 **Why do you not present Mössbauer spectra of riverine material, dissolved and particulate?**
738 **This would be most interesting to see.**

739

*We agree with the referee that such data would be extremely interesting. However the present study was not designed to include Mössbauer analysis of riverine dissolved and particulate material. This would require far greater volumes of water to be filtered (and with various grades of filters) than that required for the determination of total $Fe_{diss}$. and $Fe_{part}$. Indeed, this would constitute a separate study in itself. However in response to the comments of both referees we are currently analyzing additional sediment samples by Mössbauer. These include a sample from the river bed of the Mustionjoki, upstream from Station A, which may shed some light on the composition to material transported in the river.*

The revised manuscript includes a Mössbauer spectrum from a river bed sediment sample taken at Station 'a' (Figs. 1, 7 and Supplementary Figure 2).

**Why do you claim that hydroxylamine-extracted Fe occurs as Fe2+, when hydroxylamine is a strong reducing agent? No information on prior oxidation state is possible using this extraction method.**

*It is true that hydroxylamine-HCl is a reducing agent and in fact we did not claim in the original text that hydroxylamine-HCl extracts $Fe^{2+}$. We assume the referee is referring to our claim that dithionite (another reducing agent) extracts $Fe^{2+}$ from OM-Fe(II) complexes. In response to the comments of both referees we have addressed the issue of solubility of various Fe phases in our original extraction scheme extensively. Please refer to Figures R2–R5 and associated text in this file.*
The Fe (II) and Fe (III) components of the labile Fe fraction are now deconvolved on the basis of the additional extractions (Fig. 6, Section 4.6).

**Why do you not even consider or discuss the extraction of FexSy phases with dithionite? This is well known. Not all Fe may be organically associated to the same extent throughout the transect and not necessarily as Fe2+, because dithionite is also a reducing agent.**

*The exact nature of the dithionite-soluble phase(s) remain difficult to determine conclusively, and we require the additional Mössbauer data to advance this discussion. The additional extractions suggest that unsulfidized Fe (II), including the potential OM-Fe (II) complexes, is in fact dissolved in Stages 1+2 of the Poulton and Canfield method, rather than Stages 3+4 (dithionite + oxalate) as suggested in the original manuscript. With respect to the possible dissolution of FexSy phases in dithionite, pyrite is not considered to be dithionite-soluble (see for example Berner, Amer. J. Sci. 268, 1970; Canfield, GCA 53, 1989; Raiswell et al., Chem. Geol., 111, 1994) while greigite is not expected to be a major phase in the sediments at this location. The role of FeS has been established by the additional extractions: this is expected to be dissolved in Stages 1+2 (Fig. R3).*

The revised manuscript presents a version of Fig. R3 (as Fig. 6a) and also presents the interpretation given in the above statements (Section 4.6). Additionally, we now suggest that the major dithionite/oxalate-soluble phases observed at Station A are wüstite and magnetite, derived from industrial inputs and identified on the basis of the Mössbauer spectra (Sections 4.7 and 5.4).

**In your table or on the Mössbauer spectra you should show the patterns for FexSy or FeS2 phases to convince the reader that the composite spectrum is not influenced by these phases.**

*This is a fair criticism and similar to comments from Referee #1. We will present all relevant reference spectra in the revised version.*

We now present reference spectra from a range of potential sedimentary Fe phases alongside the sample spectra (Fig. 7). We discuss the potential interference of pyrite with superparamagnetic Fe (III), among other limitations of the Mössbauer approach for detecting minor Fe phases (Sections 3.10 and 4.7).

**The Mössbauer spectroscopy standardization and reference spectra are not explained. It is not clear why your conclusion is that the phase must be an organic Fe2+ phase. If it is ferrihydrite associated with organic matter, what kind of association is this?**

*This is a fair criticism and similar to comments from Referee #1. We will present all relevant reference spectra in the revised version. The allocation of Fe(II) vs. Fe (III) to unknown phases in Mössbauer spectra is made according to the position of the spectrum in x-y space of quadrupole splitting vs. isomer shift, as outlined in Murad and Cashion, Springer 2004. This detail will be included in the revised version.*

We now present reference spectra from a range of potential sedimentary Fe phases alongside the sample spectra (Fig. 7). The principles regarding the interpretation of Mössbauer spectra are given substantially more weight (Section 3.10).

**Is there a possibility that the unknown spectrum is an amorphous Fe-silica phase?**

*We have no basis to suspect this at the present time.*

We now suggest that the previously unknown Fe phase is wüstite (Fig. 7, Section 4.7). As reported by Manning et al., Can. Mineral. 18, 291–299, 1980, amorphous ferric silicate phases display spectra consistent with superparamagnetic Fe (III).

**Why is it that there are hardly any changes in Fe speciation at Station D, although the S content changes so significantly and therefore likely the concentrations of FeS and FeS2?**

*We apologize to the referee, but we made a significant mistake in the plotting of Figure 6 (and associated text) which is relevant to this question and likely influenced the referee's understanding of the data. The scale of the sedimentary Fe content reads 0–4 %, while it should read 0–40 %. Hence the true figure should look like this:*

[Figure]

Fig. R6. Corrigendum to Figure 6 in the original manuscript. Note the scales on the axes of weight % Fe in the sediments (0–40%).

*Note that all other plots of sedimentary Fe content in µmol/g units are correct, as are the Fe/Al weight ratios given in Fig. 8.*

*Clearly, the Fe content of the sediments at all stations is far higher than the S content, also on a molar basis, and this is the reason that the downcore changes in S content are not reflected in changing Fe speciation. This is best illustrated in Fig. R2 of this document, where "sulfide-Fe" (calculated as described in the manuscript assuming all S as pyrite) is shown to be only a minor component. This conclusion does not change when the calculation is performed taking into account the AVS (FeS) component now calculated in the additional extractions. For clarity we will add both molar and weight % scales to the figures in the revised version.*

We apologize again for this error. The data are now correct and the contribution of S-bound phases to total Fe is made clear in Figs. 5 and 6.

**What is the major Fe carrier down the river, i.e., what is the speciation of Fe in river and estuarine water? How much is associated with the organic fraction – how much is present as ferrihydrite – what is the exact molecular association?**

*For the reasons outlined earlier, we did not carry out speciation work on the suspended particulate (or operationally "dissolved") Fe. We emphasize again the relevance of the study of Neubauer et al., ES&T 47, 2013, which discusses the speciation of Fe in boreal rivers.*

The revised manuscript emphasizes that a significant proportion of total water-borne Fe is likely present as discrete colloidal ferrihydrite, which influences the partial decoupling from terrestrial OM during flocculation and sedimentation (Section 5.3). However we simply do not have the data to quantify the relative amounts of Fe-OM and ferrihydrite in the present study.

**Figure 5 figure caption and left figure panel don't agree. The caption reads Fe2+/FeT, the other Fe3+/FeT ratio.**

*We thank the referee for this observation and will correct the text and figure.*

The presentation of the Mössbauer data has been modified substantially (Fig. 7, Supplementary figure 2, Table 4).

**Figure 6: The down-core operational Fe profiles don't exactly make sense in light of the pronounced changes in S content with depth at Station A. At least, the 100% scaling makes it difficult to associate the species changes with the S concentration changes. I recommend to show the Fe species as concentrations, e.g., as summed bars totaling to the actual Fe concentration. That would help at least for comparing the data of Station A. The Fe species do not correspond at all to the sulfur concentrations. How is this possible, if FeS/FeS2 forms? My conclusion would be that the dithionite-extracted species seem to be associated at least partly with some FeS/FeS2. This needs to be accounted for.**

*Again, we apologize to the referee for the mistake in Figure 6 which is relevant to this comment (see earlier response). We will take onboard the suggestion to plot the Fe speciation data as concentrations. A version of Fig. R2 will be included in the revised manuscript.*

In the revised manuscript, the data are now correct and the contribution of S-bound phases to total Fe is made clear in Figs. 5 and 6. The 100% summation of Fe phases is removed from the downcore plots (Fig. 5 in the revised version) to make visual comparisons with the S profile easier.

**p.14 l.25: I think the authors mean 'isotopically depleted'**

*No. The deeper water samples are indeed more isotopically enriched (less negative values than surface water samples).*

This text has remained unchanged for the reason given.

**p.16, l.16-17 Fe-Si amorphous phases; FeS mackinawite-like material?: Why Fe2+, could also have been Fe3+?**

*We will present all relevant reference spectra in the revised version. The allocation of Fe(II) vs. Fe (III) to unknown phases in Mössbauer spectra is made according to the position of the spectrum in x-y space of quadrupole splitting vs. isomer shift, as outlined in Murad and Cashion, Springer 2004. This detail will be included in the revised version.*

See earlier comment. There is now more detail on Mössbauer principles in Section 3.10 and reference spectra are presented in Fig. 7 and Supplementary Figure 2.

**p.17, l.1-5 hydroxylamine hydrochloride is a strong reducing agent suggesting that the Fe could have been reduced by the extraction.**

*It is true that hydroxylamine-HCl is a reducing agent and is specifically used to target poorly crystalline Fe oxides in this scheme. We do not claim in this passage of text that hydroxylamine-HCl extracts $Fe^{2+}$.*

The revised manuscript presents a comprehensive new interpretation of the labile Fe phases (Section 4.6).

**I would also like to see a transect plot of DOM and POM.**

*We do have a transect plot of POC from the same samples used to generate the corresponding plots for N/C and $\delta^{13}C$. (Fig. R7). This can be included as a supplement if needed but does not contribute significant extra information. The distribution strongly resembles N/C, suggesting that phytoplankton dominates the POC pool at the time of sampling in June 2015.*

[Figure]

*Fig. R7. POC along the transect in June 2015 (mg/L)*

*Unfortunately we do not have a corresponding plot for DOC (or DOM) as this sampling was not included in the original experimental setup (which was focused on Fe and POM).*

This plot is visible in the online discussion, but since it is not referred to in the revised manuscript we do not intend to upload it as a supplement unless requested to do so by the Referee or Editor.

**p.18. Are the salinity differences significant enough to impact the Fe-S system? I don't think so.**

*We disagree. The changes in sulfate penetration depth from Station A–D–J (Fig. 7) indicate a significant impact of bottom water salinity on the diagenetic zonation of the sediments. As highlighted*

*in the discussion of this Figure, the depth of the SMTZ, and the intensity of the associated $H_2S$ peak,*
*contrast strongly between the sites. As mentioned also by Referee #1, the distribution of $H_2S$ in the*
*pore waters (between stations and the downcore profile at each station) should then have a strong*
*impact on the stability of Fe phases. We will devote more lines to this in the revised version.*

The revised manuscript includes a substantially expanded description and quantitative evaluation of
the diagenetic processes (Sections 3.13, 4.8, 5.5).

**p.18 l.24 what do you mean by background rates: How are these background rates? Aren't**
**these the major reactions? What about sulfate reduction rates and coupled oxidation of sulfide**
**by ferrihydrite?**

*We will rephrase the sentence to remove the term "background".*

*Undoubtedly, both sulfate reduction coupled to organic matter oxidation, and oxidation of sulfide by*
*ferrihydrite – along with many other diagenetic reactions – are also occurring at these sites. We will*
*expand the discussion to give a broader overview of the various diagenetic processes, including first*
*order estimates of process rates derived from pore water profiles. However it was not (and is still*
*not) our intention to perform a detailed diagenetic modeling study, rather to highlight the diagenetic*
*zones that are clearly visible in the pore water profiles, in order to qualitatively discuss the*
*differences that are observed along the transect (and the link to Fe inputs from flocculation).*

The revised manuscript includes a substantially expanded description and quantitative evaluation of
the diagenetic processes (Sections 3.13, 4.8, 5.5).

**The anaerobic oxidation of methane by iron is often invoked these days, but to argue for this**
**process there has to be good direct evidence. I am sure the authors are aware that AOM also**
**can be coupled indirectly to reduction of iron oxyhydroxides through sulfide oxidation, but**
**cannot be distinguished easily without performing specific experiments. Concentration profiles**
**alone are not enough. The authors should refrain from inferring that AOM by iron is a major**
**process controlling deep iron diagenesis when they have not addressed sulfide oxidation**
**processes. They do not even present DIC data to support their assertion. In addition, should**
**this process occur, it is easy to assess the quantitative significance by assessing the methane flux**
**and the required removal of Fe to account for methane oxidation.**

*We are indeed aware of the alternative indirect pathways by which reduction of Fe oxyhydroxides*
*may be coupled to methane oxidation. The most relevant is of course the so-called cryptic sulfur cycle*
*as described in Holmkvist et al., GCA 75 (2011), in which downward-diffusing $H_2S$ from the SMTZ*
*is oxidized by Fe oxyhydroxides, leading to the formation of native sulfur and subsequent*
*disproportionation to $SO_4^{2-}$ and $H_2S$ (the $SO_4^{2-}$ then going on to oxidize $CH_4$). We do not dispute the*
*validity of the mechanism presented by those authors and indeed coupled sulfur cycling and methane*
*oxidation have been confirmed by further studies (e.g. Milucka et al., Nature 491, 2012).*

*However, as discussed at length in the review process for Slomp et al. Plos ONE 8, 2013 and Egger*
*et al., ES&T 49 (2015) for sediments from the Bothnian Sea, in the low-salinity systems of the northern*
*Baltic the SMTZ is sufficiently shallow that $H_2S$ diffusing downwards from the SMTZ is exhausted*
*well above the base of a typical GEMAX core. This is confirmed in the profiles in Figure 6, where*
*$H_2S$ is undetectable below approx. 20 cm at Stations D and J. It is therefore problematic to invoke*
*the cryptic sulfur cycle as the cause of high pore water $Fe^{2+}$ at depth, because the downward diffusion*
*of $H_2S$ is the ultimate driver of this process. Although $H_2S$ is regenerated during sulfur*

*disproportionation within the cryptic sulfur cycle, only three moles are produced for every four moles of $H_2S$ that initially react with Fe oxyhydroxides. So cryptic sulfur cycling is in fact a net consumer of $H_2S$ and cannot sustain $H_2S$-driven Fe oxyhydroxide reduction well below the downward-penetrating $H_2S$ front. This concept is nicely illustrated in the study of Egger et al., Biogeosciences 13, 2016 for Black Sea sediments (see their Figure 8, where pore water $Fe^{2+}$ production due to cryptic sulfur cycling is shown to be restricted to a narrow depth interval just below to the downward-penetrating $H_2S$ front). Hence we are confident that the large increase in pore water $Fe^{2+}$ observed in the deeper parts of the cores in our study are not driven by cryptic sulfur cycling.*

*To clarify, in the original manuscript we also give two possible mechanisms for the production of pore water $Fe^{2+}$ (Eq. 7 and 8), thereby acknowledging that both dissimilatory reduction of Fe oxyhydroxides, and Fe-AOM, may be active in the deep sediments. Hence coupled to the above discussion re. cryptic sulfur cycling, we disagree with the assertion that too much weight is given to the likelihood of Fe-AOM.*

*In related work we are performing experiments to determine the rates of AOM in these sediments, but these results are beyond the scope of the current paper.*

The revised manuscript includes a substantially expanded description and quantitative evaluation of the diagenetic processes (Sections 3.13, 4.8, 5.5).

**Along these lines, generally there is also too little discussion on sulfide oxidation coupled to iron reduction in the surface sediments. These organic-rich sediments likely have very low oxygen penetration depths of a few mm. Based on many other studies in estuarine systems, it is likely that anaerobic degradation processes such as sulfate reduction commence in the first centimeter. This makes it possible that FeS phases already occur in the topmost cm, and that not only iron reduction, but co-existing iron and sulfate reduction take place in the topmost cm.**

*The referee is correct that oxygen penetration is in the order of mm, and that both sulfate reduction and Fe reduction likely co-occur in the uppermost cm. This is implied by the immediate decline in pore water sulfate below the sediment-water interface at all sites, and the basic principle that Fe oxide reduction is more than twice as energy efficient as sulfate reduction per mole carbon oxidized (Stumm and Morgan, Wiley, 1981) and therefore should activate at a shallower depth horizon. In reality the diagenetic zones in these sediments overlap extensively, which in the surface sediments is also related to bioturbation and bioirrigation processes including by the invasive polychaete Marenzellaria (e.g. Norkko et al., Glob. Ch. Biol. 18, 2011). The referee is also correct that FeS may be present in the surface cm, either formed in situ by the co-occurrence of sulfate reduction and Fe reduction, or transported vertically from deeper horizons by the "smoothing" action of bioturbation. All of these concepts will be included in an expanded discussion of the diagenetic processes in the revised version of the manuscript.*

The revised manuscript includes a substantially expanded description and quantitative evaluation of the diagenetic processes (Sections 3.13, 4.8, 5.5).

**Finally, although the authors do very well in describing the bathymetry of this estuarine system, they fail to associate the bathymetric features with the current transport/ hydrography and the resulting particle transport and accumulation. For example, Station D likely must be influenced by saline water transport upstream, which is the only explanation to explain the higher bottom salinities. Therefore, inshore/upstream transport of organic material and of Fe has to be considered for the deep depressions.**

*It is true that saline inflows across the sill are responsible for the relatively high salinity deep water of the inner estuary. Inflows typically occur in winter as established in early literature on this system (e.g. Virta, Nordic. Hydrol. 8, 1977) and Niemi (Meri, 4, 1977). We will include an expanded introduction to the hydrodynamics in Section 2 (Study location). We are also aware of the likely lateral transport of Fe and OM (indeed Page 17, Lines 4–14 of the original manuscript specifically address this issue). We will rephrase this section to emphasize that lateral transport into the bathymetric depression of Station D may occur in both and upstream and downstream direction. Figure 4 will also be modified to highlight the position of the bathymetric depression and the sill in the data series of OM and Fe.*

The revised manuscript includes the expanded background information about the hydrography and saline inflows (Section 2), and acknowledges that currents may play a role in transporting Fe from the sill towards Station D (Section 5.3).

**In addition, Station D, being the deepest station of the inner estuary, is clearly a particle trap of fine-grained material, also of organic material. As such, the focusing and accumulation of material here may override the estuarine mixing signal the authors have as their overriding study target. Stations C or E may be more informative in this context. Do the authors note the same signals at stations C, D, and E?**

*As outlined in the previous response we are aware of the potential effects of lateral transport and focusing at Station D and will emphasize these more clearly in the revised version. In terms of the relative effects of the focusing and estuarine mixing on the observed signals, and the comparison of Stations C, D, and E: It could indeed be argued that Stations C and D display slightly elevated concentrations of "phytoplankton-derived" OM relative to the offshore trend (see Fig. 4b, lower left panel). We will give more weight to this in the revised version. The missing Fe data from Station C makes the equivalent assessment more difficult in the case of Fe.*

See previous response. The revised manuscript acknowledges that both Fe and organic matter are likely transported downslope towards Station D, and discusses the decoupling of Fe from terrestrial OM that occurs during this process (Section 5.3).

**3. Marked up revised manuscript**

Flocculation of dissolved organic matter controlsImpacts of flocculation on the distribution 
[revised manuscript text omitted]

---

## Referee Report (RR1)

I must compliment the authors on this revision: all the reviewers' points have been taken into serious consideration, relevant new data have been supplied and the authors have shown to be open to drastic reinterpretation of their data. The chosen methodology still does not allow for unequivocal identification of Fe-OM (which is hard to detect in general; the article could actually serve as a catalyst for more Fe-OM standards besides Fe oxalate to be measured by Mossbauer?). But the parallel use of chemical and spectroscopic methods for identification of Fe phases paints a much more balanced and detailed picture of Fe diagenesis than in the previous version. The level of detail on Fe extraction protocols and speciation may be a bit much for some readers, but is definitely food for thought for Fe junkies. In addition, there is more balance regarding the mechanisms of Fe flocculation and transport to the sediment surface, in particular the (initially overly dominant) role of OM in these processes.

As was the original, the revised manuscript is very well-written. Due to the length of the author's replies and the final manuscript, I have only focused on the sections that have undergone serious changes, where I have not been able to find a single type-o. I would therefore support publication of this revised manuscript as is; if there are any technical corrections necessary that I missed they will undoubtedly be dealt with during the proofing stage.

One last comment, regarding p23, L10: the work of Dos Santos Afonso (Langmuir, 1992) and Poulton (GCA, 2004) adds some further insight into the reaction between Fe(III) oxides and dissolved sulfide, for instance the generation of a sulfide radical and subsequent Fe(III) reduction by that radical. But that is likely superfluous for this manuscript, and does not alter the general diagenetic framework (it may only change the reaction stoichiometries).

---

## Author Response (AR2)

**Point-by-point response to Referee #2 comments on Version #2**

*Referee* The authors have made very substantial changes to the first version and the revised version has been much improved. The authors have also thoroughly reinterpreted their data and have drawn new conclusions compared to the first version. Sedimentary iron species concentration data were recalculated and corrected and Mössbauer spectra at Station A were re-interpreted as inorganic iron oxides instead of organic Fe. Altogether, this is a very far-reaching re-write, which casts the data in a new light. I would like to thank the authors for their pro-active attitude. I still have some remaining concerns about the interpretation of the organic Fe (II) phase, which is still prominent in the abstract, because the authors rely on an operational Fe extraction without more direct evidence (see below). Since the manuscript is new in many parts, I have some comments that arise from this revised version that I would like to see addressed:

What I find confusing in this version and which I hope the authors can improve is that the extensive methods description to extract labile and refractory Fe infers inorganic Fe phases, but not one organic Fe phase, although their suggested presence is one of the major conclusions of the manuscript. Mössbauer spectra are only presented for Fe-oxalate. The fit of the spectra, however, does not suggest that Fe-oxalate is abundant in the sediment, not really a surprise. The way the authors conclude that organic Fe(II) are dominant is thus by negative evidence and mainly based on their calculation procedure to determine non-sulfidized Fe (II) i.e., based on the correct assessment of AVS-Fe from the sulfide extraction and the fact that ALL 1M extracted Fe can be classified as labile. Since this is an operational definition, the interpretation of a non-accounted difference in Fe as organic Fe remains tentative.

*Response* We thank the Referee for another careful consideration of our work. It is true that the applied methodology does not include reagents reported to target organic-bound metals, such as $NH_4$-EDTA (Scheinost et al., Environ. Sci. Technol. 36, 2002) or $H_2O_2$ (Tessier et al., An. Chem. 51, 1979). However as for all extraction approaches, the problem of operational definition would remain even if we had included these. Rather, the sequential extraction protocols were selected for comparability with previous studies in nearby settings, and to give an overview of all Fe components, not only organic-bound. The 1M HCl extraction was employed by Yu et al. (2015) in exactly the same manner as described here, to determine sulfidized and unsulfidized Fe (II) components of 1M HCl-soluble Fe. In that study, 1M HCl-soluble Fe accounted for 50-70 % of total Fe in sediments (their Figure 3f). Approximately 40-50% of total Fe was unsulfidized 1M HCl-Fe (II) (their Figure 3d). Subsequent XAS analysis showed that the co-ordination environment of Fe in bulk samples could be modeled using a humic acid standard loaded with Fe (II), together with silicate and oxide standards in a multi-component model. The humic acid standard contributed 30-40% to the bulk Fe spectrum (their Table 3), similar to the contribution of unsulfidized Fe (II) in the extraction, and therefore implying that Fe(II)-OM is soluble in 1M HCl. The plausible alternatives for unsulfidized labile Fe (II) (i.e. carbonates and phosphates) were present in negligible concentrations according to their XAS and other data. Clearly, more experimental work is required to determine the solubility of various organic Fe complexes in sediment extraction protocols, but considering the proximity of the study area of Yu et al. (2015) to our sites, we apply the same interpretation of the extraction data.

Section 3.9 is rephrased to include the following: "*Subsequently, 5 additional samples from each of the 2015 cores (Stations A, D, and J) were subjected to a 1 hour room-temperature 1M HCl extraction (Burton et al., 2011), to further investigate the labile Fe phases (Table 2, "Subsample 2"), potentially including Fe (II)-OM and Fe (III)-OM complexes as described in Yu et al. (2015).*" Table 2 also includes elucidation of the possible 1M HCl-soluble phases in the caption: "*\*Sulfidized 1M HCl-soluble Fe (II) is assumed to be present as FeS (AVS). \*\* Unsulfidized 1M HCl-soluble Fe (II) may include carbonates (e.g., siderite, ankerite), phosphates (e.g., vivianite) and labile Fe (II)-OM. \*\*\*1M HCl-soluble Fe (III) includes poorly crystalline Fe oxides (ferrihydrite, lepidocrocite) and labile Fe (III)-OM*".

*Referee* I am also concerned that the freeze-drying process alters the mineralogy and leads to some oxidation. This could have reduced the AVS-sulfide concentration and could be an explanation why non-sulfide Fe (II) is so high (and, consequently, interpreted as organic Fe (II)).

*Response* This is not a concern. The additional extractions for AVS (and other labile phases) were performed on samples that were stored frozen, and then dried in a nitrogen-filled glove bag at room temperature (see Section 3.9). i.e. these samples were not freeze-dried in the same way as those extracted by the Poulton and Canfield (2005) method. Drying from frozen under nitrogen is a commonly used method for preserving AVS during preparation of sediment samples (e.g. Kraal et al., GCA 122, 2013) and this was the specific reason to use these samples for this exercise. Furthermore, had any oxidation of FeS occurred during sample storage we would not expect to measure unsulfidized Fe (II); rather the Fe (II) released by the oxidation should be rapidly converted to Fe oxides, as shown by the relative rate constants of the following reactions (modified from Reed et al., L&O 56, 2011):

$2O_2 + FeS$ à $SO_4^{2-} + Fe^{2+}$ (K = 300 mmol$^{-1}$Lyr$^{-1}$)

$Fe^{2+} +0.25O_2 + 2HCO_3^- + 0.5 H_2O$ à $Fe(OH)_3 +2CO_2$ (K = 140000 mmol$^{-1}$Lyr$^{-1}$)

*Referee* Finally, the preservation of these substantial amount of organic Fe (II) in light of the observation that DOM and Fe are decoupled during estuarine mixing and not deposited with each other seem like a contradiction with the assessment of abundant organic Fe, in particular since concentration profiles of Fe (II) interpreted as organic fe (II) show an increase below the sediment surface.

*Response* We wish to clarify that we have not used the terms *substantial* or *abundant* in connection with the absolute amount of Fe-OM in the sediments. Our treatment of the potential relevance of Fe-OM in this version is quantitative and fair, based on the measurements of unsulfidized Fe (II) - which we have no basis to suspect are incorrect (see previous response) - and the pore water and Mössbauer evidence arguing against carbonates and phosphates. With respect to the contrast between Fe and OM distributions in the estuary, the possible mechanisms for this are discussed extensively in Sections 5.2 and 5.3. We emphasize the use of the term *partial* decoupling, because this does not preclude that some component of Fe and OM remains co-associated throughout flocculation, sedimentation and diagenesis.

*Referee* There are likely also reactive silicate phases involved in the early diagenetic Fe cycling.

*Response* We considered the literature on this topic in the interpretation of the Mössbauer data (see Section 4.7). The studies we found focused on the formation of amorphous ferric aluminosilicates in association with biogenic silica dissolution at the sediment surface, and we have included these phases in Table 3 as likely to be present in our samples. On the other hand we do not suspect Fe (II) silicates to contribute to 1M HCl-soluble Fe (II). Reactive sheet silicates require boiling concentrated HCl (Stage 5 of the Poulton and Canfield (2005) extraction) for dissolution (Table 2). Furthermore we are not aware of studies reporting amorphous Fe (II) silicates in sediments from the Baltic Sea.

*Referee* Station A is inferred to contain a substantial portion of smelter-derived Fe. This needs to be made clearer in the methods description part, since it is also presented as a potentially extracted component with 1M HCl.

*Response* We now make first reference to the blast furnace in the Study Location (Section 2): "*A blast furnace located near the town of Åminnefors at the mouth of the Mustionjoki river (Fig. 1a) was active from the late 19th century until 1977. The blast furnace and associated waste materials serve as a potential source of anthropogenic Fe pollution to the estuary.*" Since the mineralogical composition of the pollution Fe was not known *a priori*, it cannot be stated in the methods descriptions which pollution Fe phases are expected to be extracted with which reagents. After wuestite and magnetite were identified by Mössbauer, their expected extraction behavior was listed in Table 3.

***Referee*** In Figure 5 Station A shows concentrations of Fe up to 28% by weight of sediment (> 5000 µmol Fe/g sediment). This is unusually high, as noted by the authors. A simple calculation: If the Fe mineral phase were the most Fe-rich mineral possible, i.e., FeO (MW = 88 g/mol; with atom Fe proportion = 0.7), then 28% Fe by weight translates to 36% of the weight of the whole sediment represented by the FeO mineral at this depth. For Magnetite the proportion is 39%. It seems very unusual that more than every third mineral in this sediment is one of the two minerals, even if they are smelter-derived. Was material dumped in the estuary?

***Response*** We also performed these calculations and this was one line of reasoning supporting the revised interpretation of the previously unidentified Fe (II) phase in the Mössbauer data as wuestite (the Fe:C ratio is unfeasibly high for Fe-OM complexes). The data are correct; we checked the Fe content of the ICP solutions by the 1,10 phenanthroline method. Thus, indeed the concentrations are remarkably high and could indicate dumping or erosion of slag rather than simply aerosol transport. Section 5.4 now includes the statement: "*The magnitude of the enrichments of these minerals at Station A suggest that waste material may have been dumped or eroded directly into the estuary*".

***Referee*** I think it is noteworthy that the apparent dilution of the other sediment components by this smelter-Fe is not visible in the total S or organic C profiles. Further, I advise against using any of the Station A sediment data – solid-phase or porewater to generalize overall Fe dynamics, because of this contamination anomaly.

***Response*** We agree that the apparent lack of a dilution signal in the S and C data is strange. We observe a clear dilution effect in other major sediment components including Ca and Al (see below), implying that the clay mineral matrix was indeed diluted by the pollution material, as well as coincident enrichments of additional metals that may be associated with the pollution (e.g. Mn). In the case of S, it can be argued that authigenic precipitation of sulfides from pore waters will proceed regardless of the composition of the sediment, and hence that the concentration of S is not affected by the dilution. For C however this would not apply. One possibility would thus be that the pollution material also contains C, although we cannot verify this.

The Station A data will remain in the manuscript. Despite the contamination, it is remarkable that the pore water profiles show essentially the same zonation as observed at the other sites. Hence, the pollution material appears to be quite inert in terms of the major diagenetic processes. This is now stated in Section 5.5: "*
[revised manuscript text omitted]

(a)                          (b)

| Class | Mineral | |
|-------|---------|---|

[Figure]

Isomer shift

Quadrupole splitting

Silicate Fe (II)    Magn. (3⁺)    Unspecified Fe (III)
SP Fe (III)         Magn.(2.5⁺)   ○ Raw data
Wüstite             Hematite      Sum of subspectra

**2-line Fe (II) spectra**

| Silicate | Fayalite | River bed (Station 'a') |
| | Chlorite | Station A (0-1 cm) |
| | Mica | Station A (26-28 cm) |
| Phosphate | Vivianite | |
| Carbonate | Siderite | Station D (0-1 cm) |
| Organic | Ferrous oxalate | Station D (26-28 cm) |
| Oxide | Wüstite | |
| Sulfide | Pyrite | Station J (0-1 cm) |

**2-line Fe (III) spectra**

| Oxy-hydroxide | Ferrihydrite (SP) | Station J (30-32 cm) |
| Oxide | Goethite (SP) | |
| Organic | Ferric oxalate | |

-4 -3 -2 -1 0 1 2 3 4 5          -4 -3 -2 -1 0 1 2 3 4 5
Velocity (mms⁻¹)                  Velocity (mms⁻¹)

[revised manuscript text omitted]